# Improving Discrete Diffusion Unmasking Policies Beyond Explicit Reference Policies

**Chunsan Hong[1], Seonho An[2], Min-Soo Kim[2], Jong Chul Ye[1]**
[1]Graduate School of AI, KAIST     [2]School of Computing, KAIST

## Abstract

Masked diffusion models (MDMs) have recently emerged as a novel framework for language modeling. MDMs generate sentences by iteratively denoising masked sequences, filling in [MASK] tokens step by step. Although MDMs support any-order sampling, performance is highly sensitive to the choice of *which position to unmask next*. Prior work typically relies on rule-based schedules (e.g., max-confidence, max-margin), which provide ad hoc improvements. In contrast, we replace these heuristics with a *learned* scheduler. Specifically, we cast denoising as a KL–regularized Markov decision process (MDP) with an explicit reference policy and optimize a regularized objective that admits policy-improvement and convergence guarantees under standard assumptions. We prove that the optimized policy under this framework generates samples that more closely match the data distribution than heuristic schedules. Empirically, across four benchmarks, our learned policy consistently outperforms max-confidence: for example, on Sudoku, where unmasking order is critical, it yields a 20.1% gain over random and a 11.2% gain over max-confidence. Code is available at `https://github.com/chunsanHong/UPO`.

## 1 Introduction

Diffusion models have achieved remarkable success in domains such as image generation (Ho et al., 2020; Song et al., 2021), by defining a noise-adding forward process and a denoising reverse process in continuous space, and learning the reverse dynamics within this framework. Recent studies (Nie et al., 2025; Ye et al., 2025; Lou et al., 2024) have advanced this line of work into the discrete domain, applying it to language modeling through masked diffusion models (MDMs). In the discrete setting, MDMs define a forward process that replaces tokens in text with a [MASK] token and a corresponding reverse process; training typically maximizes an evidence lower bound (ELBO) (Lou et al., 2024) to fit the reverse dynamics. Small-scale MDMs (Lou et al., 2024; Sahoo et al., 2024) have matched or surpassed autoregressive models (ARMs), and recent large-scale studies (Ye et al., 2025; Nie et al., 2025) report results competitive with strong ARMs such as LLaMA-3 (Grattafiori et al., 2024).

Despite a shared template, diffusion in discrete versus continuous spaces differs substantially—especially at inference. Continuous-space diffusion models (Song et al., 2021; Karras et al., 2022) perform denoising by integrating a stochastic differential equation (SDE) or an ordinary differential equation (ODE). In contrast, discrete-space diffusion models (Sahoo et al., 2024; Nie et al., 2025) denoise by iteratively sampling concrete replacements for [MASK] tokens. These mechanisms lead to different update geometries: continuous models accumulate numerous small changes across all dimensions, whereas discrete models enact localized, high-magnitude jumps at a few coordinates. Consequently, effective denoising in discrete space requires its own methodology. Just as advances in SDE solvers unlocked large gains for continuous diffusion, improved denoising strategies are likely to yield substantial performance gains for MDMs.

This raises a natural question: what matters most in MDM denoising? Among several factors, we focus on the choice of *which position to unmask next*. Kim et al. (2025) provide a theoretical rationale for this choice, proving that no polynomial-time algorithm can solve any-order generation; *i.e.*, we cannot train MDMs that exactly recover the real data distribution for all masked sentences. However, they empirically show that selecting unmasking tokens via max-confidence (Chang et al., 2022) or

max-margin (Kim et al., 2025) at inference time can bypass sub-hard instances and even outperform ARMs. Reflecting this, contemporary large-scale MDMs such as LLaDA or Dream-7B (Nie et al., 2025; Ye et al., 2025) commonly adopt max-confidence.

In this paper, we aim to move beyond heuristic references (e.g., max-confidence) by *learning* improved unmasking policies. To this end, we cast MDM denoising as a Markov decision process (MDP) and train an unmasking position-selection policy using the group relative policy optimization (GRPO) (Shao et al., 2024). Theoretically, we prove that under this framework, the optimized policy produces samples closer to the true data distribution than the strong heuristic references. Moreover, we introduce *three* tractable surrogate objectives and provide propositions that justify optimizing them in place of the ideal, yet intractable, objective. Practically, our learned policy substitutes for max-confidence and related heuristics, achieving improved accuracy across four benchmarks. For example, on SUDOKU, where unmasking order is crucial, it delivers a 20.1% improvement over random and a 11.2% improvement over max-confidence.

**Related works.** Reinforcement learning (RL) has emerged as an effective tool to fine-tune large language models toward complex objectives or preferences (Ouyang et al., 2022; Guo et al., 2025; Shao et al., 2024; Rafailov et al., 2023). In MDMs, Zekri & Boullé (2025) apply policy gradient methods to a pretrained discrete diffusion model so as to maximize a task-specific reward at the last denoising step. Zhao et al. (2025) adopts the GRPO framework to improve the reasoning ability by rewarding only the final answer accuracy. Zhu et al. (2025) propose a variance reduced preference optimization (VRPO) for MDMs, which shares the concept of direct preference optimization (DPO) (Rafailov et al., 2023) in ARMs. While the aforementioned works utilize RL to post-train the MDM itself, they differ from ours, where we train the unmasking policy model. Recently, Huang et al. (2025) introduce DCoLT, which also trains an unmasking policy via reinforcement learning. Our approach differs in that we formulate denoising as a KL–regularized MDP with an explicit *reference policy* and optimize a regularized objective that admits policy-improvement and convergence guarantees under standard assumptions. Furthermore, while Huang et al. (2025) jointly applies RL to train both the unmasking policy and the underlying MDMs for maximum accuracy, our work focuses solely on optimizing the unmasking policy for frozen MDMs, thereby reducing computational cost and mitigating the risk of reward over-optimization.

## 2 PRELIMINARY

**Notations.** Let the distribution $p_{\text{data}}$ be the data distribution over sequences of length $L$ with vocabulary $\mathcal{V} = \{1, ..., m\}$, and let the set of all possible sequences be $\mathcal{X} = \mathcal{V}^L$. We define $p_L$ as the Kronecker delta distribution that assigns all probability mass to the sentence in which every token is masked. We denote the kernel forms of $p_{\text{data}}$ and $p_L$ by $\mathbf{P}_{\text{data}}$ and $\mathbf{P}_L$, respectively, with entries $[\mathbf{P}_{\text{data}}]\boldsymbol{x} = p_{\text{data}}(\boldsymbol{x})$. We write $\boldsymbol{x}_n = \{x_n^1, x_n^2, \ldots, x_n^L\}$ for the $n$-th step sequence with $L$ tokens. We denote the mask by $\mathbb{M}$ and the set of sequences with $n$ masks by $\mathcal{X}_n = \{\boldsymbol{x} | \boldsymbol{x} \in \{\mathcal{V} \cup \{\mathbb{M}\}\}^L, \sum_{i=1}^L \mathbf{1}(x^i = \mathbb{M}) = n\}$, so that $\boldsymbol{x}_n \in \mathcal{X}_n$ has $n$ masks and we refer $a^i[\boldsymbol{x}]$ for the $i$-th mask index of $\boldsymbol{x}$. For example, if the number of masks in $\boldsymbol{x}$ is $n$, then $a^i[\boldsymbol{x}] \in \{1, \ldots, L\}, x^{a^i[\boldsymbol{x}]} = \mathbb{M}$ and $i \in \{1, \ldots, n\}$. For brevity, we will omit $\boldsymbol{x}$ in $a^i[\boldsymbol{x}]$ for the rest of the paper. We denote MDMs as $\theta$ and their model distribution as $\pi_\theta$. We denote Markov transition kernel $\mathbf{T} : \mathcal{X} \times \mathcal{X} \to [0, 1]$ where $\mathbf{T}(\boldsymbol{x}, \boldsymbol{x}')$ is the probability of $\boldsymbol{x}$ transform into $\boldsymbol{x}'$ such that $\sum_{\boldsymbol{x}' \in \mathcal{X}} \mathbf{T}(\boldsymbol{x}, \boldsymbol{x}') = 1$ for all $\boldsymbol{x}$.

**Masked diffusion models.** Masked Diffusion Models (Shi et al., 2024; Sahoo et al., 2024) define a mask-adding forward process and corresponding reverse process in discrete space, and learn the model $\pi_\theta$ to mimic the true reverse process. The forward process gradually masks tokens independently one-by-one in $\boldsymbol{x}_0$ until the sequence is fully masked at $\boldsymbol{x}_n$ with $n = L$ (or $t = 1$ where $t := n/L$). For $t = n/L \in (0, 1)$, the sequence $\boldsymbol{x}_n$ is partially masked, with each being masked with probability $t$ or remaining unmasked with probability $1 - t$. The reverse process recovers the data distribution by iteratively predicting masked tokens as $t$ decreases from 1 to 0. While previous works (Campbell et al., 2022; Sahoo et al., 2024) propose the time-dependent MDM training framework, the recent works (Zheng et al., 2025; Ou et al., 2025) propose the time-agnostic model where time is implicitly included in the number of masks. Then, the time-agnostic loss for MDM is defined as follows:

$$\mathcal{L}_{\text{MDM}}(\theta) \triangleq -\mathbb{E}_{n, \boldsymbol{x}_0, \boldsymbol{x}_n} \left[ \tfrac{1}{n} \sum_{i=1}^L \mathbf{1}[x_n^i = \mathbb{M}] \log \pi_\theta(x_0^i | \boldsymbol{x}_n) \right]. \tag{1}$$

Zheng et al. (2025); Ou et al. (2025) have shown that $\mathcal{L}_{\text{MDM}}$ is the upper bounds of the negative log-likelihood. Zheng et al. (2025) propose the simple generation algorithm, named the First-Hitting sampler, beyond a continuous time Markov chain (CTMC) that follows the conventional reverse process. First-Hitting sampler randomly selects the index $i$ for unmasking and samples the next token sequentially as $\boldsymbol{x}_L \to \boldsymbol{x}_{L-1} \to \cdots \to \boldsymbol{x}_0$. This is proven to converge to $p_{\text{data}}$. Recent large-scale MDMs (Nie et al., 2025; Ye et al., 2025) currently adopt time-agnostic modeling and further employ sampling strategies such as the max-confidence beyond the First-Hitting sampler.

**Markov transition kernel of random sampler and training hard problem of MDM.** Large-scale MDMs first select the masked token to unmask and sample from the categorical distribution. See Algorithm 1 in Appendix B.2 for the detailed sampling algorithm. We here formulate general Markov transition kernels defined by various samplers of MDM.

$$\mathbf{T}_n^{(g)} = \sum_{i=1}^{n} \mathbf{T}_n^{i,(g)}, \quad \boldsymbol{x}_{n-1} \sim \sum_{i=1}^{n} [\mathbf{T}_n^{i,(g)}](\boldsymbol{x}_n, \boldsymbol{x}) \tag{2}$$

where

$$[\mathbf{T}_n^{i,(g)}](\boldsymbol{x}_n, \boldsymbol{x}) = \begin{cases} g(a^i|\boldsymbol{x}_n) \cdot \pi_\theta(x^{a^i}|\boldsymbol{x}_n) & \text{if } \boldsymbol{x}_n^{-a^i} = \boldsymbol{x}^{-a^i}, \\ 0 & \text{otherwise}, \end{cases} \tag{3}$$

$$\tag{4}$$

where $\sum_{i=1}^{n} g(a^i|\boldsymbol{x}_n) = 1$, $\boldsymbol{x}^{-a^i} = \{x^1, x^2, \ldots, x^{a^i-1}, x^{a^i+1}, \ldots, x^L\}$, $x_n^{a^i} = \mathbb{M}$ by definition, and $\pi_\theta(x_{n-1}^{a^i}|\boldsymbol{x}_n)$ is the categorical distribution of model prediction of the $i$-th mask of $\boldsymbol{x}_n$.

The non-homogeneous Markov kernel for First-Hitting sampler (Zheng et al., 2025) for time-agnostic MDMs can be defined as choosing $g$ by

$$g(a^i|\boldsymbol{x}_n) := g_{\text{rand}}(a^i|\boldsymbol{x}_n) = \frac{1}{n} \tag{5}$$

In algorithmic manner, this is equal to first sample $i \sim \mathcal{U}\{1, \ldots, n\}$, then sample $x^{a^i} \sim \pi_\theta(x^{a^i}|\boldsymbol{x}_n)$, and obtain $\boldsymbol{x}_{n-1}$ by replacing $i$-th mask of $\boldsymbol{x}_n$ with $x^{a^i}$. In this case, $(\prod_{n=1}^{L} \mathbf{T}_n^{g_{\text{rand}}}) \cdot \mathbf{P}_L \approx \mathbf{P}_{\text{data}}$ if $\pi_\theta$ is perfectly estimated (Zheng et al., 2025). However, Kim et al. (2025) have proven that learning $\pi_\theta$ for every combination of masked sequence cannot be solved by a polynomial algorithm, and there exist hard subproblems. This can interpreted as there exists some points that true reverse process and model's reverse process are different, such that resulting in $(\prod_{n=1}^{L} \mathbf{T}_n^{g_{\text{rand}}}) \cdot \mathbf{P}_L \neq \mathbf{P}_{\text{data}}$. We provide a simplified example to help understand:

**Example 1.** *Consider the following simplified zebra (Einstein) puzzle. There are two houses, and the task is to determine the person living in each house and their favorite food. The possible persons are* Robert *and* Tom*, and the possible foods are* pizza *and* hamburger. *The clues are given as follows: 1) The person living in the first house is Robert, and 2) Tom likes pizza.*

Now consider the MDM directly answer the questions, filling the mask of sentence: *"House 1: name: [MASK], food: [MASK], House 2: name: [MASK], food: [MASK]"*. Obviously, if MDM tries to answer the name of the person who lives in the first house, the problem becomes trivial. However, if it tries to fill attributes for the second house first before filling attributes of the first house, the problem becomes much harder. As the number of questions and clues grows, this kind of discrepancy would be much larger. Indeed, the zebra puzzle is also known as an NP-hard problem (Shah et al., 2024; Yato & Seta, 2003). Note that this is a highly simplified example to help understand the problem of MDM training, and refer to Appendix A for more details and Kim et al. (2025) for a complete explanation.

**Practical approach to avoid such hard problems.** Meanwhile, Kim et al. (2025) empirically shows that unmasking the token with max-confidence or max-margin helps to avoid such hard problems, leading to better generations. Therefore, in practice, rather than randomly sampling the token for unmasking, large-scale discrete diffusion models (Nie et al., 2025; Ye et al., 2025) utilize such rule-based samplings. The weighted kernel related to the max-confidence sampling can be defined with $g(a^i|\boldsymbol{x}_n) := g_{\text{conf}}$ defined as follows:

$$g_{\text{conf}}(a^i|\boldsymbol{x}_n) = \mathbf{1}\big(a^i = \arg\max_{a^j}(\max_{c \in \mathcal{V}} \pi_\theta(x_{n-1}^{a^j} = c|\boldsymbol{x}_n))\big), \tag{6}$$

or in other stochastic forms,

$$g_{\text{conf}^\tau}(a^i|\boldsymbol{x}_n) = \frac{\sum_{c\in\mathcal{V}}\exp(\pi_\theta(x_{n-1}^{a^i}=c|\boldsymbol{x}_n)/\tau)}{\sum_{a^j}\sum_{c\in\mathcal{V}}\exp(\pi_\theta(x_{n-1}^{a^j}=c|\boldsymbol{x}_n)/\tau)}, \quad \text{s.t.} \lim_{\tau\to0+}g_{\text{conf}^\tau}=g_{\text{conf}}, \tag{7}$$

$$g_{\text{Top-K}}(a^i|\boldsymbol{x}_n) = \frac{1}{K}\cdot\mathbf{1}(a^i\in\underset{a^j}{\operatorname{argtopk}}(\max_{c\in\mathcal{V}}\pi_\theta(\boldsymbol{x}_{n-1}^{a^j}=c|\boldsymbol{x}_n))), \quad \text{s.t.} \ g_{\text{Top-1}}=g_{\text{conf}}. \tag{8}$$

where $\operatorname{argtopk}$ denotes the indices from the top-k sampler. Similarly, the weight kernel for max-margin sampler can be defined with $g_{\text{margin}}$ whose value is 1 where $i$ is the mask index with the maximum margin between the first-confident and second-confident token.

## 3 IMPROVING UNMASKING POLICY BEYOND A REFERENCE POLICY

While twisted Markov transition kernels driven by heuristic unmasking policies—such as $g_{\text{conf}}$ or $g_{\text{margin}}$—deliver substantial gains, we posit that a stronger policy $g$ can guide MDM denoising more effectively. Motivated by this, *we learn $g_\phi$ instead of defining $g$ in a heuristic manner to find an optimal path*. We further cast the learning problem of $g_\phi$ as a KL-regularized MDP, where a strong heuristic sampler serves as a reference policy, enabling stability and policy improvement guarantees.

### 3.1 EXISTENCE OF OPTIMAL PATH BEYOND $g_{\text{conf}}$

We test various $g$ in GSM8K (Cobbe et al., 2021), the mathematical questions benchmark, to probe whether max-confidence is merely strong or whether substantially better unmasking paths exist. We compare three schedulers: $g_{\text{conf}}$, $g_{\text{rand}}$, and $g_{\text{Top-K}}$. Here, $g_{\text{conf}}$ is deterministic, while $g_{\text{Top-K}}$ smoothly interpolates between random and max-confidence. For controlled comparison to analyze the effect of generation ordering and following the default setting used in LLaDA (Nie et al., 2025), we conducted token sampling with argmax operator, *i.e.*, $x_{n-1}^{a_n} = \arg\max_{x\in\mathcal{V}}\pi_\theta(x^{a_n}=x|\mathbf{x}_n)$.

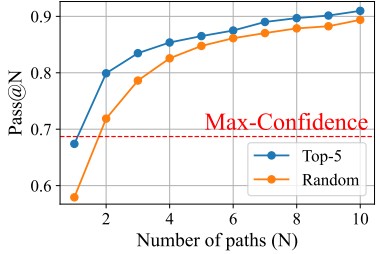

Figure 1: Pass@N on GSM8K. The dashed red line is the single-trajectory accuracy of max-confidence $g_{\text{conf}}$.

For each test question, we draw $N$ independent trajectories using either $g_{\text{rand}}$ or $g_{\text{Top-5}}$ and report Pass@N, the fraction of questions for which at least one trajectory is correct. Figure 1 shows three consistent trends: (i) $g_{\text{conf}}$ significantly outperforms $g_{\text{rand}}$ when $N=1$; (ii) $g_{\text{Top-5}}$ dominates $g_{\text{rand}}$ across $N$, highlighting the strength of confidence-based ordering; (iii) as $N$ grows, both curves surpass the max-confidence line by a clear margin, and $g_{\text{Top-5}}$ exceeds 0.92 when $N=10$.

These results carry two implications. First, max-confidence is indeed a strong single-path baseline. Second, the rise of Pass@N above the max-confidence line indicates the presence of higher-yield unmasking trajectories than the deterministic max-confidence order. Brute-force search for such paths is computationally prohibitive and requires unobserved oracle feedback, motivating our learned policy $g_\phi$ that aims to discover high-reward paths without enumerating many trajectories.

### 3.2 REFORMULATING THE PROBLEM AS REINFORCEMENT LEARNING

Motivated by the strength of $g_{\text{conf}}$ and the existence of higher-yield unmasking paths, we set two goals: (i) increase the probability that an MDM produces the correct answer, and (ii) move the induced sampling distribution closer to $p_{\text{data}}$ than a strong heuristic reference $g_{\text{ref}}$. We therefore learn a stochastic position-selection policy $g_\phi$ rather than fixing $g$ heuristically. The policy is parameterized as a softmax over masked positions using a learnable scorer $h_\phi(\boldsymbol{x})\in\mathbb{R}^L$:

$$g_\phi(a^i\,|\,\boldsymbol{x}_n) = \frac{\exp\big([h_\phi(\boldsymbol{x}_n)]_{a^i}\big)}{\sum_{i'=1}^n\exp\big([h_\phi(\boldsymbol{x}_n)]_{a^{i'}}\big)}. \tag{9}$$

We cast unmasking as a regularized Markov decision process (MDP) with verifiable terminal rewards. Each episode begins from a prompt–answer pair where the answer is fully masked, $[\boldsymbol{x}_L:\mathbf{q}]$ with

$\mathbf{q} \sim \rho_Q$ referring to the answer distribution and $\boldsymbol{x}_L = \mathbb{M}^L$, and terminates when all masks are removed at $n = 0$, yielding $\boldsymbol{x}_0 \in \mathcal{X} = \mathcal{V}^L$. At state $\boldsymbol{x}_n$ the action space is the set of masked indices $\mathcal{A}_{\boldsymbol{x}_n} = \{a^1[\boldsymbol{x}_n], \ldots, a^n[\boldsymbol{x}_n]\}$; the policy $g_\phi(\cdot \mid \boldsymbol{x}_n)$ selects one index $a_n \in \mathcal{A}_{\boldsymbol{x}_n}$ to unmask. Transitions factor through the fixed MDM denoiser $\pi_\theta$:

$$p_{g_\phi}(\boldsymbol{x}_{n-1} \mid \boldsymbol{x}_n) = \mathbf{T}_n^{g_\phi}(\boldsymbol{x}_n, \boldsymbol{x}_{n-1}) = g_\phi(a_n \mid \boldsymbol{x}_n) \cdot \pi_\theta(\boldsymbol{x}_{n-1} \mid \boldsymbol{x}_n, a_n),$$

where $\pi_\theta(\boldsymbol{x}_{n-1} \mid \boldsymbol{x}_n, a_n)$ is the token distribution for position $a_n$. The agent thus controls *where* to unmask, while the environment dynamics are induced by $\pi_\theta$. Rewards are verifiable: an external checker evaluates the fully unmasked output $\boldsymbol{x}_0$ and returns $r(\mathbf{q}, \boldsymbol{x}_0) \in \{0, 1\}$.

To stabilize and accelerate training, we introduce a strong *reference policy* $g_{\mathrm{ref}}$ (e.g., Top-$K$) and optimize a theoretical KL-regularized GRPO-style objective (Mroueh, 2025). We optimize directly over the *terminal-output distribution* induced by the policy: all expectations are taken with respect to $p_g(\boldsymbol{x}_0 \mid \mathbf{q})$, and the regularizer compares terminal-output marginals. Accordingly, our theoretical loss is an *output-level* GRPO objective without clipping:

$$\max_\phi \ \mathbb{E}_{\mathbf{q} \sim \rho_Q} \left[ \mathbb{E}_{\boldsymbol{x}_0 \sim p_{g_{\phi_{\mathrm{old}}}}(\cdot|\mathbf{q})} \left[ \frac{p_{g_\phi}(\boldsymbol{x}_0 \mid \mathbf{q})}{p_{g_{\phi_{\mathrm{old}}}}(\boldsymbol{x}_0 \mid \mathbf{q})} \, A(\mathbf{q}, \boldsymbol{x}_0) \right] - \beta \, D_{\mathrm{KL}}\big( p_{g_\phi}(\boldsymbol{x}_0|\mathbf{q}) \, \| \, p_{g_{\mathrm{ref}}}(\boldsymbol{x}_0|\mathbf{q}) \big) \right],$$
$$(10)$$

where $A(\mathbf{q}, \boldsymbol{x}_0)$ is a standardized advantage. Let $r_g(\mathbf{q}) = \mathbb{E}_{\boldsymbol{x}_0 \sim p_g(\cdot|\mathbf{q})}[r(\mathbf{q}, \boldsymbol{x}_0)] \in [0, 1]$ and $\mathrm{std}_g(\mathbf{q}) = \sqrt{\mathbb{E}_{\boldsymbol{x}_0 \sim p_g(\cdot|\mathbf{q})}[(r(\boldsymbol{x}_0) - r_g(\mathbf{q}))^2]}$, then $A(\mathbf{q}, \boldsymbol{x}_0) = (r(\mathbf{q}, \boldsymbol{x}_0) - r_{g_{\phi_{\mathrm{old}}}}(\mathbf{q}))/(\mathrm{std}_{g_{\phi_{\mathrm{old}}}}(\mathbf{q}) + \epsilon)$. This objective follows the *output-level* GRPO loss used in DeepSeek-R1 (Guo et al., 2025) rather than the original *token-wise* GRPO used in DeepSeekMath (Shao et al., 2024). The regularizer keeps $g_\phi$ close to $g_{\mathrm{ref}}$, acting as a trust-region that mitigates instability while enabling improvement over a strong baseline; selecting an appropriate $g_{\mathrm{ref}}$ provides a high-accuracy starting point and accelerates convergence.

## 3.3 THEORETICAL ANALYSIS

In this section, we provide how $g_\phi$ optimizing Eq. (10) can guarantee the convergence of the policy model and superior performance over the reference policy $g_{\mathrm{ref}}$. Mroueh (2025) have proven that a policy model learning output-level GRPO loss (Eq. (10)) can converge to its optimal performance $r_{g^*}$, which is higher than $r_{g_{\mathrm{ref}}}$:

**Theorem 1** (Restatement of GRPO convergence theorem (Mroueh, 2025)). *Assume $1 > r_{g_{\mathrm{ref}}} > 0$, define for $\beta > 0$:*

$$h_{\epsilon, p_{\mathrm{ref}}}(r_g) = \frac{1}{1 + \frac{1 - r_{g_{\mathrm{ref}}}}{r_{g_{\mathrm{ref}}}} \exp\left( -\frac{1}{\beta} \frac{1}{\sqrt{r_g(1 - r_g) + \epsilon}} \right)} \tag{11}$$

*Then, for the local optimizer $\phi_n = \max_\phi$ Eq. (10) where $\phi_{\mathrm{old}} = \phi_{n-1}$, the probability of success satisfies the following fixed point iteration i.e we have almost surely for all $\mathbf{q}$ for $n \geq 1$,*

$$r_{g_{\phi_n}}(\mathbf{q}) = h_{\epsilon, p_{\mathrm{ref}}}(r_{g_{\phi_{n-1}}}(\mathbf{q})) \tag{12}$$

*and $r_{g_{\phi_0}}(\mathbf{q}) = r_{g_{\mathrm{ref}}}(\mathbf{q})$. Also, let $r_{g^*}$ be a fixed point of $h_{\epsilon, p_{\mathrm{ref}}}$ and assume that have $|h'_{\epsilon, p_{\mathrm{ref}}}(r_{g^*})| < 1$. Given that $h_{\epsilon, p_{\mathrm{ref}}}$ and $h'_{\epsilon, p_{\mathrm{ref}}}$ are continuous in $[0, 1]$, following local fixed point convergence is established:*

$$\lim_{n \to \infty} r_{g_{\phi_n}} = r_{g^*}, \tag{13}$$

*and $r_{g^*} > r_{g_{\mathrm{ref}}}$.*

Therefore, setting $g_{\mathrm{ref}}$ as a strong proxy model is important, as we can always get a better policy $g_{\phi^*}$. Moreover, in the following we prove that policy model $\phi$ maximizing Eq. (10) can restore $p_{\mathrm{data}}$ better than $g_{\mathrm{ref}}$:

**Theorem 2** (Reference-KL Tightening for MDM Policy Improvement). *Assume there exists an ideal MDM $\theta^*$ satisfying MDM $\mathcal{L}_{MDM}(\theta^*) = 0$ for $p_{data}$ where data is composed of prompt and oracle*

answer $\{\mathbf{q}, \mathbf{a}\}$. *Assume we select well-defined reward* $r(\mathbf{q}, \mathbf{a}) \in \{0, 1\}$ *such that* $p_{data}(\mathbf{q}, \mathbf{a}) > 0 \Rightarrow$ $r(\mathbf{q}, \mathbf{a}) = 1$ *and* $0 < r_{g_{ref}} < 1$. *Now consider incomplete MDM* $\theta$ *such that* $\mathcal{L}_{\mathrm{MDM}}(\theta) > 0$, *and* $g_{\phi^*}$ *that satisfies Eq. (13) for* $\theta$. *Let* $g_{\mathrm{ref}}$ *be a policy model such that, for every* $\mathbf{q}$, $p_{g_{\mathrm{ref}}}(\boldsymbol{x}_0 \mid \mathbf{q}) > 0$ *whenever* $p_{\theta^*}(\boldsymbol{x}_0 \mid \mathbf{q}) > 0$, *i.e.,* $\mathrm{supp}(p_{\theta^*}(\cdot \mid \mathbf{q})) \subseteq \mathrm{supp}(p_{g_{\mathrm{ref}}}(\cdot \mid \mathbf{q}))$. *Then, the following inequality holds:*

$$D_{\mathrm{KL}}(p_{\theta^*}(\boldsymbol{x}_0|\mathbf{q})||p_{g_{\phi^*}}(\boldsymbol{x}_0|\mathbf{q})) < D_{\mathrm{KL}}(p_{\theta^*}(\boldsymbol{x}_0|\mathbf{q})||p_{g_{\mathrm{ref}}}(\boldsymbol{x}_0|\mathbf{q})). \tag{14}$$

*Or equivalently,*

$$D_{\mathrm{KL}}(p_{\mathrm{data}}(\mathbf{a}|\mathbf{q})||p_{g_{\phi^*}}(\boldsymbol{x}_0|\mathbf{q})) < D_{\mathrm{KL}}(p_{\mathrm{data}}(\mathbf{a}|\mathbf{q})||p_{g_{\mathrm{ref}}}(\boldsymbol{x}_0|\mathbf{q})). \tag{15}$$

*Proof sketch.* Mroueh (2025) provides policy dynamic under training Eq. (10). Using the result given in Lemma 1 in Appendix, we can express $g_{\phi^*}$ with $g_{\mathrm{ref}}$ and thereby derive KL divergence. Refer to Appendix C.1 for detailed proof and reason why the support assumption holds. $\square$

The interpretation of Theorem 2 is as follows. If we choose a binary reward $r(\mathbf{q}, \mathbf{a}) \in \{0, 1\}$ satisfying $p_{\mathrm{data}}(\mathbf{q}, \mathbf{a}) > 0 \Rightarrow r(\mathbf{q}, \mathbf{a}) = 1$ and $0 < r_{g_{ref}} < 1$, then the theorem guarantees that the KL divergence is strictly tightened compared to the reference policy. Consequently, one may view any reward that satisfies these conditions as a "good" reward in the sense that it provably induces learning dynamics that bring the sampling distribution closer to $p_{\mathrm{data}}$. Under the assumption that the dataset is well constructed, such binary rewards naturally correspond to properties such as correctness, the absence of reasoning errors, grammatical validity, or even combinations thereof, any of which encourage the model to move its output distribution closer to the true data distribution.

Our provided Theorem 2 guarantees the closer sampling to $p_{\mathrm{data}}$ or ideal MDM $\theta^*$ than $g_{\mathrm{ref}}$. Consequently, under the assumptions of two theorems, initializing with $g_{\phi_0} = g_{\mathrm{ref}}$ and repeatedly optimizing $g_\phi$ using Eq. (10) yields a convergent policy $g_{\phi^*}$ that (i) attains a higher expected reward than $g_{\mathrm{ref}}$ and (ii) generates samples that are closer to the real data distribution than those from $g_{\mathrm{ref}}$.

## 3.4 PRACTICAL REALIZATION OF UNMASKING POLICY OPTIMIZATION

Here, we provide practical and tractable training objectives with various $g_{\mathrm{ref}}$. In a real-world setting where $L$ and $|\mathcal{V}|$ are high, Eq. (10) is not directly learnable since $p_{g_\phi}(\boldsymbol{x}_0|\mathbf{q})$ is intractable in MDMs. It requires marginalizing over every trajectory to $\boldsymbol{x}_0$, *i.e.*, $p_{g_\phi}(\boldsymbol{x}_0|\mathbf{q}) = \sum_{\boldsymbol{x}_1, \dots, \boldsymbol{x}_L} p_\phi(\boldsymbol{x}_{0:L}|\mathbf{q})$, making both the main objective (reward maximization) and the KL term in Eq. (10) intractable. To address this, we provide a surrogate loss that can alternate the main objective.

**Proposition 1** (Output–Token Level Gradient Alignment (informal)). *For MDM $\pi_\theta$ and unmasking policy model $g_\phi$ where $p_{g_\phi}(\boldsymbol{x}_{n-1} \mid \boldsymbol{x}_n) = g_\phi(a_n \mid \boldsymbol{x}_n) \cdot \pi_\theta(\boldsymbol{x}_{n-1} \mid \boldsymbol{x}_n, a_n)$ and $p_{g_\phi}(\boldsymbol{x}_0|\mathbf{q}) = \sum_{\boldsymbol{x}_1, \dots, \boldsymbol{x}_L} p_\phi(\boldsymbol{x}_{0:L}|\mathbf{q})$, consider the output level loss $\mathcal{L}_{\mathrm{output}}$ and token-wise policy loss $\mathcal{L}_{\mathrm{token}}$:*

$$\mathcal{L}_{\mathrm{output}}(\phi) = \mathbb{E}_{\mathbf{q} \sim \rho_{\mathcal{Q}}} \mathbb{E}_{\boldsymbol{x}_0 \sim p_{g_{\phi_{\mathrm{old}}}}(\cdot|\mathbf{q})} \Big[ \frac{p_{g_\phi}(\boldsymbol{x}_0 \mid \mathbf{q})}{p_{g_{\phi_{\mathrm{old}}}}(\boldsymbol{x}_0 \mid \mathbf{q})} A(\mathbf{q}, \boldsymbol{x}_0) \Big], \tag{16}$$

$$\mathcal{L}_{\mathrm{token}}(\phi) = \mathbb{E}_{\mathbf{q} \sim \rho_{\mathcal{Q}}} \mathbb{E}_{\{a_{1:L}, \boldsymbol{x}_{0:L}\} \sim p_{g_{\phi_{\mathrm{old}}}}(\cdot|\mathbf{q})} \Big[ \sum_{n=1}^{L} \frac{g_\phi(a_n \mid \boldsymbol{x}_n)}{g_{\phi_{\mathrm{old}}}(a_n \mid \boldsymbol{x}_n)} A(\mathbf{q}, \boldsymbol{x}_0) \Big]. \tag{17}$$

*Assume that we optimize $\phi$ carefully such that $\phi \approx \phi_{old}$, the two gradients are approximately equal:*

$$\nabla_\phi \mathcal{L}_{\mathrm{token}} \approx \nabla_\phi \mathcal{L}_{\mathrm{output}}, \tag{18}$$

*and $\nabla_\phi \mathcal{L}_{\mathrm{token}} = \nabla_\phi \mathcal{L}_{\mathrm{output}}$ at first optimization step where $\phi = \phi_{\mathrm{old}}$.*

*Proof sketch.* Although the result can be readily inferred from standard RL, we provide a rigorous proof for our setting: sparse-reward, finite-horizon, episodic, and non-stationary MDPs where we maximize a GRPO-style objective. The full proof is in Appendix C.2. $\square$

While Eq. (16) optimizes $p_{g_\phi}(\boldsymbol{x}_0|\mathbf{q}) = \sum_{\boldsymbol{x}_1, \dots, \boldsymbol{x}_L} p_\phi(\boldsymbol{x}_{0:L}|\mathbf{q})$ which is intractable, our surrogate loss in Eq. (17) optimizes $g_\phi(a_n \mid \boldsymbol{x}_n)/g_{\phi_{\mathrm{old}}}(a_n \mid \boldsymbol{x}_n)$ which is tractable once trajectory is sampled.

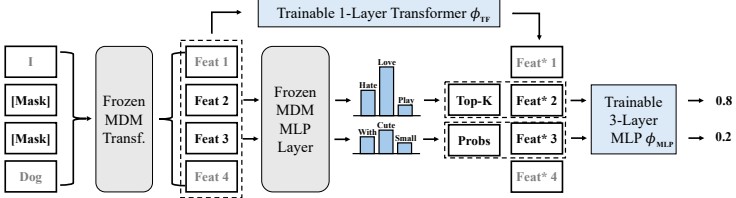

Figure 2: The structure of our unmasking policy model. The model is composed of a single Transformer layer and a 3-layer MLP. In inference time, the model proceeds as follows: 1) Given an input sentence, we run the base MDMs transformer and extract a feature. 2) This feature branches into two paths: first, the base MDMs continue to produce a prediction of token distribution, and second, fed to the policy model's transformer layer. 3) We concatenate the extracted feature with the base MDM's Top-K probabilities. 4) The concatenated feature is then processed by a 3-layer MLP to yield a policy over unmasking positions. The shared structure with frozen MDM supports memory-efficient training (*e.g.*, LLaDA: 8B, policy model: 134M), where we only update the unmasking policy model. The details and memory-efficient training algorithm are provided in Appendix B.2

Table 1: Various realization of $\mathcal{L}_{\text{UPO}}(g_\phi, g_{\text{ref}}, D)$ varying $g_{\text{ref}}$.

| Reference policy $g_{\text{ref}}$ | Max-Confidence $g_{\text{conf}}$ | Softmax Realization $g_{\text{conf}}^\tau$ | Top-K Realization $g_{\text{Top-K}}$ |
|---|---|---|---|
| Divergence $D(p_{g_\phi}\|p_{g_{\text{ref}}})$ | Cross-Entropy Loss | $\mathcal{L}_{\text{KL}}(\boldsymbol{x}_{0:L}, a_{1:L}, \mathbf{q}) = \text{StopGrad}\left(\frac{p_{g_\phi}(\boldsymbol{x}_{0:L}|\mathbf{q})}{p_{g_{\phi_{\text{old}}}}(\boldsymbol{x}_{0:L}|\mathbf{q})} \cdot \left(1 + \log\frac{p_{g_\phi}(\boldsymbol{x}_{0:L}|\mathbf{q})}{p_{g_{\text{ref}}}(\boldsymbol{x}_{0:L}|\mathbf{q})}\right)\right) \cdot \sum_{n=1}^{L} \log g_\phi(a_n \mid \boldsymbol{x}_n)$ | |
| Parametrization $g_\phi(a^i \mid \boldsymbol{x})$ | | $\frac{\exp([h_\phi(\boldsymbol{x})]_{a^i})}{\sum_{j\in\{\text{All Mask indices}\}} \exp([h_\phi(\boldsymbol{x})]_{a^j})}$ | $\frac{\exp([h_\phi(\boldsymbol{x})]_{a^i})}{\sum_{j\in\{\text{Top-K Mask indices}\}} \exp([h_\phi(\boldsymbol{x})]_{a^j})}$ |
| Model initialization $\phi_0$ | | Model Pretrained with Cross-Entropy Loss w.r.t. $g_{\text{conf}}$ | Random Initialization |

Now, we detail how we train the policy model $\phi$ efficiently. Our unmasking policy model structure is shown in Figure 2, and the full training algorithm table can be found in Appendix B.2.

**Final Training Objective.** By using Proposition 1, clipping, and Monte Carlo approximation, we transform the theoretical loss (Eq. (10)) into tractable unmasking policy optimization (UPO) loss as follows:

$$
\mathcal{L}_{\text{UPO}}(g_\phi, g_{\text{ref}}, D) = \mathbb{E}_{\mathbf{q}\sim\rho_Q, \{\boldsymbol{x}_{0:L}^{(g)}, a_{1:L}^{(g)}\}_{g=1}^G \sim p_{g_{\phi_{\text{old}}}}} \left[ \frac{1}{G} \sum_g^G \left( \left(\frac{1}{L} \sum_{n=1}^L \min\left(\frac{g_\phi(a_n^{(g)}|\boldsymbol{x}_n^{(g)})}{g_{\phi_{\text{old}}}(a_n^{(g)}|\boldsymbol{x}_n^{(g)})} A_g, \right.\right.\right.\right.
$$
$$
\left.\left.\left.\left. \text{clip}\left(\frac{g_\phi(a_n^{(g)}|\boldsymbol{x}_n^{(g)})}{g_{\phi_{\text{old}}}(a_n^{(g)}|\boldsymbol{x}_n^{(g)})}, 1-\epsilon, 1+\epsilon\right) A_g \right)\right) - \beta \cdot D(p_{g_\phi}\|p_{g_{\text{ref}}}) \right) \right],
$$
(19)

where $D(p_{g_\phi}\|p_{g_{\text{ref}}})$ should be empirical and tractable estimator of $D_{\text{KL}}\left(p_{g_\phi}(\boldsymbol{x}_0|\mathbf{q}) \| p_{g_{\text{ref}}}(\boldsymbol{x}_0|\mathbf{q})\right)$ and clipped loss has been proven to attain global optimality (Huang et al., 2024). Note that $(g)$ in superscript of Eq. (19) is the group index.

**Realizations of $\mathcal{L}_{\text{UPO}}$ via the choice of $g_{\text{ref}}$.** We consider three reference policies—$g_{\text{conf}}$, $g_{\text{conf}}^\tau$, and $g_{\text{Top-K}}$—and summarize their associated divergence terms in Table 1. In Table 1, we provide the tractable surrogate $\mathcal{L}_{\text{KL}}$ that is justified by Propositions 2 and 3 in Appendix C.3. Since we are sampling the trajectory from old policy model while training, we design the estimator $\mathcal{L}_{\text{KL}}$ that can be measured by the probability of the trajectories, $p_{g_\phi}(\boldsymbol{x}_{0:L}|\mathbf{q})$, $p_{g_{\phi_{\text{old}}}}(\boldsymbol{x}_{0:L}|\mathbf{q})$, and $p_{g_{\text{ref}}}(\boldsymbol{x}_{0:L}|\mathbf{q})$. For softmax realization, since $g_{\text{conf}^\tau}$ assigns nonzero probability to all trajectories, we can replace $D_{\text{KL}}$ with the tractable surrogate $\mathcal{L}_{\text{KL}}$. We parameterize $g_\phi$ as the softmax distribution over all masked positions by default (see Eq. (9)) for softmax realization. For pretraining, although one could initialize by matching $g_{\phi_0}$ to $g_{\text{conf}^\tau}$ via KL minimization, we observe stronger performance with cross-entropy. For $g_{\text{Top-K}}$, there exists some sampled trajectory where $p_{g_{\text{Top-K}}}(\boldsymbol{x}_{0:L}|\mathbf{q}) = 0$ since $g_{\text{Top-K}}$ assigns zero probability to mask index out of Top-$K$. In this regard, we reparameterize $g_{\phi_{\text{Top-K}}}$ as the softmax distribution of $h_\phi$ over the Top-$K$ set only. This reparametrization gives two

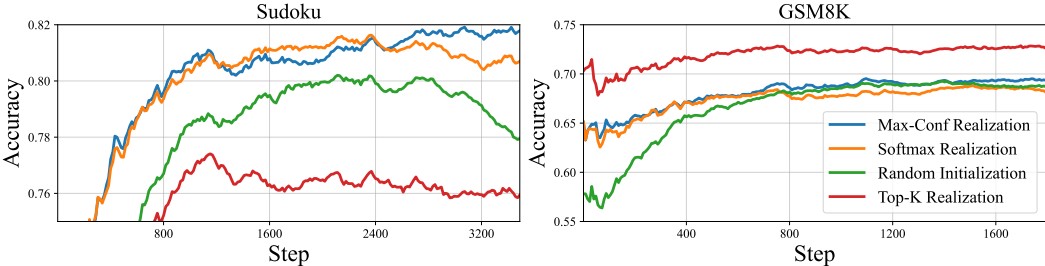

Figure 3: Training accuracy on SUDOKU and GSM8K under $\mathcal{L}_{\mathrm{UPO}}(g_\phi, g_{\mathrm{ref}}, D)$ for different choices of $g_{\mathrm{ref}}$. Reference realizations are summarized in Table 1. "Random initialization" denotes training a randomly initialized model with $\mathcal{L}_{\mathrm{UPO}}(g_\phi, \varnothing, \varnothing)$.

benefits: 1) we can utilize $\mathcal{L}_{\mathrm{KL}}$ since $p_{g_{\mathrm{Top-K}}}$ is non-zero for all trajectory sampled from $p_{g_{\phi_{\mathrm{Top-K}}}}$, and 2) we do not need to pretrain policy model since it is same as $g_{\mathrm{Top-K}}$ when randomly initialized. In contrast, the determinism of $g_{\mathrm{conf}}$ renders $\mathcal{L}_{\mathrm{KL}}$ ill-posed whenever $g_\phi$ assigns zero mass to the chosen index, so we train with $\mathrm{CE}(g_{\mathrm{conf}} \| g_\phi)$ while retaining the regularized-MDP perspective. Full derivations and detailed explanations are provided in Appendix B.1.

## 4 EXPERIMENTAL RESULTS

### 4.1 EXPERIMENTAL SETTING

**Models and benchmarks.** We evaluate on the widely used large-scale MDM LLADA-8B-INSTRUCT (Nie et al., 2025) and four datasets: (i) logic-puzzle benchmarks—SUDOKU (Zhao et al., 2025) and ZEBRA (Shah et al., 2024)—and (ii) mathematical-reasoning benchmarks—GSM8K (Cobbe et al., 2021) and MATH500 (Lightman et al., 2023).

**Generation protocols and rewards.** We utilize LLADA-8B-INSTRUCT (Nie et al., 2025) for all experiments. For logic puzzles, the model receives a prompt and a masked solution; it then denoises by unmasking one position at a time. The reward is the fraction of slots predicted correctly (e.g., in SUDOKU, the proportion of blank cells filled with the correct digits). For mathematical reasoning, the model is given a problem and denoises a length-128 masked sequence. Following the LLADA generation protocol, we partition the 128 positions into four bins of size 32 and perform sampling by each baselines sequentially over the bins. The reward is 1 if the final answer matches the ground truth and 0 otherwise. In addition, we incorporate the sum of log-probabilities produced by $\pi_\theta(\cdot)$ as an auxiliary reward signal. Detailed explanation and settings such as training hyper-parameters are provided in Appendix E. We employ the dense reward rather than the binary reward because it leads to faster convergence and higher accuracy in practice. Nonetheless, we also verify that our proposed method outperforms the max-confidence baseline under the binary reward and report these results in Appendix D.1.

### 4.2 TRAINING DYNAMICS ACROSS REFERENCE POLICIES

We train our policy model on GSM8K and SUDOKU by optimizing $\mathcal{L}_{\mathrm{UPO}}(g_\phi, g_{\mathrm{ref}}, D)$ with three reference realizations from Table 1: $g_{\mathrm{conf}}$, $g_{\mathrm{conf}}^\tau$, and $g_{\mathrm{Top-K}}$. We also report ablation that drops the divergence term, *i.e.* $\mathcal{L}_{\mathrm{UPO}}(g_\phi, \varnothing, \varnothing)$, with a randomly initialized policy, which is conceptually equivalent to DCOLT. The result is shown in Figure 3. The strongest configuration depends on the task: on SUDOKU, max-confidence realization achieves the best accuracy, while on GSM8K, Top-$K$ realization performs best. We conjecture that in GSM8K, restricting exploration to the Top–$K$ indices stabilizes learning under $\mathcal{L}_{\mathrm{UPO}}$ due to the much larger number of masked tokens. Across the two benchmarks, our optimal configuration outperforms the randomly initialized model—that is, DCOLT (green line)—by margins of at least 2% and 3%, respectively.

Table 2: Accuracy of each sampling strategy. Best per row in bold.

| Benchmark | Sampler | | | | |
|---|---|---|---|---|---|
| | Random (Zheng et al., 2025) | Margin (Kim et al., 2025) | Entropy (Ye et al., 2025) | Confidence (Chang et al., 2022) | Ours |
| SUDOKU | 0.616 | 0.713 | 0.671 | 0.705 | **0.817** |
| Zebra | 0.339 | 0.346 | 0.351 | 0.337 | **0.362** |
| GSM8K | 0.612 | 0.671 | 0.667 | 0.684 | **0.703** |
| Math500 | 0.196 | **0.284** | 0.266 | 0.272 | **0.284** |

Table 3: Accuracy on GSM8K of post-trained LLaDA with diffu-GRPO by sampling strategy.

| Random (Zheng et al., 2025) | Margin (Kim et al., 2025) | Entropy (Ye et al., 2025) | Confidence (Chang et al., 2022) | Ours |
|---|---|---|---|---|
| 0.638 | 0.750 | 0.740 | 0.751 | **0.764** |

## 4.3 MAIN RESULT

**Baselines and setup.** We compare our learned unmasking policy with four widely used schedulers: random (Zheng et al., 2025), margin (Kim et al., 2025), entropy (Ye et al., 2025), and confidence (Chang et al., 2022). For SUDOKU and ZEBRA, we utilize max-confidence realization, and for GSM8K and MATH500, we use Top-$K$ realization where $K = 5$.

**Results.** Table 2 shows that our policy matches or surpasses all heuristics on every benchmark. For SUDOKU and ZEBRA, where we use $g_{conf}$ as the reference, our model exceeds this reference: SUDOKU improves from 70.5% to 81.7%, and ZEBRA from 33.7% to 36.2%. On GSM8K, trained with a Top-$K$ reference, our policy reaches 70.3% and exceeds the strong max-confidence baseline at 68.4%. On MATH500, also trained with a Top-$K$ reference, our policy attains 28.4%, tying the best baseline while improving over max-confidence at 27.2%. Overall, max-confidence remains a strong single-path heuristic, yet a learned policy trained in our regularized framework consistently surpasses it.

## 4.4 ABLATION STUDY

**Compatibility with diffu-GRPO.** Zhao et al. (2025) introduces diffu-GRPO, an RL method that post-trains the MDM itself via GRPO. Our approach also uses RL but targets a different component: we optimize the unmasking policy while keeping the base MDM fixed. The two methods are therefore complementary and can be combined. To demonstrate this, we first post-train LLADA-8B-INSTRUCT on GSM8K with diffu-GRPO, then train our policy model with Top-$K$ realization, $\mathcal{L}_{UPO}(g_{\phi_{Top-5}}, g_{Top-5}, \mathcal{L}_{KL})$. As reported in Table 3, our unmasking policy model yields additional gains on top of the diffu-GRPO model: +12.6% over random and +1.3% over max-confidence.

**Power of regularization.** We compare the mean and standard deviation of the group reward while optimizing $\mathcal{L}_{UPO}(g_\phi, g_{ref}, D)$ in two setups: for SUDOKU, we train a pretrained policy with/without CE to $g_{conf}$, and for GSM8K, we train a randomly initialized $g_{\phi_{Top-K}}$ with/without $\mathcal{L}_{KL}$ to $g_{Top-K}$. As shown in Figure 4, adding the divergence term yields higher final accuracy and, at similar mean rewards—even near 1—maintains a larger group-reward standard deviation. This helps the policy model avoid premature convergence under the GRPO framework and reach a higher convergence point. We conjecture that, for GSM8K, $\mathcal{L}_{KL}$ to $g_{Top-K}$ force the probability of the trajectory sampled from $g_{\phi_{Top-K}}$ to be all equal and prevent early path collapse. For SUDOKU, we suppose that CE toward $g_{conf}$ counteracts collapse onto a suboptimal order by pulling the policy back to high-confidence alternatives, keeping multiple competitive paths active and maintaining variance.

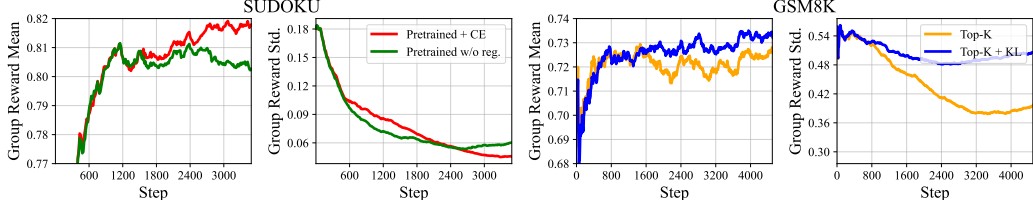

Figure 4: Mean and standard deviation of group reward during training under $\mathcal{L}_{\text{UPO}}$. For SUDOKU, we compare pretrained policy trained with/without CE to $g_{\text{conf}}$, and for GSM8K, we compare randomly initialized $g_{\phi_{\text{Top-K}}}$ trained with/without KL to $g_{\text{Top-K}}$.

## 5 CONCLUSION

We introduced a KL-regularized MDP formulation for learning the unmasking policy in masked diffusion models. Our analysis provides two guarantees: convergence to a fixed point that improves expected reward over the reference policy, and a KL tightening showing that the terminal-output distribution is closer to the real data than the reference. To make the framework practical, we derived a tractable surrogate loss and proposed several reference realizations. Across SUDOKU, ZEBRA, GSM8K, and MATH500, the learned unmasking policy model consistently matches or surpasses strong heuristics.

**Discussion.** Our learned unmasking policy improves performance across benchmarks, though the gains on GSM8K and MATH500 are relatively smaller than on SUDOKU. We conjecture that this stems from two differences between the domains. First, SUDOKU exhibits relatively clear and consistent ordering structure—for example, certain positions are statistically more constraining—whereas such regularity is much weaker in mathematical reasoning tasks. Second, while SUDOKU requires learning ordering patterns over a small digit space, GSM8K requires learning over the full vocabulary, despite having far fewer training examples. These factors likely make it more challenging to extract reliable ordering cues in GSM8K, resulting in more modest gains.

**Future directions.** Our work demonstrates that GRPO-based training can yield task-specific unmasking policies that outperform max-confidence and other rule-based heuristics. As discussed above, an important next step is to develop more generalizable policies that remain effective on larger and more diverse language datasets; we explore this direction further in Appendix D.2 with additional experiments. Another promising direction is to design more stabilized policy optimization methods. On SUDOKU, the full reparametrization $g_\phi$ outperforms $g_{\phi_{\text{Top-K}}}$, suggesting that near-optimal training benefits from occasionally selecting low-confidence positions. In contrast, the much larger search space in GSM8K favors the stability of the Top-K–restricted policy. Developing an optimization scheme that enables effective exploration of all masked positions, even in large search spaces, constitutes an important avenue for future work.

## ACKNOWLEDGMENTS

This work was supported by the National Research Foundation of Korea under Grant RS-2024-00336454, by the Institute of Information & Communications Technology Planning & Evaluation(IITP) grant funded by the Korea government(MSIT) (RS-2025-02304967, AI Star Fellowship(KAIST) & RS-2024-00457882, AI Research Hub Project), and by AI Computing Infrastructure Enhancement (GPU Rental Support) User Support Program funded by the Ministry of Science and ICT (MSIT), Republic of Korea (RQT-25-120217).

## ETHICS STATEMENT

We follow the ICLR Code of Ethics. Our work trains a lightweight *unmasking policy* while keeping the base MDM frozen, and evaluates only on public benchmarks (SUDOKU, ZEBRA, GSM8K, MATH500) that contain no personally identifiable information; no human-subject data were collected and no IRB approval was required. As our method optimizes only a lightweight scheduling policy to maximize a verifiable terminal reward and does not modify the base MDM's parameters or training

data, the safety and fairness properties are largely inherited from the underlying model. Absent defects in the base model, the incremental risk is limited to potential amplification of existing behaviors; in practice, this can be mitigated by pairing our scheduler with standard safety filters and auditing the combined system on downstream tasks.

## REPRODUCIBILITY STATEMENT

The training algorithm is detailed in Algorithm 2, the overall experimental setup is described in Section 4, and the hyperparameters and prompt templates used for evaluation are provided in Appendix E. Model architecture and the policy inference procedure are summarized in Appendix B. Complete proofs of all theorems and propositions stated in the main paper appear in Appendix C.

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

CONTENTS

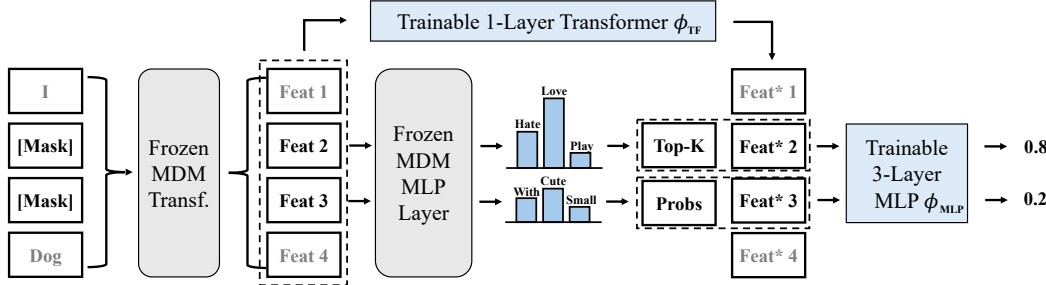

Figure 5: The structure of our unmasking policy model. The model is composed of a single Transformer layer and a 3-layer MLP. In inference time, the model proceeds as follows: 1) Given an input sentence, we run the base MDMs transformer and extract a feature. 2) This feature branches into two paths: first, the base MDMs continue to produce a prediction of token distribution, and second, fed to the policy model's transformer layer. 3) We concatenate the extracted feature with the base MDM's Top-K probabilities. 4) The concatenated feature is then processed by a 3-layer MLP to yield a policy over unmasking positions.

# A  RELATED WORKS

**Discrete Diffusion Models.** Diffusion probabilistic models have become the state-of-the-art approach for continuous data domains such as images, audio, and video, by iteratively denoising Gaussian-corrupted inputs (Ho et al., 2020; Song et al., 2021). Extending this success to *discrete* data (e.g. text or sequences) has led to the development of discrete diffusion models built on discrete-state Markov chains rather than continuous noise processes (Austin et al., 2021; Hoogeboom et al., 2021). Lou et al. (2024) proposed SEDD, which utilizes theoretical score entropy objective. Meanwhile, masked diffusion models (MDMs), which utilize an absorbing mask strategy, have emerged as a leading class of discrete diffusion models for text generation (Lou et al., 2024; Sahoo et al., 2024; Shi et al., 2024). By using an absorbing masking state, MDMs simplify the corruption process: once a token is masked, it remains masked until generation, analogous to absorbing diffusion in continuous models (Shi et al., 2024). This simplification yields a clean and principled training recipe aligned with continuous-time diffusion theory, including an evidence lower bound (ELBO) objective (Shi et al., 2024). Beyond continuous-time discrete diffusion, Zheng et al. (2025); Ou et al. (2025) provides a time-agnostic training framework and simple cross-entropy loss. These advances have significantly closed the performance gap between MDMs and traditional autoregressive models on language tasks (Ye et al., 2025; Nie et al., 2025) when the scale of the model is similar.

**Unmasking Strategy in Masked Diffusion Inference.** A crucial factor for MDM performance is the *unmasking strategy* used during inference—i.e., how the model decides which masked tokens to reveal at each generation step. While MDMs have shown superior performance on text generations, earlier works (Ou et al., 2025; Shih et al., 2022; Li et al., 2021) have empirically observed that a random unmasking strategy is suboptimal. Moreover, Kim et al. (2025) proves that no polynomial-time algorithm can solve any-order generation; *i.e.*, we cannot train MDMs that exactly recovers the real data distribution for all masked sentences. However, they empirically show that selecting unmasking tokens via max-confidence (Chang et al., 2022) or max-margin (Kim et al., 2025) at inference time can bypass sub-hard instances and even outperform ARMs. As a result, contemporary large-scale MDMs (Nie et al., 2025; Ye et al., 2025) commonly adopt max-confidence scheduling at inference.

**Remark. Detailed Explanation of Hardness of MDM Training.** Kim et al. (2025) show that the conditional prediction tasks induced by *Any-order* masked diffusion models fall into the planted constrained satisfaction problem (CSP) hard phase (Krzakala & Zdeborová, 2009), implying that no polynomial-time algorithm can achieve optimal performance in general. Consequently, MDM training loss, Eq. (1) (the reverse-process KL objective driven by Sahoo et al. (2024); Zheng et al. (2025)) cannot be driven to zero, which is consistent with their empirical finding that loss remains high on text and L&O-NAE-SAT datasets.

## B  TRAINING RECIPE IN DETAIL

In this section, we explain our memory-efficient training recipe in detail. We provide full derivations of various reference policy realizations (Appendix B.1), and algorithms for training and generating (Appendix B.2). For a deeper understanding of our memory-efficient training algorithm and model architecture, we first provide notations and model inference in detail.

**Policy model inference process.**  Our rationale behind policy model structure is two: 1) utilizing the MDM's knowledge (extracted feature), which allows for avoiding from-scratch training and allow memory-efficient training, and 2) utilizing the MDM's token prediction distribution, which can help the model mimic the behavior of $g_{\text{conf}}$. The main paper outlined the policy model's inference at a high level; here, we provide a notation and description of how, given a masked sentence $\boldsymbol{x}_n$, the model generates an unmasking trajectory. For reference, we reproduce the model architecture from the main paper (Figure 2) as Figure 5. To recap, the unmasking policy model is parameterized as a softmax over masked positions using a learnable scorer $h_\phi(\boldsymbol{x}) \in \mathbb{R}^L$:

$$g_\phi(a^i \,|\, \boldsymbol{x}_n) = \frac{\exp\big([h_\phi(\boldsymbol{x}_n)]_{a^i}\big)}{\sum_{i'=1}^n \exp\big([h_\phi(\boldsymbol{x}_n)]_{a^{i'}}\big)}, \tag{20}$$

where $a^i$ is $i$-th mask of $\boldsymbol{x}_n$. As illustrated in Figure 5, the policy model decomposes as $\phi = (\phi_{\text{MLP}}, \phi_{\text{TF}})$, where $\phi_{\text{TF}}$ is a 1-layer Transformer and $\phi_{\text{MLP}}$ is a 3-layer MLP. We similarly write the MDM as $\theta = (\theta_{\text{TF}}, \theta_{\text{MLP}})$ with a feature extractor transformer $\theta_{\text{TF}}$ and a token prediction head $\theta_{\text{MLP}}$. Given the current masked sequence $\boldsymbol{x}_n$ and selected mask $a^i$, the MDM computes the token distribution as follows:

$$\mathbf{H}_n = \theta_{\text{TF}}(\boldsymbol{x}_n) \in \mathbb{R}^{L \times d}, \qquad \mathbf{u}_{a^i} = \mathbf{H}_n[a^i] \in \mathbb{R}^d,$$

$$\mathbf{z}_{a^i} = \theta_{\text{MLP}}(\mathbf{u}_{a^i}) \in \mathbb{R}^{|\mathcal{V}|}, \qquad \pi_\theta\Big(x_{n-1}^{a^i} = c \,\Big|\, \boldsymbol{x}_n\Big) = \text{softmax}(\mathbf{z}_{a^i})_c \quad (c \in \mathcal{V}),$$

where $d$ is the output dimension of the transformer feature. Let $\mathbf{p}_{a^i} = \text{softmax}(\mathbf{z}_{a^i})$ the token posterior at index $a^i$ with dimension $|\mathcal{V}|$. The policy encoder first refines the sequence features,

$$\widetilde{\mathbf{H}}_n = \phi_{\text{TF}}(\mathbf{H}_n) \in \mathbb{R}^{L \times d}, \qquad \widetilde{\mathbf{u}}_{a^i} = \widetilde{\mathbf{H}}_n[a^i] \in \mathbb{R}^d.$$

Let $\text{TopK}(\mathbf{p}_{a^i}) \in \mathbb{R}^K$ denote the vector of the $K$ largest probabilities of $\mathbf{p}_{a^i}$, sorted in descending order. We form the concatenated feature

$$\mathbf{s}_{a^i} = \big[\widetilde{\mathbf{u}}_{a^i} \,;\, \text{TopK}(\mathbf{p}_{a^i})\big] \in \mathbb{R}^{d+K}, \tag{21}$$

and map it to a scalar score via the 3-layer MLP:

$$[h_\phi(\boldsymbol{x}_n)]_{a^i} = \phi_{\text{MLP}}(\mathbf{s}_{a^i}) \in \mathbb{R}.$$

Finally, the unmasking policy over the masked index set is the softmax of these scores (Eq. (20)).

### B.1  FULL DERIVATIONS FOR $\mathcal{L}_{\text{UPO}}$ REALIZATIONS

**General formulations.**  To recap, our final objective is given as follows:

$$\mathcal{L}_{\text{UPO}}(g_\phi, g_{\text{ref}}, D) = \mathbb{E}_{\mathbf{q} \sim \rho_Q, \{\boldsymbol{x}_{0:L}^{(g)}, a_{1:L}^{(g)}\}_{g=1}^G \sim p_{g_{\phi_{\text{old}}}}} \left[ \frac{1}{G} \sum_g^G \left( \Big( \frac{1}{L} \sum_{n=1}^L \min\big( \frac{g_\phi(a_n^{(g)}|\boldsymbol{x}_n^{(g)})}{g_{\phi_{\text{old}}}(a_n^{(g)}|\boldsymbol{x}_n^{(g)})} A_g, \right. \right.$$

$$\left. \left. \text{clip}\big( \frac{g_\phi(a_n^{(g)}|\boldsymbol{x}_n^{(g)})}{g_{\phi_{\text{old}}}(a_n^{(g)}|\boldsymbol{x}_n^{(g)})}, 1-\epsilon, 1+\epsilon \big) A_g \big) \Big) - \beta \cdot D(p_{g_\phi} \| p_{g_{\text{ref}}}) \right) \right], \tag{22}$$

The divergence term $D$ in Eq. (22) is a *functional* that, given a prompt $\mathbf{q}$, a full trajectory $\tau = (\boldsymbol{x}_{0:L}, a_{1:L})$, and models $(g_\phi, g_{\text{ref}}, \pi_\theta)$, returns a scalar estimate. Formally,

$$D_{g_\phi, g_{\text{ref}}, \pi_\theta} : \mathcal{Q} \times \mathcal{X}_{0:L} \times \mathcal{A}_{1:L} \to \mathbb{R},$$

and its empirical expectation serves as a tractable estimator of the *output-level* KL divergence $D_{\mathrm{KL}}\big(p_{g_\phi}(\boldsymbol{x}_0 \mid \mathbf{q}) \,\|\, p_{g_{\mathrm{ref}}}(\boldsymbol{x}_0 \mid \mathbf{q})\big)$. Following Proposition 2 and Proposition 3, we can decrease the *output-level* KL divergence with $\mathcal{L}_{\mathrm{KL}}$ defined as follows:

$$\mathcal{L}_{\mathrm{KL}}(\boldsymbol{x}_{0:L}, a_{1:L}, \mathbf{q}) = \mathrm{StopGrad}\Big(\frac{p_{g_\phi}(\boldsymbol{x}_{0:L}|\mathbf{q})}{p_{g_{\phi_{\mathrm{old}}}}(\boldsymbol{x}_{0:L}|\mathbf{q})} \cdot \big(1 + \log \frac{p_{g_\phi}(\boldsymbol{x}_{0:L}|\mathbf{q})}{p_{g_{\mathrm{ref}}}(\boldsymbol{x}_{0:L}|\mathbf{q})}\big)\Big) \cdot \sum_{n=1}^{L} \log g_\phi(a_n \mid \boldsymbol{x}_n),$$
(23)

where $g_\phi$, $g_{\mathrm{ref}}$ and $\pi_\theta$ is omitted in $\mathcal{L}_{\mathrm{KL}}$ for brevity. We can convert $p_g(\boldsymbol{x}_{0:L}|\mathbf{q})$ into $\prod_{n=1}^{L} g(a_n|\boldsymbol{x}_n) \cdot \pi_\theta(\boldsymbol{x}_{n-1}|\boldsymbol{x}_n, a_n, \mathbf{q})$ in Eq. (23):

$$\mathcal{L}_{\mathrm{KL}}(\cdot) = \mathrm{StopGrad}\Big(\prod_{n=1}^{L} \frac{g_\phi(a_n|\boldsymbol{x}_n)}{g_{\phi_{\mathrm{old}}}(a_n|\boldsymbol{x}_n)} \cdot \big(1 + \sum \log \frac{g_\phi(a_n|\boldsymbol{x}_n)}{g_{\mathrm{ref}}(a_n|\boldsymbol{x}_n)}\big)\Big) \cdot \sum_{n=1}^{L} \log g_\phi(a_n \mid \boldsymbol{x}_n).$$
(24)

We already computed $g_{\phi_{\mathrm{old}}}(\cdot)$ while sampling, and $g_\phi(\cdot)$ while computing reward-maximization objective term in $\mathcal{L}_{\mathrm{UPO}}$, and $g_{\mathrm{ref}}$ does not require network evaluations. Therefore, we can measure $\mathcal{L}_{\mathrm{KL}}$ without no additional computation. Furthermore, since $\sum_{n=1}^{L} \log g_\phi(a_n \mid \boldsymbol{x}_n)$ is summation, we can perform gradient update token-by-token, which allows tractable and memory-efficient training.

**Softmax realization.** For softmax realization of $g_{\mathrm{ref}}$, we consider $g_{\mathrm{conf}^\tau}$ given as follows:

$$g_{\mathrm{conf}^\tau}(a^i|\boldsymbol{x}_n) = \frac{\sum_{c\in\mathcal{V}} \exp(\pi_\theta(x_{n-1}^{a^i} = c|\boldsymbol{x}_n)/\tau)}{\sum_{a^j} \sum_{c\in\mathcal{V}} \exp(\pi_\theta(x_{n-1}^{a^j} = c|\boldsymbol{x}_n)/\tau)}, \quad \mathrm{s.t.} \lim_{\tau\to 0+} g_{\mathrm{conf}^\tau} = g_{\mathrm{conf}}. \quad (25)$$

Here, since $g_{\mathrm{conf}^\tau}$ assigns non-zero probability to every unmasking index, we can estimate the terminal KL divergence with a sampled trajectory. We utilize $\mathcal{L}_{\mathrm{KL}}$ (Eq. (24)) by replacing $g_{\mathrm{ref}}$ with $g_{\mathrm{conf}^\tau}$.

**Top-$K$ realization** For Top-$K$ realization of $g_{\mathrm{ref}}$, we consider $g_{\mathrm{conf}^\tau}$ given as follows:

$$g_{\mathrm{Top-K}}(a^i|\boldsymbol{x}_n) = \frac{1}{K} \cdot \mathbf{1}(a^i \in \underset{a^j}{\mathrm{argtopk}}(\max_{c\in\mathcal{V}} \pi_\theta(x_{n-1}^{a^j} = c|\boldsymbol{x}_n))), \quad \mathrm{s.t.} \ g_{\mathrm{Top-1}} = g_{\mathrm{conf}}. \quad (26)$$

where we reparametrize $g_{\phi_{\mathrm{Top-K}}}$ defined as follows:

$$g_{\phi_{\mathrm{Top-K}}}(a^i|\boldsymbol{x}_n) = \frac{\exp([h_\phi(\boldsymbol{x}_n)]_{a^i})}{\sum_{a^j \in \mathrm{argtopk} \max \pi_\theta(\cdot)} \exp([h_\phi(\boldsymbol{x}_n)]_{a^j})}, \quad (27)$$

which is the softmax distribution of $h_\phi$ on the Top-$K$ set of unmasking positions. This reparametrization offers two advantages: 1) when $\phi$ is randomly initialized, we can expect $g_{\phi_{\mathrm{Top-K}}} = g_{\mathrm{Top-K}}$, and 2) for sampled trajectory, $g_{\mathrm{Top-K}}(a_n|\boldsymbol{x}_n)$ is non-zero for all $n \in [1, L]$ since we sample from $g_{\phi_{\mathrm{Top-K}}}$. Therefore, we can train $g_{\phi_{\mathrm{Top-K}}}$ from scratch with $\mathcal{L}_{\mathrm{KL}}$ by replacing $g_{\mathrm{ref}} = g_{\mathrm{Top-K}}$.

**Max-confidence realization.** For max-confidence realization of $g_{\mathrm{ref}}$, we consider $g_{\mathrm{conf}}$ given as follows:

$$g_{\mathrm{conf}}(a^i|\boldsymbol{x}_n) = \mathbf{1}\big(a^i = \arg\max_{a^j}(\max_{c\in\mathcal{V}} \pi_\theta(x_{n-1}^{a^j} = c|\boldsymbol{x}_n))\big), \quad (28)$$

where $g_{\mathrm{conf}}$ is deterministic. The terminal KL divergence $D_{\mathrm{KL}}\big(p_{g_\phi}(\boldsymbol{x}_0 \mid \mathbf{q}) \,\|\, p_{g_{\mathrm{ref}}}(\boldsymbol{x}_0 \mid \mathbf{q})\big)$ can be defined where $\pi_\theta$ is stochastic, but we cannot empirically estimate the terminal KL divergence with sampled trajectory since there exists trajectory sampled from $p_{g_\phi}$ is 0 for $p_{g_{\mathrm{conf}}}$. Therefore, we just utilize cross-entropy given as follows:

$$\mathrm{CE}_{g_\phi, g_{\mathrm{conf}}, \pi_\theta}(\boldsymbol{x}_{0:L}, \mathbf{q}) = \sum_{n=1}^{L} \sum_{a\in\mathcal{A}_{\boldsymbol{x}_n}} g_{\mathrm{conf}}(a|\boldsymbol{x}_n) \log g_\theta(a|\boldsymbol{x}_n). \quad (29)$$

Although not theoretical, we can expect to regularize $g_\phi$ being far away from $g_{\mathrm{conf}}$, and we have empirically confirmed the performance gain (Section 4.4).

---

**Algorithm 1** Generalized Sampling of MDMs

---

**Require:** Sequence length $L$, vocabulary $\mathcal{V} = \{1, \ldots, m\}$, pretrained masked diffusion model $\pi_\theta$, unmasking policy $g$

1:  $\tau_L \leftarrow 1$
2:  **for** $n \leftarrow L$ **to** $1$ **do**
3:      sample $a_n \sim g(a|\boldsymbol{x}_n)$                $\triangleright$ unmasking token distribution. *e.g.*, $\mathcal{U}(\{a^i\}_{i=1}^n)$
4:      sample $x_{n-1}^{a_n} \sim \pi_\theta(x_{n-1}^{a_n}|\boldsymbol{x}_n)$            $\triangleright$ per-position categorical probabilities
5:      $\boldsymbol{x}_{n-1} \leftarrow \{x_n^1, x_n^2, \ldots, x_n^{a_n-1}, x_{n-1}^{a_n}, x_n^{a_n+1}, \ldots, x_n^L\}$
6:  **end for**
7:  **Output:** $\boldsymbol{x}_0$

---

## B.2 ALGORITHMS

**Sampling algorithm.** We provide generalized sampling algorithm of large-scale MDMs in Algorithm 1. The sampling trajectory differs by unmasking policy $g$, and we can perform various sampling by setting $g$ as $g_{\text{rand}}, g_{\text{conf}}, g_{\text{conf}^\tau}, g_{\text{Top}-\text{K}}$, or $g_\phi$.

**Unmasking policy training algorithm.** We provide a memory-efficient unmasking policy model training algorithm in Algorithm 2. We have previously mentioned how we utilize the extracted features and token distribution of MDM as input to the unmasking policy model. We also have provided how the gradient of $\mathcal{L}_{\text{UPO}}$ can be measured by the token-wise gradient of $g_\phi(\cdot|\boldsymbol{x})$. Summing up all techniques, we can perform training without using GPU memory more than the amount of original MDM occupies. Indeed, we have confirmed that training can be performed with only one A100 40GB GPU.

Specifically, the algorithm is divided into two phases: the sampling phase and the gradient update phase. At the sampling phase, we sample various sequences from $p_{g_{\phi_{\text{old}}}}(\cdot|\mathbf{q})$. In this phase, we gather the extracted feature of the MDM transformer, the token distribution of MDM, the mask token index, and record the probability of the trajectory. At the gradient update phase, we can perform mini-batch training where all the gradients of $g_\phi(\cdot|\boldsymbol{x}_n)$ in $\mathcal{L}_{\text{UPO}}$ are independent of others. Furthermore, since we have gathered all the information of $\pi_\theta$ that is needed to update the gradient in the sampling phase, we can free the memory of MDM $\theta$. The only parameters requiring forward and backward process in the gradient update phase is the unmasking policy model, which is much smaller than MDM (*e.g.*, LLaDA: 8B, corresponding policy model: 134M). Therefore, we can perform memory-efficient training. For more details, refer to the algorithm table.

---

**Algorithm 2** Memory-efficient training of the unmasking policy model

---

**Require:** Pretrained masked diffusion model $\pi_\theta$ composed of feature extracting transformer $\theta_{\text{TF}}$ and token estimator $\theta_{\text{MLP}}$, unmasking policy model $g_\phi$ composed of 1-layer transformer $\phi_{\text{TF}}$ and 3-Layer MLP $\phi_{\text{MLP}}$, divergence loss $D$, reference policy $g_{\text{ref}}$, prompt distribution $\mathcal{Q}$, number of inner updates $\mu$

1: **while** not converge **do**
    **Sampling phase**
2:    **with** `torch.no_grad()`:
3:        $g_{\phi_{\text{old}}} \leftarrow g_\phi$
4:        Sample a prompt $\mathbf{q} \sim \mathcal{Q}$
5:        Sample G trajectories $\{\boldsymbol{x}_{0:L}^{(g)}, a_{1:L}^{(g)}\}_{g=1}^G \sim p_{g_{\phi_{\text{old}}}}(\cdot|\mathbf{q}), g \in [G]$
6:            $\triangleright$ while sampling, save $\mathbf{H}_n^{(g)} = \theta_{\text{TF}}(\boldsymbol{x}_n^{(g)}), \mathcal{A}_{\boldsymbol{x}_n^{(g)}}, \pi_\theta(\cdot|\boldsymbol{x}_n^{(g)}), g_{\phi_{old}}(a_n^{(g)}|\boldsymbol{x}_n^{(g)})$
7:            $\triangleright$ if $D = \mathcal{L}_{\text{KL}}$, record $p_{g_{\phi_{\text{old}}}}(\boldsymbol{x}_{0:L}^{(g)}, a_{1:L}^{(g)}|\mathbf{q}), p_{g_{\phi_{\text{ref}}}}(\boldsymbol{x}_{0:L}^{(g)}, a_{1:L}^{(g)}|\mathbf{q})$ while sampling
8:        For each $\boldsymbol{x}_0^{(g)}$, compute reward $r_g$ and advantage $A_g$
9:    **end with**
    **Gradient update phase**
10:    **for** gradient update iterations $n = 1, \ldots, \mu$ **do**
11:        **with** `torch.no_grad()`:
12:            For each $\boldsymbol{x}_{0:L}^{(g)}$, pre-compute $p_{g_\phi}(\boldsymbol{x}_{0:L}^{(g)}|\mathbf{q})$
13:            **if** $D = \mathcal{L}_{\text{KL}}$ **then**
14:                $\kappa_g \leftarrow \left( \frac{p_{g_\phi}(\boldsymbol{x}_{0:L})}{p_{g_{\phi_{\text{old}}}}(\boldsymbol{x}_{0:L})} \cdot \left(1 + \log \frac{p_{g_\phi}(\boldsymbol{x}_{0:L})}{p_{g_{\text{ref}}}(\boldsymbol{x}_{0:L})}\right) \right)$
15:            **end if**
16:        **end with**
17:        Make mini-batch $\{B\}_{i=1}^b$ such that $B_1 \cup B_2 \cup \cdots \cup B_b = \{1, \ldots, L\}$ and $B_i \cap B_j = \emptyset$ for all $i \neq j$
18:        **for** $B \in \{B_i\}_{i=1}^b$ **do**
19:            For each $n \in B$ and $g \in \{1, .., G\}$, $\widetilde{\mathbf{H}}_n^{(g)} \leftarrow \phi_{\text{TF}}(\mathbf{H}_n^{(g)})$
20:            For each $n \in B$ and $g \in \{1, .., G\}$, construct $\mathbf{s}_n^{(g)}$ using Eq. (21)
21:            $g_\phi(\cdot|\boldsymbol{x}_n^{(g)}) \leftarrow \phi_{\text{MLP}}(\mathbf{s}_n^{(g)})$
22:            $\mathcal{L}_\phi \leftarrow \frac{1}{|B|} \sum_{n \in B, g \in G} \min\left( \frac{g_\phi(a_n^{(g)}|\boldsymbol{x}_n^{(g)})}{g_{\phi_{\text{old}}}(a_n^{(g)}|\boldsymbol{x}_n^{(g)})} A_g, \text{clip}\left(\frac{g_\phi(a_n^{(g)}|\boldsymbol{x}_n^{(g)})}{g_{\phi_{\text{old}}}(a_n^{(g)}|\boldsymbol{x}_n^{(g)})}, 1 - \epsilon, 1 + \epsilon\right) A_g \right)$
23:            **if** $D = \text{CE}$ **then**
24:                $\mathcal{L}_\phi \leftarrow \mathcal{L}_\phi - \beta \cdot \sum_{g \in G} \sum_{n \in B} \text{CE}(g_{\text{conf}}(\cdot|\boldsymbol{x}_n^{(g)}) || g_\phi(\cdot|\boldsymbol{x}_n^{(g)}))$
25:            **end if**
26:            **if** $D = \mathcal{L}_{\text{KL}}$ **then**
27:                $\mathcal{L}_\phi \leftarrow \mathcal{L}_\phi - \beta \cdot \sum_{g \in G} \kappa_g \sum_{n \in B} \log g_\phi(a_n^{(g)} \mid \boldsymbol{x}_n^{(g)})$
28:            **end if**
29:            Update $\phi$ by performing gradient ascent on $\mathcal{L}_{\text{UPO}}$
30:        **end for**
31:    **end for**
32: **end while**
33: **Output:** $g_\phi$

---

## C DETAILED PROOF

In this section, we provide full proofs omitted in the main paper.

### C.1 PROOF OF KL TIGHTENING FOR MDM POLICY

**Theorem 2** (Reference-KL Tightening for MDM Policy Improvement). *Assume there exists an ideal MDM $\theta^*$ satisfying MDM $\mathcal{L}_{MDM}(\theta^*) = 0$ for $p_{data}$ where data is composed of prompt and oracle answer $\{\mathbf{q}, \mathbf{a}\}$. Assume we select well-defined reward $r(\mathbf{q}, \mathbf{a}) \in \{0, 1\}$ such that $p_{data}(\mathbf{q}, \mathbf{a}) > 0 \Rightarrow r(\mathbf{q}, \mathbf{a}) = 1$ and $0 < r_{g_{ref}} < 1$. Now consider incomplete MDM $\theta$ such that $\mathcal{L}_{MDM}(\theta) > 0$, and $g_{\phi^*}$ that satisfies Eq. (13) for $\theta$. Let $g_{\mathrm{ref}}$ be a policy model such that, for every $\mathbf{q}$, $p_{g_{\mathrm{ref}}}(\boldsymbol{x}_0 \mid \mathbf{q}) > 0$ whenever $p_{\theta^*}(\boldsymbol{x}_0 \mid \mathbf{q}) > 0$, i.e., $\mathrm{supp}(p_{\theta^*}(\cdot \mid \mathbf{q})) \subseteq \mathrm{supp}(p_{g_{\mathrm{ref}}}(\cdot \mid \mathbf{q}))$. Then, the following inequality holds:*

$$D_{\mathrm{KL}}(p_{\theta^*}(\boldsymbol{x}_0|\mathbf{q})||p_{g_{\phi^*}}(\boldsymbol{x}_0|\mathbf{q})) < D_{\mathrm{KL}}(p_{\theta^*}(\boldsymbol{x}_0|\mathbf{q})||p_{g_{\mathrm{ref}}}(\boldsymbol{x}_0|\mathbf{q})). \tag{14}$$

*Or equivalently,*

$$D_{\mathrm{KL}}(p_{\mathrm{data}}(\mathbf{a}|\mathbf{q})||p_{g_{\phi^*}}(\boldsymbol{x}_0|\mathbf{q})) < D_{\mathrm{KL}}(p_{\mathrm{data}}(\mathbf{a}|\mathbf{q})||p_{g_{\mathrm{ref}}}(\boldsymbol{x}_0|\mathbf{q})). \tag{15}$$

*Proof.* We first recall the GRPO policy training dynamics learning Eq. (10) proven by Mroueh (2025):

**Lemma 1** (GRPO Policy Dynamic (Mroueh, 2025)). *Optimal GRPO iterations policies solving Eq. (10) satisfy the following recursion, for $n \geq 1$:*

$$p_{g_{\phi_n}}(\boldsymbol{x}_0 \mid \mathbf{q}) = \frac{1}{Z_{n-1}(\mathbf{q})} \, p_{g_{\mathrm{ref}}}(\boldsymbol{x}_0 \mid \mathbf{q}) \tag{30}$$

$$\times \exp\left(\frac{1}{\beta}\left[w_\epsilon^+\big(r_{g_{\phi_{n-1}}}(\mathbf{q})\big)\, r(\mathbf{q}, \boldsymbol{x}_0) - w_\epsilon^-\big(r_{g_{\phi_{n-1}}}(\mathbf{q})\big)(1 - r(\mathbf{q}, \boldsymbol{x}_0))\right]\right). \tag{31}$$

*where*

$$Z_{n-1}(\mathbf{q}) = r_{g_{\mathrm{ref}}}(\mathbf{q}) \exp\left(\frac{1}{\beta} w_\epsilon^+\big(r_{g_{\phi_{n-1}}}(\mathbf{q})\big)\right) + \big(1 - r_{g_{\mathrm{ref}}}(\mathbf{q})\big) \exp\left(-\frac{1}{\beta} w_\epsilon^-\big(r_{g_{\phi_{n-1}}}(\mathbf{q})\big)\right). \tag{32}$$

*and $w_\epsilon^+(r)$ and $w_\epsilon^-(r)$ are defined as follows:*

$$w_\epsilon^+(r) = \frac{1-r}{r(1-r) + \epsilon}, \quad w_\epsilon^-(r) = \frac{r}{r(1-r) + \epsilon} \tag{33}$$

Under Theorem 1, at the fixed point, we can substitute the converged policy into the recursion and compare the two KL terms. The next steps simply expand the KL difference. Then, for the convergence point where $\lim_{n\to\infty} r_{g_{\phi_n}} = r_{g^*}$:

$$D_{\mathrm{KL}}(p_{\theta^*}(\boldsymbol{x}_0|\mathbf{q}) \,\|\, p_{g_{\mathrm{ref}}}(\boldsymbol{x}_0|\mathbf{q})) - D_{\mathrm{KL}}(p_{\theta^*}(\boldsymbol{x}_0|\mathbf{q}) \,\|\, p_{g_{\phi^*}}(\boldsymbol{x}_0|\mathbf{q})) \tag{34}$$

$$= \mathbb{E}_{p_{\theta^*}}\left[\log p_{\theta^*}(\boldsymbol{x}_0|\mathbf{q}) - \log p_{g_{\mathrm{ref}}}(\boldsymbol{x}_0|\mathbf{q})\right] - \mathbb{E}_{p_{\theta^*}}\left[\log p_{\theta^*}(\boldsymbol{x}_0|\mathbf{q}) - \log p_{g_{\phi^*}}(\boldsymbol{x}_0|\mathbf{q})\right] \tag{35}$$

$$= \mathbb{E}_{p_{\theta^*}}\left[\log p_{g_{\phi^*}}(\boldsymbol{x}_0|\mathbf{q}) - \log p_{g_{\mathrm{ref}}}(\boldsymbol{x}_0|\mathbf{q})\right] \tag{36}$$

$$= \mathbb{E}_{p_{\theta^*}}\left[\frac{1}{\beta}\left(w_\epsilon^+\big(r_{g_{\phi^*}}(\mathbf{q})\big)\,r(\mathbf{q},\boldsymbol{x}_0) - w_\epsilon^-\big(r_{g_{\phi^*}}(\mathbf{q})\big)\big(1 - r(\mathbf{q},\boldsymbol{x}_0)\big)\right) - \log Z_{\phi^*}\right] \tag{37}$$

$$= \frac{1}{\beta}\,w_\epsilon^+\big(r_{g_{\phi^*}}(\mathbf{q})\big) - \log Z_{\phi^*} \qquad \because \forall \boldsymbol{x}_0 \in \mathcal{X},\ p_{\theta^*}(\boldsymbol{x}_0|\mathbf{q}) > 0 \Rightarrow r(\mathbf{q},\boldsymbol{x}_0) = 1 \tag{38}$$

$$= \frac{1}{\beta}\,w_\epsilon^+\big(r_{g_{\phi^*}}(\mathbf{q})\big)$$

$$\quad - \log\left(r_{g_{\mathrm{ref}}}(\mathbf{q})\,\exp\!\Big(\frac{1}{\beta}\,w_\epsilon^+\big(r_{g_{\phi^*}}(\mathbf{q})\big)\Big) + \big(1 - r_{g_{\mathrm{ref}}}(\mathbf{q})\big)\,\exp\!\Big(-\frac{1}{\beta}\,w_\epsilon^-\big(r_{g_{\phi^*}}(\mathbf{q})\big)\Big)\right) \tag{39}$$

$$> \frac{1}{\beta}\,w_\epsilon^+\big(r_{g_{\phi^*}}(\mathbf{q})\big)$$

$$\quad - \log\left(r_{g_{\mathrm{ref}}}(\mathbf{q})\,\exp\!\Big(\frac{1}{\beta}\,w_\epsilon^+\big(r_{g_{\phi^*}}(\mathbf{q})\big)\Big) + \big(1 - r_{g_{\mathrm{ref}}}(\mathbf{q})\big)\,\exp\!\Big(\frac{1}{\beta}\,w_\epsilon^+\big(r_{g_{\phi^*}}(\mathbf{q})\big)\Big)\right) \tag{40}$$

$$\because\ 1 \geq r_{g_{\phi^*}}(\mathbf{q}) > r_{g_{\mathrm{ref}}}(\mathbf{q}) > 0,\ \text{s.t.}\ w_\epsilon^+\big(r_{g_{\phi^*}}(\mathbf{q})\big) \geq 0,\ w_\epsilon^-\big(r_{g_{\phi^*}}(\mathbf{q})\big) > 0$$

$$= 0, \tag{41}$$

where $\mathrm{supp}(p_{\theta^*}(\cdot \mid \mathbf{q})) \subseteq \mathrm{supp}(p_{g_{\mathrm{ref}}}(\cdot \mid \mathbf{q}))$. Therefore, the inequality in Eq. (14) holds. Accordingly, inequality in Eq. (15) holds where $p_{\theta^*} = p_{\mathrm{data}}$ for ideal MDM $\theta^*$. $\qquad\square$

**Remark. Why does the support assumption** $\mathrm{supp}(p_{\theta^*}(\cdot \mid q)) \subseteq \mathrm{supp}(p_{g_{\mathrm{ref}}}(\cdot \mid q))$ **hold?**
Even if $g_{\mathrm{ref}}$ is deterministic (e.g., max-confidence), the support assumption naturally holds for masked diffusion models. The reason is that the MDM denoiser $\pi_\theta(\cdot \mid x_n)$ assigns strictly positive probability to *every* token in the vocabulary at each unmasking step. Therefore, for any terminal sequence $\mathbf{x}_0 \in \mathcal{V}^L$, there always exists at least one valid denoising trajectory under $g_{\mathrm{ref}}$ that reaches $\mathbf{x}_0$, implying $p_{g_{\mathrm{ref}}}(\mathbf{x}_0 \mid q) > 0$. Formally, the terminal-output likelihood marginalizes over all trajectories:

$$p_{g_{\mathrm{ref}}}(\mathbf{x}_0 \mid q) = \sum_{\mathbf{x}_{1:L},\, a_{1:L}} \prod_{n=1}^{L} g_{\mathrm{ref}}(a_n \mid \mathbf{x}_n, q)\,\pi_\theta(\mathbf{x}_{n-1} \mid \mathbf{x}_n, a_n, q).$$

Because $\pi_\theta$ has full support, the product above is strictly positive for at least one trajectory. To illustrate, take any target sequence $\mathbf{x}_0^*$. Starting from the fully masked state, $g_{\mathrm{ref}}$ selects an index $a_L$; since $\pi_\theta(x^{a_L} = x_0^{*,a_L} \mid \mathbf{x}_L) > 0$, the next state $\mathbf{x}_{L-1}$ satisfying $x_{L-1}^{a_L} = x_0^{*,a_L}$ is reachable. Repeating this inductively shows that $\mathbf{x}_0^*$ is reachable regardless of $g_{\mathrm{ref}}$. Thus the support assumption holds.

### C.2 Surrogate loss of output-level reward maximization

Our goal here is to show that the output-level reward maximization term in Eq. (10) can be converted into a tractable token-wise reward maximization term. The conclusion of Proposition 1 aligns with conventional RL that have developed with rigorous proofs, yet in our setting, where training the MDM unmasking policy, the assumptions are slightly different. More specifically, our MDP is a finite-horizon MDP with fixed length $L$ (text length is determined), sparse reward (only a verifiable reward for the terminal state exists), and non-stationary (for different time steps, states do not overlap). To fill this gap, we rigorously show that Proposition 1 holds.

We cannot even directly apply the policy gradient theorem (Sutton et al., 1999), which is essential in the proof. Therefore, we start with justifying how the policy gradient theorem also works in our framework, which can be easily proven:

**Lemma 2** (Policy gradient theorem in sparse-reward, finite-horizon, episodic, and non-stationary MDP). *Fix a horizon $L$. For each $i \in \{0, 1, \ldots, L\}$ let the state space be $\mathcal{X}_n$, and assume the spaces are pairwise disjoint: $\mathcal{X}_n \cap \mathcal{X}_j = \emptyset$ for $i \neq j$. At step $i \in \{1, \ldots, L\}$, from state $\boldsymbol{x}_n \in \mathcal{X}_n$ a discrete action $a_n \in \mathcal{A}_n$ is drawn from a stochastic policy $g_\phi(a_n \mid \boldsymbol{x}_n)$, and the environment transitions to the next layer's state $\boldsymbol{x}_{n-1} \in \mathcal{X}_{n-1}$ according to dynamics $\pi(\boldsymbol{x}_{n-1} \mid \boldsymbol{x}_n, a_n)$. Assume the start state is fixed as $\boldsymbol{x}_L$. The trajectory is $\tau = (\boldsymbol{x}_{0:L}, a_{1:L})$ with joint density*

$$p_{g_\phi}(\tau \mid \boldsymbol{x}_L) = \prod_{n=1}^{L} g_\phi(a_n \mid \boldsymbol{x}_n)\, \pi(\boldsymbol{x}_{n-1} \mid \boldsymbol{x}_n, a_n).$$

*Rewards are* terminal-only*: $r(\boldsymbol{x}_0)$ at $i = 0$ and $0$ otherwise. The objective is the expected terminal reward*

$$J(\phi) \;=\; \mathbb{E}_{\tau \sim p_{g_\phi}(\cdot \mid \boldsymbol{x}_L)}\big[r(\boldsymbol{x}_0)\big].$$

*Define the value function and state-action value function under $g_\phi$ by*

$$V_{g_\phi}(\boldsymbol{x}_n) \;=\; \mathbb{E}[r(\boldsymbol{x}_0) \mid \boldsymbol{x}_n], \qquad Q_{g_\phi}(\boldsymbol{x}_n, a_n) \;=\; \mathbb{E}[r(\boldsymbol{x}_0) \mid \boldsymbol{x}_n, a_n].$$

*Under the above assumptions, the policy gradient depends only on the policy $g_\phi$:*

$$\nabla_\phi J(\phi) = \nabla_\phi V_{g_\phi}(\boldsymbol{x}_L) = \sum_{\boldsymbol{x} \in \mathcal{X}_1 \cup \mathcal{X}_2 \cdots \cup \mathcal{X}_L} p_{g_\phi}(\boldsymbol{x}) \sum_a Q_{g_\phi}(\boldsymbol{x}, a)\nabla_\phi g_\phi(a|\boldsymbol{x}). \tag{42}$$

*Proof.* By definition,

$$J(\phi) = \mathbb{E}_{\tau \sim p_{g_\phi}(\cdot \mid \boldsymbol{x}_L)}[r(\boldsymbol{x}_0)].$$

Using the log-derivative trick,

$$\nabla_\phi J(\phi) = \sum_\tau r(\boldsymbol{x}_0)\, \nabla_\phi p_{g_\phi}(\tau \mid \boldsymbol{x}_L)$$

$$= \sum_\tau r(\boldsymbol{x}_0)\, p_{g_\phi}(\tau \mid \boldsymbol{x}_L)\, \nabla_\phi \log p_{g_\phi}(\tau \mid \boldsymbol{x}_L).$$

Since only the policy $g_\phi$ depends on $\phi$, we have

$$\log p_{g_\phi}(\tau \mid \boldsymbol{x}_L) = \sum_{n=1}^{L} \log g_\phi(a_n \mid \boldsymbol{x}_n) + \text{const},$$

and therefore

$$\nabla_\phi \log p_{g_\phi}(\tau \mid \boldsymbol{x}_L) = \sum_{n=1}^{L} \nabla_\phi \log g_\phi(a_n \mid \boldsymbol{x}_n).$$

Thus,

$$\nabla_\phi J(\phi) = \mathbb{E}_{\tau \sim p_{g_\phi}(\cdot \mid \boldsymbol{x}_L)}\left[ r(\boldsymbol{x}_0) \sum_{n=1}^{L} \nabla_\phi \log g_\phi(a_n \mid \boldsymbol{x}_n) \right].$$

Taking conditional expectations with respect to $(\boldsymbol{x}_n, a_n)$ yields for each $i \in \{1, \ldots, L\}$ we have

$$\mathbb{E}_\tau\Big[ r(\boldsymbol{x}_0)\, \nabla_\phi \log g_\phi(a_n \mid \boldsymbol{x}_n) \Big] \tag{43}$$

$$= \mathbb{E}_{\boldsymbol{x}_n, a_n}\Big[ \mathbb{E}[r(\boldsymbol{x}_0) \mid \boldsymbol{x}_n, a_n]\, \nabla_\phi \log g_\phi(a_n \mid \boldsymbol{x}_n) \Big] \qquad \text{(tower property)}$$

$$= \mathbb{E}_{\boldsymbol{x}_n, a_n}\Big[ Q_{g_\phi}(\boldsymbol{x}_n, a_n)\, \nabla_\phi \log g_\phi(a_n \mid \boldsymbol{x}_n) \Big] \tag{44}$$

$$= \sum_{\boldsymbol{x}_n \in \mathcal{X}_n} \sum_{a_n \in \mathcal{A}_n} p_{g_\phi}(\boldsymbol{x}_n, a_n)\, Q_{g_\phi}(\boldsymbol{x}_n, a_n)\, \nabla_\phi \log g_\phi(a_n \mid \boldsymbol{x}_n) \tag{45}$$

$$= \sum_{\boldsymbol{x}_n \in \mathcal{X}_n} p_{g_\phi}(\boldsymbol{x}_n) \sum_{a_n \in \mathcal{A}_n} g_\phi(a_n \mid \boldsymbol{x}_n)\, Q_{g_\phi}(\boldsymbol{x}_n, a_n)\, \nabla_\phi \log g_\phi(a_n \mid \boldsymbol{x}_n)$$

$$\qquad\qquad\qquad\qquad (p(\boldsymbol{x}_n, a_n) = p(\boldsymbol{x}_n)g(a_n \mid \boldsymbol{x}_n))$$

$$= \sum_{\boldsymbol{x}_n \in \mathcal{X}_n} p_{g_\phi}(\boldsymbol{x}_n) \sum_{a_n \in \mathcal{A}_n} Q_{g_\phi}(\boldsymbol{x}_n, a_n)\, \nabla_\phi g_\phi(a_n \mid \boldsymbol{x}_n), \qquad (g\,\nabla \log g = \nabla g)$$

where $p_{g_\phi}(\boldsymbol{x}_n) = \Pr_{\tau \sim p_{g_\phi}(\cdot | \boldsymbol{x}_L)}(X_n = \boldsymbol{x}_n)$ and $p_{g_\phi}(\boldsymbol{x}_n, a_n) = p_{g_\phi}(\boldsymbol{x}_n) g_\phi(a_n | \boldsymbol{x}_n)$ are the marginals induced by the rollout distribution. Therefore,

$$\nabla_\phi J(\phi) = \mathbb{E}_{\tau \sim p_{g_\phi}(\cdot | \boldsymbol{x}_L)} \left[ r(\boldsymbol{x}_0) \sum_{n=1}^{L} \nabla_\phi \log g_\phi(a_n | \boldsymbol{x}_n) \right]$$

$$= \sum_{\boldsymbol{x} \in \mathcal{X}_1 \cup \mathcal{X}_2 \cdots \cup \mathcal{X}_L} p_{g_\phi}(\boldsymbol{x}) \sum_a Q_{g_\phi}(\boldsymbol{x}, a) \nabla_\phi g_\phi(a | \boldsymbol{x}),$$

which ends the proof. $\qquad\qquad\qquad\qquad\qquad\qquad\qquad\qquad\qquad\qquad\qquad\qquad\quad\square$

This can also be proved in another way, just iteratively deriving the $\nabla_\phi V_{g_\phi}$. Using Lemma 2, we can prove our main objective as follows:

**Proposition 1** (Output–Token Level Gradient Alignment (informal)). *For MDM $\pi_\theta$ and unmasking policy model $g_\phi$ where $p_{g_\phi}(\boldsymbol{x}_{n-1} | \boldsymbol{x}_n) = g_\phi(a_n | \boldsymbol{x}_n) \cdot \pi_\theta(\boldsymbol{x}_{n-1} | \boldsymbol{x}_n, a_n)$ and $p_{g_\phi}(\boldsymbol{x}_0 | \mathbf{q}) = \sum_{\boldsymbol{x}_1, \ldots, \boldsymbol{x}_L} p_\phi(\boldsymbol{x}_{0:L} | \mathbf{q})$, consider the output level loss $\mathcal{L}_{\text{output}}$ and token-wise policy loss $\mathcal{L}_{\text{token}}$:*

$$\mathcal{L}_{\text{output}}(\phi) = \mathbb{E}_{\mathbf{q} \sim \rho_\mathcal{Q}} \mathbb{E}_{\boldsymbol{x}_0 \sim p_{g_{\phi_{\text{old}}}}(\cdot | \mathbf{q})} \left[ \frac{p_{g_\phi}(\boldsymbol{x}_0 | \mathbf{q})}{p_{g_{\phi_{\text{old}}}}(\boldsymbol{x}_0 | \mathbf{q})} A(\mathbf{q}, \boldsymbol{x}_0) \right], \tag{16}$$

$$\mathcal{L}_{\text{token}}(\phi) = \mathbb{E}_{\mathbf{q} \sim \rho_\mathcal{Q}} \mathbb{E}_{\{a_{1:L}, \boldsymbol{x}_{0:L}\} \sim p_{g_{\phi_{\text{old}}}}(\cdot | \mathbf{q})} \left[ \sum_{n=1}^{L} \frac{g_\phi(a_n | \boldsymbol{x}_n)}{g_{\phi_{\text{old}}}(a_n | \boldsymbol{x}_n)} A(\mathbf{q}, \boldsymbol{x}_0) \right]. \tag{17}$$

*Assume that we optimize $\phi$ carefully such that $\phi \approx \phi_{old}$, the two gradients are approximately equal:*

$$\nabla_\phi \mathcal{L}_{\text{token}} \approx \nabla_\phi \mathcal{L}_{\text{output}}, \tag{18}$$

*and $\nabla_\phi \mathcal{L}_{\text{token}} = \nabla_\phi \mathcal{L}_{\text{output}}$ at first optimization step where $\phi = \phi_{\text{old}}$.*

*Proof.* We derive $\nabla_\phi \mathcal{L}_{\text{output}}$ and $\nabla_\phi \mathcal{L}_{\text{token}}$ separately. Then, by applying Lemma 2 to $\nabla_\phi \mathcal{L}_{\text{output}}$, we obtain a form that matches $\nabla_\phi \mathcal{L}_{\text{token}}$. First, expanding $\mathcal{L}_{\text{output}}$ shows that it is equivalent to maximizing the advantage from the initial state:

$$\mathcal{L}_{\text{output}}(\phi) = \mathbb{E}_{\mathbf{q} \sim \rho_\mathcal{Q}} \mathbb{E}_{\boldsymbol{x}_0 \sim p_{g_{\phi_{\text{old}}}}(\cdot | \mathbf{q})} \left[ \frac{p_{g_\phi}(\boldsymbol{x}_0 | \mathbf{q})}{p_{g_{\phi_{\text{old}}}}(\boldsymbol{x}_0 | \mathbf{q})} A(\mathbf{q}, \boldsymbol{x}_0) \right] \tag{46}$$

$$= \mathbb{E}_{\mathbf{q} \sim \rho_\mathcal{Q}} \left[ \sum_{\boldsymbol{x}_0 \in \mathcal{X}} p_{g_{\phi_{\text{old}}}}(\boldsymbol{x}_0 | \mathbf{q}) \frac{p_{g_\phi}(\boldsymbol{x}_0 | \mathbf{q})}{p_{g_{\phi_{\text{old}}}}(\boldsymbol{x}_0 | \mathbf{q})} A(\mathbf{q}, \boldsymbol{x}_0) \right] \tag{47}$$

$$= \mathbb{E}_{\mathbf{q} \sim \rho_\mathcal{Q}} \left[ \sum_{\boldsymbol{x}_0 \in \mathcal{X}} p_{g_\phi}(\boldsymbol{x}_0 | \mathbf{q}) A(\mathbf{q}, \boldsymbol{x}_0) \right] \tag{48}$$

$$= \mathbb{E}_{\mathbf{q} \sim \rho_\mathcal{Q}} \left[ \sum_{\boldsymbol{x}_0 \in \mathcal{X}} \sum_{\boldsymbol{x}_1, \ldots \boldsymbol{x}_L} p_{g_\phi}(\boldsymbol{x}_{0:L} | \mathbf{q}) A(\mathbf{q}, \boldsymbol{x}_0) \right] \tag{49}$$

$$= \mathbb{E}_{\mathbf{q} \sim \rho_\mathcal{Q}} \left[ \sum_{\boldsymbol{x}_0, \boldsymbol{x}_1, \ldots \boldsymbol{x}_L} p_{g_\phi}(\boldsymbol{x}_{0:L} | \mathbf{q}) A(\mathbf{q}, \boldsymbol{x}_0) \right] \tag{50}$$

$$= \mathbb{E}_{\mathbf{q} \sim \rho_\mathcal{Q}} \mathbb{E}_{\boldsymbol{x}_{0:L} \sim p_{g_\phi}(\cdot | \mathbf{q})} \left[ A(\mathbf{q}, \boldsymbol{x}_0) \right] \tag{51}$$

Now we derive $\nabla_\phi \mathcal{L}_{\text{output}}$ as follows:

$$\nabla_\phi \mathcal{L}_{\text{output}}(\phi) = \nabla_\phi \left[ \mathbb{E}_{\mathbf{q} \sim \rho_Q} \mathbb{E}_{\boldsymbol{x}_{0:L} \sim p_{g_\phi}(\cdot | \mathbf{q})} \left[ A(\mathbf{q}, \boldsymbol{x}_0) \right] \right] \tag{52}$$

$$= \nabla_\phi \left[ \mathbb{E}_{\mathbf{q} \sim \rho_Q} \left[ \sum_{\boldsymbol{x}_0, \boldsymbol{x}_1, \ldots \boldsymbol{x}_L} p_{g_\phi}(\boldsymbol{x}_{0:L} \mid \mathbf{q}) A(\mathbf{q}, \boldsymbol{x}_0) \right] \right] \tag{53}$$

$$= \nabla_\phi \left[ \mathbb{E}_{\mathbf{q} \sim \rho_Q} \left[ \sum_{\boldsymbol{x}_0, \boldsymbol{x}_1, \ldots \boldsymbol{x}_L} p_{g_\phi}(\boldsymbol{x}_{0:L} \mid \mathbf{q}) \cdot \frac{(r(\mathbf{q}, \boldsymbol{x}_0) - r_{g_{\phi_{\text{old}}}}(\mathbf{q}))}{\text{std}_{g_{\phi_{\text{old}}}}(\mathbf{q}) + \epsilon} \right] \right] \tag{54}$$

$$= \nabla_\phi \left[ \mathbb{E}_{\mathbf{q} \sim \rho_Q} \left[ \sum_{\boldsymbol{x}_0, \boldsymbol{x}_1, \ldots \boldsymbol{x}_L} p_{g_\phi}(\boldsymbol{x}_{0:L} \mid \mathbf{q}) \frac{r(\mathbf{q}, \boldsymbol{x}_0)}{\text{std}_{g_{\phi_{\text{old}}}}(\mathbf{q}) + \epsilon} \right] \right]$$
$$- \nabla_\phi \left[ \mathbb{E}_{\mathbf{q} \sim \rho_Q} \left[ \sum_{\boldsymbol{x}_0, \boldsymbol{x}_1, \ldots \boldsymbol{x}_L} p_{g_\phi}(\boldsymbol{x}_{0:L} \mid \mathbf{q}) \frac{r_{g_{\phi_{\text{old}}}}(\mathbf{q}))}{\text{std}_{g_{\phi_{\text{old}}}}(\mathbf{q}) + \epsilon} \right] \right] \tag{55}$$

$$= \nabla_\phi \left[ \mathbb{E}_{\mathbf{q} \sim \rho_Q} \left[ \sum_{\boldsymbol{x}_0, \boldsymbol{x}_1, \ldots \boldsymbol{x}_L} p_{g_\phi}(\boldsymbol{x}_{0:L} \mid \mathbf{q}) \frac{r(\mathbf{q}, \boldsymbol{x}_0)}{\text{std}_{g_{\phi_{\text{old}}}}(\mathbf{q}) + \epsilon} \right] \right]$$
$$- \mathbb{E}_{\mathbf{q} \sim \rho_Q} \left[ \frac{r_{g_{\phi_{\text{old}}}}(\mathbf{q}))}{\text{std}_{g_{\phi_{\text{old}}}}(\mathbf{q}) + \epsilon} \nabla_\phi \left[ \underbrace{\sum_{\boldsymbol{x}_0, \boldsymbol{x}_1, \ldots \boldsymbol{x}_L} p_{g_\phi}(\boldsymbol{x}_{0:L} \mid \mathbf{q})}_{=1} \right] \right] \tag{56}$$

$$= \nabla_\phi \left[ \mathbb{E}_{\mathbf{q} \sim \rho_Q} \left[ \sum_{\boldsymbol{x}_0, \boldsymbol{x}_1, \ldots \boldsymbol{x}_L} p_{g_\phi}(\boldsymbol{x}_{0:L} \mid \mathbf{q}) \frac{r(\mathbf{q}, \boldsymbol{x}_0)}{\text{std}_{g_{\phi_{\text{old}}}}(\mathbf{q}) + \epsilon} \right] \right] \tag{57}$$

$$= \nabla_\phi \left[ \mathbb{E}_{\mathbf{q} \sim \rho_Q} \frac{\mathbb{E}_{\boldsymbol{x}_{0:L} \sim p_{g_\phi}(\cdot | \mathbf{q})} \left[ r(\mathbf{q}, \boldsymbol{x}_0) \right]}{\text{std}_{g_{\phi_{\text{old}}}}(\mathbf{q}) + \epsilon} \right] = \mathbb{E}_{\mathbf{q} \sim \rho_Q} \left[ \frac{\nabla_\phi V_{g_\phi}(\mathbf{q})}{\text{std}_{g_{\phi_{\text{old}}}}(\mathbf{q}) + \epsilon} \right], \tag{58}$$

where we define state-wise value function as the expectation of terminal reward starting from the state, *i.e.*, $V_{g_\phi}(\boldsymbol{x}_n | \mathbf{q}) := \mathbb{E}_{\boldsymbol{x}_{0:n-1} \sim p_{g_\phi}(\cdot | \mathbf{q}, \boldsymbol{x}_n)} \left[ r(\mathbf{q}, \boldsymbol{x}_0) \right]$ and we denote value function at the initial state as $V_{g_\phi}(\mathbf{q}) := \mathbb{E}_{\boldsymbol{x}_{0:L} \sim p_{g_\phi}(\cdot | \mathbf{q})} \left[ r(\mathbf{q}, \boldsymbol{x}_0) \right]$. Accordingly, state-wise action-value function can be defined as $Q_{g_\phi}(\boldsymbol{x}_n, a_n | \mathbf{q}) = \sum_{\boldsymbol{x}_{n-1}} \pi_\theta(\boldsymbol{x}_{n-1} | \boldsymbol{x}_n, a_n) \cdot V_{g_\phi}(\boldsymbol{x}_{n-1} | \mathbf{q})$. Note that Eq. (52)-57 is conceptually equivalent to reversing the derivation process of REINFORCE with baseline (Williams, 1992). Since we want to show that $\nabla_\phi \mathcal{L}_{\text{token}} \propto \nabla_\phi \mathcal{L}_{\text{output}}$, we breakdown $\nabla_\phi V_{g_\phi}(\mathbf{q})$ using Lemma 2:

$$\nabla_\phi \mathcal{L}_{\text{output}}(\phi) = \mathbb{E}_{\mathbf{q} \sim \rho_Q} \left[ \frac{\nabla_\phi V_{g_\phi}(\mathbf{q})}{\text{std}_{g_{\phi_{\text{old}}}}(\mathbf{q}) + \epsilon} \right] \tag{59}$$

$$= \mathbb{E}_{\mathbf{q} \sim \rho_Q} \left[ \frac{\sum_{\boldsymbol{x}} p_{g_\phi}(\boldsymbol{x} | \mathbf{q}) \sum Q_{g_\phi}(\boldsymbol{x}, a | \mathbf{q}) \nabla_\phi g_\phi(a | \boldsymbol{x})}{\text{std}_{g_{\phi_{\text{old}}}}(\mathbf{q}) + \epsilon} \right] \tag{60}$$

$$\approx \mathbb{E}_{\mathbf{q} \sim \rho_Q} \left[ \frac{\sum_{\boldsymbol{x}} p_{g_{\phi_{\text{old}}}}(\boldsymbol{x} | \mathbf{q}) \sum Q_{g_{\phi_{\text{old}}}}(\boldsymbol{x}, a | \mathbf{q}) \nabla_\phi g_\phi(a | \boldsymbol{x})}{\text{std}_{g_{\phi_{\text{old}}}}(\mathbf{q}) + \epsilon} \right]. \tag{61}$$

where $\boldsymbol{x}$ comes from every state, *i.e.*, $\boldsymbol{x} \in \mathcal{X}_0 \cup \mathcal{X}_1 \cup \cdots \cup \mathcal{X}_L$. Here, Eq. (59) = Eq. (61) exactly holds at the first optimization step where $\phi = \phi_{\text{old}}$. On the other side, we now show that $\nabla_\phi \mathcal{L}_{\text{token}}$ is equivalent to Eq. (61). We start by converting the expectation into a probability-weighted sum as follows:

$$\mathcal{L}_{\text{token}}(\phi) = \mathbb{E}_{\mathbf{q} \sim \rho_Q} \mathbb{E}_{\boldsymbol{x}_{0:L}, a_{1:L} \sim p_{g_{\phi_{\text{old}}}}(\cdot | \mathbf{q})} \left[ \sum_{n=1}^{L} \frac{g_\phi(a_n \mid \boldsymbol{x}_n)}{g_{\phi_{\text{old}}}(a_n \mid \boldsymbol{x}_n)} A(\mathbf{q}, \boldsymbol{x}_0) \right] \tag{62}$$

$$= \mathbb{E}_{\mathbf{q} \sim \rho_Q} \left[ \sum_{\substack{\boldsymbol{x}_0, \ldots, \boldsymbol{x}_L \\ a_1, \ldots, a_L}} \prod_{n=1}^{L} \left( \pi_\theta(\boldsymbol{x}_{n-1} | \boldsymbol{x}_n, a_n) \cdot g_{\phi_{\text{old}}}(a_n, \boldsymbol{x}_n) \right) \left( \sum_{n=1}^{L} \frac{g_\phi(a_n \mid \boldsymbol{x}_n)}{g_{\phi_{\text{old}}}(a_n \mid \boldsymbol{x}_n)} A(\mathbf{q}, \boldsymbol{x}_0) \right) \right]. \tag{63}$$

For further explanation, we want to recap $p_{g_{\phi_{\text{old}}}}(\boldsymbol{x}_{n-1}|\boldsymbol{x}_n) = \pi_\theta(\boldsymbol{x}_{n-1}|\boldsymbol{x}_n, a_n) \cdot g_{\phi_{\text{old}}}(a_n|\boldsymbol{x}_n)$. Note that $p_{g_{\phi_{\text{old}}}} > 0$ only if $\boldsymbol{x}_n^{a_n} = \mathbb{M}$ and $\boldsymbol{x}_{n-1}^{a_n} \neq \mathbb{M}$ since $g_{\phi_{\text{old}}}(a_n|\boldsymbol{x}_n) = 0$ for $\boldsymbol{x}_n^{a_n} \neq \mathbb{M}$, $\pi_\theta(\boldsymbol{x}_{n-1}|\boldsymbol{x}_n, a_n) = 0$ for all $\boldsymbol{x}_{n-1}$ satisfying $x_{n-1}^{a_n} = \mathbb{M}$ by definition. To simplify the derivation process, we consider $\widehat{\mathcal{L}}_{\text{token,i}}(\phi, \mathbf{q})$ defined as follows:

$$\widehat{\mathcal{L}}_{\text{token,i}}(\phi, \mathbf{q}) = \sum_{\substack{\boldsymbol{x}_0,\ldots,\boldsymbol{x}_L \\ a_1,\ldots,a_L}} p_{g_{\phi_{\text{old}}}}(\boldsymbol{x}_{0:L}|\mathbf{q}) \frac{g_\phi(a_i \mid \boldsymbol{x}_i)}{g_{\phi_{\text{old}}}(a_i \mid \boldsymbol{x}_i)} r(\mathbf{q}, \boldsymbol{x}_0) \tag{64}$$

$$= \sum_{\substack{\boldsymbol{x}_0,\ldots,\boldsymbol{x}_L \\ a_1,\ldots,a_L}} \left( (\prod_{j=1}^L p_{g_{\phi_{\text{old}}}}(\boldsymbol{x}_{j-1}|\boldsymbol{x}_j, \mathbf{q})) \frac{g_\phi(a_i \mid \boldsymbol{x}_i)}{g_{\phi_{\text{old}}}(a_i \mid \boldsymbol{x}_i)} r(\mathbf{q}, \boldsymbol{x}_0) \right) \tag{65}$$

$$= \sum_{\substack{\boldsymbol{x}_0,\ldots,\boldsymbol{x}_L \\ a_1,\ldots,a_L}} \left( (\prod_{j=i+1}^L p_{g_{\phi_{\text{old}}}}(\boldsymbol{x}_{j-1}|\boldsymbol{x}_j, \mathbf{q})) \right.$$
$$\left. \cdot (p_{g_{\phi_{\text{old}}}}(\boldsymbol{x}_{i-1}|\boldsymbol{x}_i, \mathbf{q}) \frac{g_\phi(a_i \mid \boldsymbol{x}_i)}{g_{\phi_{\text{old}}}(a_i \mid \boldsymbol{x}_i)}) \cdot (\prod_{j=1}^{i-1} p_{g_{\phi_{\text{old}}}}(\boldsymbol{x}_{j-1}|\boldsymbol{x}_j, \mathbf{q})) \cdot r(\mathbf{q}, \boldsymbol{x}_0) \right) \tag{66}$$

$$= \sum_{\substack{\boldsymbol{x}_0,\ldots,\boldsymbol{x}_L \\ a_1,\ldots,a_L}} \left( (\prod_{j=i+1}^L p_{g_{\phi_{\text{old}}}}(\boldsymbol{x}_{j-1}|\boldsymbol{x}_j, \mathbf{q})) \right.$$
$$\left. (\pi_\theta(\boldsymbol{x}_{i-1}|\boldsymbol{x}_i, a_i, \mathbf{q}) g_\phi(a_i \mid \boldsymbol{x}_i)) \cdot (\prod_{j=1}^{i-1} p_{g_{\phi_{\text{old}}}}(\boldsymbol{x}_{j-1}|\boldsymbol{x}_j, \mathbf{q})) \cdot r(\mathbf{q}, \boldsymbol{x}_0) \right) \tag{67}$$

$$= \sum_{\substack{\boldsymbol{x}_i,\ldots,\boldsymbol{x}_L \\ a_{i+1},\ldots,a_L}} \left( (\prod_{j=i+1}^L p_{g_{\phi_{\text{old}}}}(\boldsymbol{x}_{j-1}|\boldsymbol{x}_j, \mathbf{q})) \right.$$
$$\left. \cdot \sum_{\substack{\boldsymbol{x}_{i-1},\ldots,\boldsymbol{x}_0 \\ a_i,\ldots,a_1}} \left( (\pi_\theta(\boldsymbol{x}_{i-1}|\boldsymbol{x}_i, a_i, \mathbf{q}) g_\phi(a_i \mid \boldsymbol{x}_i)) \cdot (\prod_{j=1}^{i-1} p_{g_{\phi_{\text{old}}}}(\boldsymbol{x}_{j-1}|\boldsymbol{x}_j, \mathbf{q})) \cdot r(\mathbf{q}, \boldsymbol{x}_0) \right) \right) \tag{68}$$

$$= \sum_{\substack{\boldsymbol{x}_i,\ldots,\boldsymbol{x}_L \\ a_{i+1},\ldots,a_L}} \left( (\prod_{j=i+1}^L p_{g_{\phi_{\text{old}}}}(\boldsymbol{x}_{j-1}|\boldsymbol{x}_j, \mathbf{q})) \right.$$
$$\left. \cdot \sum_{a_i} \sum_{\boldsymbol{x}_{i-1}} \left( g_\phi(a_i \mid \boldsymbol{x}_i) \cdot \pi_\theta(\boldsymbol{x}_{i-1}|\boldsymbol{x}_i, a_i, \mathbf{q}) \sum_{\substack{\boldsymbol{x}_{i-2},\ldots,\boldsymbol{x}_0 \\ a_{i-1},\ldots,a_1}} \left( (\prod_{j=1}^{i-1} p_{g_{\phi_{\text{old}}}}(\boldsymbol{x}_{j-1}|\boldsymbol{x}_j, \mathbf{q})) \cdot r(\mathbf{q}, \boldsymbol{x}_0) \right) \right) \right) \tag{69}$$

$$= \sum_{\boldsymbol{x}_i,\ldots\boldsymbol{x}_L} \sum_{a_{i+1},\ldots,a_L} \left( (\prod_{j=i+1}^L p_{g_{\phi_{\text{old}}}}(\boldsymbol{x}_{j-1}|\boldsymbol{x}_j, \mathbf{q})) \right.$$
$$\left. \cdot \sum_{a_i} g_\phi(a_i \mid \boldsymbol{x}_i) \cdot \sum_{\boldsymbol{x}_{i-1}} \left( \pi_\theta(\boldsymbol{x}_{i-1}|\boldsymbol{x}_i, a_i, \mathbf{q}) \sum_{\substack{\boldsymbol{x}_{i-2},\ldots,\boldsymbol{x}_0 \\ a_{i-1},\ldots,a_1}} \left( (\prod_{j=1}^{i-1} p_{g_{\phi_{\text{old}}}}(\boldsymbol{x}_{j-1}|\boldsymbol{x}_j, \mathbf{q})) \cdot r(\mathbf{q}, \boldsymbol{x}_0) \right) \right) \right) \tag{70}$$

$$= \sum_{\boldsymbol{x}_i,\ldots\boldsymbol{x}_L} \sum_{a_{i+1},\ldots,a_L} \left( (\prod_{j=i+1}^L p_{g_{\phi_{\text{old}}}}(\boldsymbol{x}_{j-1}|\boldsymbol{x}_j, \mathbf{q})) \right.$$
$$\left. \cdot \sum_{a_i} g_\phi(a_i \mid \boldsymbol{x}_i) \cdot \sum_{\boldsymbol{x}_{i-1}} \left( \pi_\theta(\boldsymbol{x}_{i-1}|\boldsymbol{x}_i, a_i, \mathbf{q}) \cdot V_{\phi_{\text{old}}}(\boldsymbol{x}_{i-1}|\mathbf{q}) \right) \right) \tag{71}$$

$$= \mathbb{E}_{\boldsymbol{x}_i \sim p_{g_{\phi_{\text{old}}}}(\cdot|\mathbf{q})} \left[ \sum g_\phi(a_i|\boldsymbol{x}_i) \cdot Q_{g_{\phi_{\text{old}}}}(\boldsymbol{x}_i, a_i|\mathbf{q}) \right], \tag{72}$$

where we color parentheses by their nesting depth to improve readability of the multi-level factorization: outer, middle, inner, and deepest inner. The expansion proceeds by repeatedly pulling out terms that are independent of the summation variables, and keeping only dependent factors inside the corresponding sums. Then,

$$\nabla_\phi \mathcal{L}_{\text{token}}(\phi) = \sum_{i=1}^{L} \mathbb{E}_{\mathbf{q} \sim \rho_Q} \left[ \frac{\widehat{\mathcal{L}}_{\text{token,i}}(\phi)}{\text{std}_{g_{\phi_{\text{old}}}}(\mathbf{q}) + \epsilon} \right] \tag{73}$$

$$= \mathbb{E}_{\mathbf{q} \sim \rho_Q} \frac{\sum_{i=1}^{L} \mathbb{E}_{\boldsymbol{x}_i \sim p_{g_{\phi_{\text{old}}}}(\cdot|\mathbf{q})} [\sum \nabla_\phi g_\phi(a_i|\boldsymbol{x}_i) \cdot Q_{g_{\phi_{\text{old}}}}(\boldsymbol{x}_i, a_i|\mathbf{q})]}{\text{std}_{g_{\phi_{\text{old}}}}(\mathbf{q}) + \epsilon} \tag{74}$$

$$= \mathbb{E}_{\mathbf{q} \sim \rho_Q} \left[ \frac{\sum_{\boldsymbol{x}} p_{g_{\phi_{\text{old}}}}(\boldsymbol{x}|\mathbf{q}) \sum Q_{g_{\phi_{\text{old}}}}(\boldsymbol{x}, a|\mathbf{q}) \nabla_\phi g_\phi(a|\boldsymbol{x}, \mathbf{q})}{\text{std}_{g_{\phi_{\text{old}}}}(\mathbf{q}) + \epsilon} \right]. \tag{75}$$

Eq. (73) holds where $r_{g_{\phi_{\text{old}}}}(\mathbf{q})$ in $A(\mathbf{q}, \boldsymbol{x}_0) = (r(q, \boldsymbol{x}_0) - r_{g_{\phi_{\text{old}}}}(\mathbf{q})) / (\text{std}_{g_{\phi_{\text{old}}}}(\mathbf{q}) + \epsilon)$ can be deleted as we have done in $\nabla_\phi \mathcal{L}_{\text{output}}$. Since Eq. (61) and Eq. (75) are identical, we conclude $\nabla_\phi \mathcal{L}_{\text{token}} \approx \nabla_\phi \mathcal{L}_{\text{output}}$ and $\nabla_\phi \mathcal{L}_{\text{token}} = \nabla_\phi \mathcal{L}_{\text{output}}$ at the first optimization step where $\phi = \phi_{\text{old}}$. $\qquad \square$

### C.3 SURROGATE LOSS OF OUTPUT-LEVEL KL DIVERGENCE

Our final goal is to provide evidence that by performing gradient ascent on $\mathcal{L}_{\text{KL}}$, we can expect to decrease $D_{\text{KL}}(p_{g_\phi}(\boldsymbol{x}_0|\mathbf{q}) \| p_{g_{\text{ref}}}(\boldsymbol{x}_0|\mathbf{q}))$. We provide two propositions here. The first proposition shows that trajectory-wise KL divergence is the upper bound of output-level KL divergence. The second proposition shows that the gradient of trajectory-wise KL divergence is equivalent to $\mathcal{L}_{\text{KL}}$.

**Proposition 2.** *For MDMs where* $p_{g_\phi}(\boldsymbol{x}_0|\mathbf{q}) = \sum_{\boldsymbol{x}_1,\dots,\boldsymbol{x}_L} p_\phi(\boldsymbol{x}_{0:L}|\mathbf{q})$, *consider the terminal KL divergence* $D_{\text{KL}}(p_{g_\phi}(\boldsymbol{x}_0|\mathbf{q}) \| p_{g_{\text{ref}}}(\boldsymbol{x}_0|\mathbf{q}))$ *and trajectory-wise KL divergence* $D_{\text{KL}}(p_{g_\phi}(\boldsymbol{x}_{0:L}|\mathbf{q}) \| p_{g_{\text{ref}}}(\boldsymbol{x}_{0:L}|\mathbf{q}))$ *defined as follows:*

$$D_{\text{KL}}(p_{g_\phi}(\boldsymbol{x}_0|\mathbf{q}) \| p_{g_{\text{ref}}}(\boldsymbol{x}_0|\mathbf{q})) = \mathbb{E}_{\tau \sim p_{g_\phi}(\boldsymbol{x}_0|\mathbf{q})} \left[ \log \frac{p_{g_\phi}(\boldsymbol{x}_0|\mathbf{q})}{p_{g_{\text{ref}}}(\boldsymbol{x}_0|\mathbf{q})} \right], \tag{76}$$

$$D_{\text{KL}}(p_{g_\phi}(\boldsymbol{x}_{0:L}|\mathbf{q}) \| p_{g_{\text{ref}}}(\boldsymbol{x}_{0:L}|\mathbf{q})) = \mathbb{E}_{\boldsymbol{x}_{0:L} \sim p_{g_\phi}(\boldsymbol{x}_{0:L}|\mathbf{q})} \left[ \log \frac{p_{g_\phi}(\boldsymbol{x}_{0:L}|\mathbf{q})}{p_{g_{\text{ref}}}(\boldsymbol{x}_{0:L}|\mathbf{q})} \right] \tag{77}$$

*Then,* $D_{\text{KL}}(p_{g_\phi}(\boldsymbol{x}_{0:L}|\mathbf{q}) \| p_{g_{\text{ref}}}(\boldsymbol{x}_{0:L}|\mathbf{q}))$ *is upper bound of* $D_{\text{KL}}(p_{g_\phi}(\boldsymbol{x}_0|\mathbf{q}) \| p_{g_{\text{ref}}}(\boldsymbol{x}_0|\mathbf{q}))$:

$$D_{\text{KL}}(p_{g_\phi}(\boldsymbol{x}_0|\mathbf{q}) \| p_{g_{\text{ref}}}(\boldsymbol{x}_0|\mathbf{q})) \leq D_{\text{KL}}(p_{g_\phi}(\boldsymbol{x}_{0:L}|\mathbf{q}) \| p_{g_{\text{ref}}}(\boldsymbol{x}_{0:L}|\mathbf{q})). \tag{78}$$

*Proof.*

$$D_{\text{KL}}(p_{g_\phi}(\boldsymbol{x}_0 \mid \mathbf{q}) \| p_{g_{\text{ref}}}(\boldsymbol{x}_0 \mid \mathbf{q})) = \mathbb{E}_{\boldsymbol{x}_0 \sim p_{g_\phi}(\boldsymbol{x}_0|\mathbf{q})} \left[ \log \frac{p_{g_\phi}(\boldsymbol{x}_0 \mid \mathbf{q})}{p_{g_{\text{ref}}}(\boldsymbol{x}_0 \mid \mathbf{q})} \right] \tag{79}$$

$$= \mathbb{E}_{\boldsymbol{x}_{0:L} \sim p_{g_\phi}(\boldsymbol{x}_{0:L}|\mathbf{q})} \left[ \log \frac{p_{g_\phi}(\boldsymbol{x}_0 \mid \mathbf{q})}{p_{g_{\text{ref}}}(\boldsymbol{x}_0 \mid \mathbf{q})} \right] \tag{80}$$

$$= \mathbb{E}_{\boldsymbol{x}_{0:L} \sim p_{g_\phi}(\boldsymbol{x}_{0:L}|\mathbf{q})} \left[ \log \frac{p_{g_\phi}(\boldsymbol{x}_{0:L} \mid \mathbf{q})}{p_{g_{\text{ref}}}(\boldsymbol{x}_{0:L} \mid \mathbf{q})} - \log \frac{p_{g_\phi}(\boldsymbol{x}_{1:L} \mid \boldsymbol{x}_0, \mathbf{q})}{p_{g_{\text{ref}}}(\boldsymbol{x}_{1:L} \mid \boldsymbol{x}_0, \mathbf{q})} \right] \tag{81}$$

$$= D_{\text{KL}}(p_{g_\phi}(\boldsymbol{x}_{0:L} \mid \mathbf{q}), |, p_{g_{\text{ref}}}(\boldsymbol{x}_{0:L} \mid \mathbf{q})) \tag{82}$$

$$- \mathbb{E}_{\boldsymbol{x}_0 \sim p_{g_\phi}(\boldsymbol{x}_0|\mathbf{q})} \left[ D_{\text{KL}}(p_{g_\phi}(\boldsymbol{x}_{1:L} \mid \boldsymbol{x}_0, \mathbf{q}) \| p_{g_{\text{ref}}}(\boldsymbol{x}_{1:L} \mid \boldsymbol{x}_0, \mathbf{q})) \right] \tag{83}$$

$$\leq D_{\text{KL}}(p_{g_\phi}(\boldsymbol{x}_{0:L} \mid \mathbf{q}) \| p_{g_{\text{ref}}}(\boldsymbol{x}_{0:L} \mid \mathbf{q})). \tag{84}$$

$$\square$$

Therefore, we can lower $D_{\text{KL}}(p_{g_\phi}(\boldsymbol{x}_0|\mathbf{q}) \| p_{g_{\text{ref}}}(\boldsymbol{x}_0|\mathbf{q}))$ by decreasing $D_{\text{KL}}(p_{g_\phi}(\boldsymbol{x}_{0:L}|\mathbf{q}) \| p_{g_{\text{ref}}}(\boldsymbol{x}_{0:L}|\mathbf{q}))$. Now we show that $\nabla_\phi D_{\text{KL}}(p_{g_\phi}(\boldsymbol{x}_{0:L}|\mathbf{q}) \| p_{g_{\text{ref}}}(\boldsymbol{x}_{0:L}|\mathbf{q})) = \nabla_\phi \mathcal{L}_{\text{KL}}$:

**Proposition 3.** *Consider the trajectory-wise KL divergence* $D_{\mathrm{KL}}\left(p_{g_\phi}(\boldsymbol{x}_{0:L}\mid\mathbf{q})\|p_{g_{\mathrm{ref}}}(\boldsymbol{x}_{0:L}\mid\mathbf{q})\right)$ *and tractable* $\mathcal{L}_{\mathrm{KL}}$ *defined as follows:*

$$\mathcal{L}_{\mathrm{KL}}(\boldsymbol{x}_{0:L}, a_{1:L}, \mathbf{q}) = \mathrm{StopGrad}\Big(\frac{p_{g_\phi}(\boldsymbol{x}_{0:L}|\mathbf{q})}{p_{g_{\phi_{\mathrm{old}}}}(\boldsymbol{x}_{0:L}|\mathbf{q})}\cdot\big(1+\log\frac{p_{g_\phi}(\boldsymbol{x}_{0:L}|\mathbf{q})}{p_{g_{\mathrm{ref}}}(\boldsymbol{x}_{0:L}|\mathbf{q})}\big)\Big)\cdot\sum_{n=1}^{L}\log g_\phi(a_n\mid\boldsymbol{x}_n) \tag{85}$$

*Then, the gradient of* $D_{\mathrm{KL}}\left(p_{g_\phi}(\boldsymbol{x}_{0:L}\mid\mathbf{q})\|p_{g_{\mathrm{ref}}}(\boldsymbol{x}_{0:L}\mid\mathbf{q})\right)$ *and* $\mathbb{E}[\mathcal{L}_{\mathrm{KL}}]$ *are equal:*

$$\nabla_\phi D_{\mathrm{KL}}\left(p_{g_\phi}(\boldsymbol{x}_{0:L}\mid\mathbf{q})\|p_{g_{\mathrm{ref}}}(\boldsymbol{x}_{0:L}\mid\mathbf{q})\right) = \nabla_\phi\mathbb{E}_{\boldsymbol{x}_{0:L}, a_{0:L}\sim p_{g_{\phi_{\mathrm{old}}}}}[\mathcal{L}_{\mathrm{KL}}] \tag{86}$$

*Proof.*

$$D_{\mathrm{KL}}\left(p_{g_\phi}(\boldsymbol{x}_{0:L}\mid\mathbf{q})\|p_{g_{\mathrm{ref}}}(\boldsymbol{x}_{0:L}\mid\mathbf{q})\right) \tag{87}$$

$$= \mathbb{E}_{\boldsymbol{x}_{0:L}\sim p_{g_\phi}(\boldsymbol{x}_{0:L}|\mathbf{q})}\left[\log\frac{p_{g_\phi}(\boldsymbol{x}_{0:L}\mid\mathbf{q})}{p_{g_{\mathrm{ref}}}(\boldsymbol{x}_{0:L}\mid\mathbf{q})}\right] \tag{88}$$

$$= \mathbb{E}_{\boldsymbol{x}_{0:L}\sim p_{g_{\phi_{\mathrm{old}}}}(\boldsymbol{x}_{0:L}|\mathbf{q})}\left[\frac{p_{g_\phi}(\boldsymbol{x}_{0:L}\mid\mathbf{q})}{p_{g_{\phi_{\mathrm{old}}}}(\boldsymbol{x}_{0:L}\mid\mathbf{q})}\cdot\log\frac{p_{g_\phi}(\boldsymbol{x}_{0:L}\mid\mathbf{q})}{p_{g_{\mathrm{ref}}}(\boldsymbol{x}_{0:L}\mid\mathbf{q})}\right] \tag{89}$$

$$= \mathbb{E}_{\boldsymbol{x}_{0:L}\sim p_{g_{\phi_{\mathrm{old}}}}(\boldsymbol{x}_{0:L}|\mathbf{q})}\left[\frac{p_{g_\phi}(\boldsymbol{x}_{0:L}\mid\mathbf{q})}{p_{g_{\phi_{\mathrm{old}}}}(\boldsymbol{x}_{0:L}\mid\mathbf{q})}\cdot\sum_{i=1}^{L}\log\frac{p_{g_\phi}(\boldsymbol{x}_{i-1}\mid\boldsymbol{x}_i,\mathbf{q})}{p_{g_{\mathrm{ref}}}(\boldsymbol{x}_{i-1}\mid\boldsymbol{x}_i,\mathbf{q})}\right] \tag{90}$$

where the change of measure from $p_{g_\phi}$ to $p_{g_{\phi_{\mathrm{old}}}}$ (Eq. (88) to Eq. (89)) is justified by the absolute continuity $p_{g_\phi}(\mathbf{x}_{0:L}|\mathbf{q}) \ll p_{g_{\phi_{\mathrm{old}}}}(\mathbf{x}_{0:L}|\mathbf{q})$.

$$\nabla_\phi D_{\mathrm{KL}}\left(p_{g_\phi}(\boldsymbol{x}_{0:L}\mid\mathbf{q})\|p_{g_{\mathrm{ref}}}(\boldsymbol{x}_{0:L}\mid\mathbf{q})\right) \tag{91}$$

$$= \nabla_\phi\mathbb{E}_{\boldsymbol{x}_{0:L}\sim p_{g_{\phi_{\mathrm{old}}}}(\boldsymbol{x}_{0:L}|\mathbf{q})}\left[\frac{p_{g_\phi}(\boldsymbol{x}_{0:L}\mid\mathbf{q})}{p_{g_{\phi_{\mathrm{old}}}}(\boldsymbol{x}_{0:L}\mid\mathbf{q})}\cdot\sum_{i=1}^{L}\log\frac{p_{g_\phi}(\boldsymbol{x}_{i-1}\mid\boldsymbol{x}_i,\mathbf{q})}{p_{g_{\mathrm{ref}}}(\boldsymbol{x}_{i-1}\mid\boldsymbol{x}_i,\mathbf{q})}\right] \tag{92}$$

$$= \mathbb{E}\left[\nabla_\phi\frac{p_{g_\phi}(\boldsymbol{x}_{0:L}\mid\mathbf{q})}{p_{g_{\phi_{\mathrm{old}}}}(\boldsymbol{x}_{0:L}\mid\mathbf{q})}\cdot\sum_{i=1}^{L}\log\frac{p_{g_\phi}(\boldsymbol{x}_{i-1}\mid\boldsymbol{x}_i,\mathbf{q})}{p_{g_{\mathrm{ref}}}(\boldsymbol{x}_{i-1}\mid\boldsymbol{x}_i,\mathbf{q})}\right.$$
$$\left. + \frac{p_{g_\phi}(\boldsymbol{x}_{0:L}\mid\mathbf{q})}{p_{g_{\phi_{\mathrm{old}}}}(\boldsymbol{x}_{0:L}\mid\mathbf{q})}\cdot\nabla_\phi\sum_{i=1}^{L}\log\frac{p_{g_\phi}(\boldsymbol{x}_{i-1}\mid\boldsymbol{x}_i,\mathbf{q})}{p_{g_{\mathrm{ref}}}(\boldsymbol{x}_{i-1}\mid\boldsymbol{x}_i,\mathbf{q})}\right] \tag{93}$$

$$= \mathbb{E}\left[\left(\sum_{i=1}^{L}\frac{\nabla_\phi p_{g_\phi}(\boldsymbol{x}_{i-1}\mid\boldsymbol{x}_i,\mathbf{q})}{p_{g_\phi}(\boldsymbol{x}_{i-1}\mid\boldsymbol{x}_i,\mathbf{q})}\right)\left(\frac{p_{g_\phi}(\boldsymbol{x}_{0:L}\mid\mathbf{q})}{p_{g_{\phi_{\mathrm{old}}}}(\boldsymbol{x}_{0:L}\mid\mathbf{q})}\right)\cdot\left(\sum_{i=1}^{L}\log\frac{p_{g_\phi}(\boldsymbol{x}_{i-1}\mid\boldsymbol{x}_i,\mathbf{q})}{p_{g_{\mathrm{ref}}}(\boldsymbol{x}_{i-1}\mid\boldsymbol{x}_i,\mathbf{q})}\right)\right.$$
$$\left. + \frac{p_{g_\phi}(\boldsymbol{x}_{0:L}\mid\mathbf{q})}{p_{g_{\phi_{\mathrm{old}}}}(\boldsymbol{x}_{0:L}\mid\mathbf{q})}\left(\sum_{i=1}^{L}\frac{\nabla_\phi p_{g_\phi}(\boldsymbol{x}_{i-1}\mid\boldsymbol{x}_i,\mathbf{q})}{p_{g_\phi}(\boldsymbol{x}_{i-1}\mid\boldsymbol{x}_i,\mathbf{q})}\right)\right] \tag{94}$$

$$= \mathbb{E}\left[\left(\frac{p_{g_\phi}(\boldsymbol{x}_{0:L}\mid\mathbf{q})}{p_{g_{\phi_{\mathrm{old}}}}(\boldsymbol{x}_{0:L}\mid\mathbf{q})}\right)\cdot\left(1+\sum_{i=1}^{L}\log\frac{p_{g_\phi}(\boldsymbol{x}_{i-1}\mid\boldsymbol{x}_i,\mathbf{q})}{p_{g_{\mathrm{ref}}}(\boldsymbol{x}_{i-1}\mid\boldsymbol{x}_i,\mathbf{q})}\right)\left(\sum_{i=1}^{L}\frac{\nabla_\phi p_{g_\phi}(\boldsymbol{x}_{i-1}\mid\boldsymbol{x}_i,\mathbf{q})}{p_{g_\phi}(\boldsymbol{x}_{i-1}\mid\boldsymbol{x}_i,\mathbf{q})}\right)\right] \tag{95}$$

$$= \nabla_\phi\mathbb{E}\left[\mathrm{StopGrad}\Big(\frac{p_{g_\phi}(\boldsymbol{x}_{0:L}\mid\mathbf{q})}{p_{g_{\phi_{\mathrm{old}}}}(\boldsymbol{x}_{0:L}\mid\mathbf{q})}\cdot(1+\log\frac{p_{g_\phi}(\boldsymbol{x}_{0:L}\mid\mathbf{q})}{p_{g_{\mathrm{ref}}}(\boldsymbol{x}_{0:L}\mid\mathbf{q})})\Big)\cdot\sum_{i=1}^{L}\log g_\phi(a_i|\boldsymbol{x}_i)\right] \tag{96}$$

$$= \nabla_\phi\mathbb{E}[\mathcal{L}_{\mathrm{KL}}] \tag{97}$$

**Remark. Why the absolute continuity** $p_{g_\phi}(\mathbf{x}_{0:L}|\mathbf{q}) \ll p_{g_{\phi_{\mathrm{old}}}}(\mathbf{x}_{0:L}|\mathbf{q})$ **holds?** Recall that we consider two parametrization of our unmasking policy model, $g_\phi$ and $g_{\phi_{\mathrm{Top\text{-}K}}}$ given as follows:

$$g_\phi(a^i\mid\boldsymbol{x}_n) = \frac{\exp\big([h_\phi(\boldsymbol{x}_n)]_{a^i}\big)}{\sum_{i'=1}^{n}\exp\big([h_\phi(\boldsymbol{x}_n)]_{a^{i'}}\big)}\text{ for all mask indices } a^i\in\mathcal{A}_{\mathbf{x}_n}, \tag{98}$$

Table 4: Comparison of different unmasking policies across benchmarks. For our method, the *dense reward* follows the definition in Section 4.1 and Appendix E. The *binary reward* assigns 1 only when (i) for SUDOKU, all masked positions are correctly predicted by LLADA, and (ii) for GSM8K, the final answer predicted by LLADA matches the ground truth.

| Benchmark | Random | Max-Confidence | Ours (Binary Reward) | Ours (Dense Reward) |
|-----------|--------|----------------|----------------------|---------------------|
| SUDOKU    | 0.616  | 0.705          | 0.801                | 0.817               |
| GSM8K     | 0.612  | 0.684          | 0.701                | 0.703               |

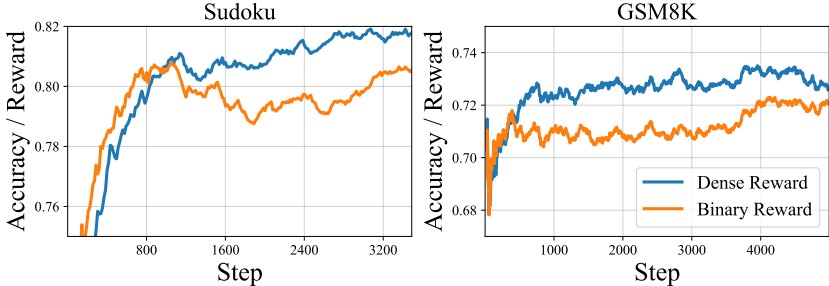

Figure 6: Average reward per training step when optimizing the unmasking policy using (i) dense reward and (ii) binary reward.

$$
g_{\phi_{\text{Top-K}}}(a^i \mid \mathbf{x}_n) =
\begin{cases}
\dfrac{\exp\big([h_\phi(\mathbf{x}_n)]_{a^i}\big)}{\sum_{a^j \in \operatorname{argtopk\,max}\pi_\theta(\cdot)} \exp\big([h_\phi(\mathbf{x}_n)]_{a^j}\big)} & \text{if } a^i \in \operatorname{argtopk\,max}\pi_\theta(\cdot), \\
0 & \text{otherwise,}
\end{cases}
\tag{99}
$$

where $\operatorname{argtopk\,max}\pi_\theta(\cdot)$ denotes the set of Top-$K$ mask indices corresponding to the highest-confidence predictions given by the MDM denoiser $\pi_\theta$. For the parametrization $g_\phi$, absolute continuity holds trivially. This is because $g_\phi$ assigns non-zero probability to every unmasking index, and $\pi_\theta$ also assigns non-zero probability to every token. Consequently, every sequence $\mathbf{x}_{0:L}$ receives strictly positive probability under $g_\phi$, i.e., $p_{g_\phi}(\mathbf{x}_{0:L}) > 0$ for all $\mathbf{x}_{0:L} \in \mathcal{X}_0 \times \mathcal{X}_1 \times \cdots \times \mathcal{X}_L$. Our remaining question is whether absolute continuity $p_{g_\phi}(\mathbf{x}_{0:L} \mid \mathbf{q}) \ll p_{g_{\phi_{\text{old}}}}(\mathbf{x}_{0:L} \mid \mathbf{q})$ also holds for $g_{\phi_{\text{Top-K}}}$. This condition is indeed satisfied: $p_{g_{\phi_{\text{Top-K}}}}(\mathbf{x}_{0:L} \mid \mathbf{q})$ and $p_{g_{\phi_{\text{Top-K,old}}}}(\mathbf{x}_{0:L} \mid \mathbf{q})$ are *equivalent measures*, that is, they are *mutually absolutely continuous* ($p_{g_{\phi_{\text{Top-K}}}} \ll p_{g_{\phi_{\text{Top-K,old}}}}$ and $p_{g_{\phi_{\text{Top-K,old}}}} \ll p_{g_{\phi_{\text{Top-K}}}}$). This is because the set of sequences $\mathbf{x}_{0:L}$ for which $p_{g_{\phi_{\text{Top-K}}}}(\mathbf{x}_{0:L}) > 0$ depends entirely on the denoiser $\pi_\theta$, and not on the particular choice of $g_{\phi_{\text{Top-K}}}$. Formally, recall that $p_{g_{\phi_{\text{Top-K}}}}$ can be written as:

$$
p_{g_{\phi_{\text{Top-K}}}}(\mathbf{x}_{0:L} \mid \mathbf{q}) = \prod_{n=1}^{L} g_{\phi_{\text{Top-K}}}(a_n \mid \mathbf{x}_n, \mathbf{q}) \cdot \pi_\theta(\mathbf{x}_{n-1} \mid \mathbf{x}_n, a_n, \mathbf{q}).
\tag{100}
$$

Here, since $\pi_\theta$ is strictly positive at every step, whether a given path $\mathbf{x}_{0:L}$ has zero probability is determined solely by $g_{\phi_{\text{Top-K}}}(a_n \mid \mathbf{x}_n, \mathbf{q})$. Inspecting Eq. (99), we see that $g_{\phi_{\text{Top-K}}}(a_n \mid \mathbf{x}_n, \mathbf{q})$ becomes zero exactly when $a_n \notin \operatorname{argtopk\,max}\pi_\theta(\cdot \mid \mathbf{x}_n, \mathbf{q})$. Thus, whether $p_{g_{\phi_{\text{Top-K}}}}(\mathbf{x}_{0:L} \mid \mathbf{q})$ is zero depends entirely on $\pi_\theta$ and is independent of the specific choice of $g_{\phi_{\text{Top-K}}}$. Consequently, $p_{g_{\phi_{\text{Top-K}}}}(\mathbf{x}_{0:L} \mid \mathbf{q})$ and $p_{g_{\phi_{\text{Top-K,old}}}}(\mathbf{x}_{0:L} \mid \mathbf{q})$ are equivalent measures, and hence $p_{g_{\phi_{\text{Top-K}}}}(\mathbf{x}_{0:L} \mid \mathbf{q}) \ll p_{g_{\phi_{\text{Top-K,old}}}}(\mathbf{x}_{0:L} \mid \mathbf{q})$ readily follows. $\qquad\square$

# D  ADDITIONAL EXPERIMENTS

## D.1  UNMASKING POLICY OPTIMIZATION IN BINARY REWARD SETTING

In the experiments reported in Section 4, we employed the dense reward defined in Section 4.1 and Appendix E. Specifically, the dense reward corresponds to the proportion of correctly predicted masked positions for SUDOKU, and for GSM8K, it consists of the binary correctness signal augmented with the sum of log-probabilities from $\pi_\theta(\cdot)$.

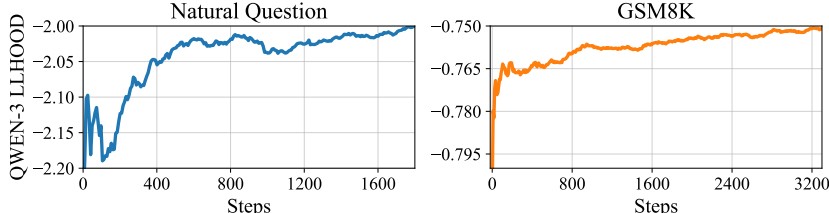

Figure 7: Training dynamics of UPO when using the average log-likelihood of Qwen-3 32B as the reward. The left plot corresponds to the NATURAL QUESTION benchmark, a general-purpose natural language generation dataset, and the right plot shows results on GSM8K. In both cases, the log-likelihood increases monotonically throughout training, indicating that the learned unmasking policy improves the generation quality.

In contrast, Theorem 1 and Theorem 2 theoretically show that under the **binary reward setting**, our KL-regularized unmasking policy optimization can outperform the reference policy. Therefore, in this section, we additionally evaluate our method under the *theoretically aligned* binary reward setting and demonstrate that our KL-regularized unmasking policy continues to surpass the reference policy.

The experimental results are presented in Table 4, and the training dynamics are shown in Figure 6. On SUDOKU, the dense reward yields slightly higher final accuracy and converges faster than the binary reward; however, in both SUDOKU and GSM8K, our method consistently outperforms the max-confidence baseline. These empirical observations are aligned with the theoretical predictions of Theorem 1 and Theorem 2.

### D.2 TOWARD SCALABLE AND GENERALIZABLE UNMASKING POLICIES

While the main experiments focus on correctness-based rewards within structured reasoning benchmarks, our next goal is to learn an unmasking policy that scales beyond task-specific supervision and remains useful across natural language domains. To this end, we conduct a preliminary study replacing the verifiable reward with a preference-based signal obtained from a stronger autoregressive model (ARM).

Specifically, we use the average log-likelihood of Qwen-3 32B (Yang et al., 2025) as the reward signal, keeping all training settings identical to the GSM8K setup except for the reward definition. We evaluate this configuration on two datasets: GSM8K, which enables comparison to correctness metrics, and NATURAL QUESTIONS (Kwiatkowski et al., 2019), a large-scale open-domain question–answering benchmark representing unconstrained natural language generation.

As shown in Figure 7, the learned policy consistently improves the reward across both datasets. Since the log-likelihood of a strong ARM is a standard proxy for generation quality, this trend indicates that the policy can adapt even without task-specific correctness signals. Moreover, on GSM8K, this preference reward also yields measurable downstream improvement, achieving 70.5% accuracy compared to 68.2% from the max-confidence baseline.

These findings provide early but encouraging evidence that unmasking policies may be learned under general preference signals rather than explicit oracle correctness. We view this as a step toward scalable, general-purpose token ordering strategies aligned with natural language structure rather than tied to individual benchmarks or reward designs.

### D.3 PASS AT N RESULT OF UNMASKING POLICY MODEL

In Section 3.1, we demonstrated that the Pass@N of $g_{\text{Top-K}}$ can grow significantly as $N$ increases, surpassing the single-trajectory accuracy of $g_{\text{conf}}$. This shows that performance can improve through the diversity induced by exploring multiple denoising paths. Our method can similarly benefit from both single-shot and multi-shot generation, which can be achieved by employing Gumbel-Softmax (Jang et al., 2017). When using Gumbel-Softmax, sampling is defined as:

$$a_n = \arg\max_{a^i \in \mathcal{A}_{\mathbf{x}_n}} \left[ g_{\phi_{\text{Top-K}}}(a^i \mid \mathbf{x}_n) + \epsilon_i \right], \qquad \epsilon_i \sim \mathcal{N}(0, \sigma^2),$$

Therefore, if one wishes to perform single-shot generation, setting $\sigma = 0$ yields a deterministic selection and achieves high accuracy. Conversely, increasing $\sigma$ enables multi-shot generation by injecting diversity into the sampled trajectories. In fact, when $\sigma$ is sufficiently large, the resulting behavior becomes theoretically identical to that of $g_{\text{Top-K}}$. We provide empirical evidence in Figure 8, where the Pass@N curves for $g_{\phi_{\text{Top-K}}}$ (Ours) and $g_{\text{Top-K}}$ match almost perfectly, confirming this correspondence.

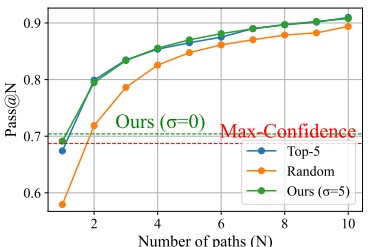

Figure 8: Pass@N on GSM8K. The dashed red line is the single-trajectory accuracy of max-confidence $g_{\text{conf}}$, and the dashed green line is the single-trajectory accuracy of our unmasking policy model where $\sigma = 0$ in Gumbel-Softmax.

## E EXPERIMENTAL DETAILS

### E.1 HYPER-PARAMETERS AND OTHER DETAILS

For all experiments, we set the group size $G$ as 6. We utilize AdamW optimizer with $\beta_1 = 0.9, \beta_2 = 0.99$ and weight decay is $0.1$. We perform gradient clipping where the maximum gradient norm is set as $0.2$. For computational efficiency, we utilized Flash Attention 2 and 4-bit quantization for MDM parameters. We set the gradient accumulation steps as 8. For SUDOKU and ZEBRA We utilize 3 40GB A100 GPUs for SUDOKU and ZEBRA and use a constant learning rate of $3 \times 10^{-6}$. We perform a gradient update 16 times for each prompt. Since MATH500 and GSM8K are much harder to learn, we use more computation resources and set stable hyperparameters. We utilize 8 40GB A100 GPUs and use cosine learning rate scheduler starting from $2 \times 10^{-6}$ and ends at $10,000$ updates. We perform a gradient update 8 times for each prompt. Following Diffu-GRPO (Zhao et al., 2025), we save models for every 100 steps and report the best accuracy. For all experiments, we set $\beta = 10^{-4}$. For softmax realizations we set $\tau = 0.05$ for $g_{\text{conf}\tau}$. Since ZEBRA benchmark is a symbolic dataset that differs from human language, we performed supervised finetuning[1].

### E.2 REWARD AND PROMPT TEMPLATE FOR EACH BENCHMARK

For SUDOKU and ZEBRA, we set the reward function as the matching rate between the generated sequence and the answer. For example, if there are 8 masks in a sudoku puzzle and the model got 5 correct answers, then the reward is $5/8 = 0.625$. For MATH500 and GSM8K, the reward is 1 if the model got the correct answer and 0 otherwise. We do not use formatting rewards such as other GRPO papers since unmasking policy models do not have the ability to change the token probability distribution itself. However, for MATH500 and GSM8K, the reward is too sparse, as it only gives 0 or 1. Therefore, we utilize additional reward $\sum_{i=1}^{L} \log \pi_\theta(\boldsymbol{x}_{i-1} | \boldsymbol{x}_i, a_i)$. We do not utilize it directly, but we assigned relative reward, e.g., 0.25 for the trajectory with the largest probability, 0 for the trajectory with the smallest probability. We have confirmed that trajectories with higher probability are more likely to get a correct answer, and this reward indeed stabilized the unmasking policy model training process. Similarly, large-scale MDM such as Seed diffusion (Song et al., 2025) post-trains the MDM to fit to the generated sequence with a higher ELBO.

We provide a prompt template for SUDOKU, MATH500, and GSM8K below, and ZEBRA is omitted since it is a symbolic dataset.

---

[1]We have utilized GitHub code from (Zhao et al., 2025).

Prompt template used for SUDOKU

```
Please solve the following 4x4 Sudoku puzzle. The puzzle is provided as a
    16-character string reading left-to-right, top-to-bottom, where ' '
    represents empty cells.

Rules:
- Fill empty cells with digits 1-4
- Each row must contain digits 1-4 exactly once
- Each column must contain digits 1-4 exactly once
- Each 2x2 box must contain digits 1-4 exactly once

Important: Your solution must be a COMPLETE 16-character string with only
    the digits 1-4, representing your final solved grid. Never leave it
    as ' '.

Respond in this exact format:
</reasoning>
<answer>
[First row of 4-character solution]
[Second row of 4-character solution]
[Third row of 4-character solution]
[Fourth row of 4-character solution]
</answer>

Solve the following Sudoku puzzle:
 1 3   2
    2
 2 4   3
 3
<answer>
 1 3 [M] 2
[M]2[M][M]
 2 4 [M] 3
 3[M][M][M]
  </answer>
```

Prompt template used for GSM8K and MATH500

```
You are a math expert. You will be given a question to solve. Solve it
    step by step. Wrap the final answer in a \\boxed{}.
Respond in the following format:
<reasoning>
Your reasoning here
</reasoning>
<answer>
\\boxed{...}
</answer>
<reasoning>
... 128 Masks
```

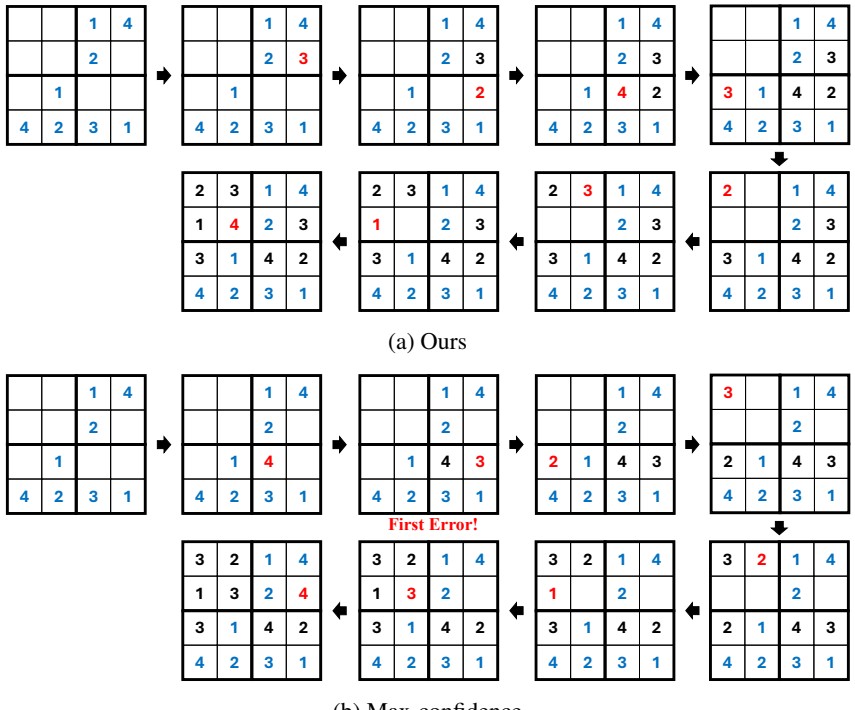

(a) Ours

(b) Max-confidence

Figure 9: (a) Unmasking path of our unmasking policy model and (b) unmasking path of the max-confidence baseline on the same 4×4 Sudoku instance. Blue digits indicate the original clues in the puzzle, red digits denote the value unmasked at the current step by the unmasking policy and sampled by the LLADA, and black digits represent previously filled values. Our model successfully solves the entire puzzle, whereas the max-confidence method begins to make mistakes from the point marked "First error!".

## F   SUDOKU UNMASKING EXAMPLES

To better illustrate the qualitative behavior of our learned unmasking policy, we visualize representative unmasking trajectories from the main SUDOKU experiment. In all examples, the underlying MDM samples the token values via argmax over the model's categorical distribution, so the only difference between methods lies in the choice of the unmasking position. As shown in Figures 9 and 10, our learned policy tends to prioritize positions whose values are already strongly determined by the puzzle constraints, leading to stable infilling and consistent recovery of the correct solution.

In contrast, the max-confidence heuristic shows two clear failure behaviors. In Figure 9, it chooses a reasonable unmasking position, but the predicted value is wrong. This single mistake immediately propagates and breaks the remaining steps. In Figure 10, it selects a position whose correct value is not yet identifiable. The predicted token does not violate any explicit Sudoku rule, but it places the puzzle in a state that cannot lead to a valid solution, highlighted by "Bump!". The next prediction therefore fails.

These visualizations highlight a central claim of our work: the choice of where to unmask is as crucial as the values predicted by the MDM, and a learned policy can more reliably identify structurally deterministic positions than hand-crafted schedules.

## G   USAGE OF LARGE LANGUAGE MODEL

We have utilized a large language model to assist with grammar and wording for writing the paper.

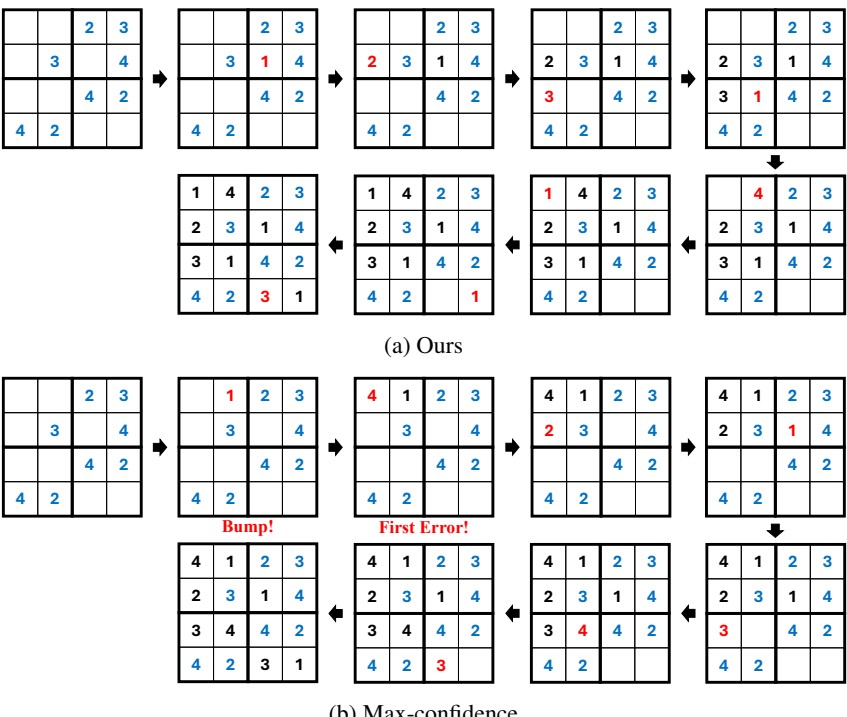

(a) Ours

(b) Max-confidence

Figure 10: (a) Unmasking path of our unmasking policy model and (b) unmasking path of the max-confidence baseline on the same 4×4 Sudoku instance. Blue digits indicate the original clues in the puzzle, red digits denote the value unmasked at the current step by the unmasking policy and sampled by the LLADA, and black digits represent previously filled values. Our model successfully solves the entire puzzle, whereas the max-confidence method reaches a point labeled "Bump!" where the unmasking decision does not immediately violate Sudoku constraints but nonetheless leads to an unsatisfiable partial state. Consequently, the very next prediction results in an error.

