# OpenReview forum: "Improving Discrete Diffusion Unmasking Policies Beyond Explicit Reference Policies"
_ICLR.cc/2026/Conference — ICLR 2026 Poster_

### Official Review · Reviewer_9cnV · 2025-10-21

**Soundness:** 2
**Presentation:** 2
**Contribution:** 3
**Rating:** 4
**Confidence:** 4

**Summary:**

This paper presents an algorithm to train a unmasking-position policy for masked diffusion models. While rule-based unmasking policies based on confidence or probability margin are popular,  the authors propose reinforcement learning (more concretely, GRPO) with KL-divergence regularization starting from such rule-based reference policies. The proposed algorithms have theoretical guarantees under technical assumptions, and show practical improvements on puzzle and math benchmarks.

**Strengths:**

Adding KL-divergence regularization in the RL of unmasking policy yields theoretical tractability and improvement in ablations. It also proposes a KL-divergence inequality regarding among the data distribution, reference policy, and optimized policy, which can be more informative than the reward inequality.

**Weaknesses:**

The weakness of the paper primarily lies in the mismatch of theory and practice.
- [W1] While the analyzed algorithm in Section 3.2-3.3 is based on 0-1 reward, it is totally different from what is described in Section D.2; the hitting rate in puzzle benchmarks, and more heuristic additional dense reward in math reasoning benchmarks. The readers might think that the practical performance is just based on the heuristic choise of rewards rather than theoretical insights.
- [W2] Theorem 2 suggests the KL divergence is closer (or equal to) the reference policy's KL divergence, but I feel that it only reflects the reward/rate of correct answer. For example, if there are multiple answers in each query, then there is a case where $L_{\text{MDM}}>0$ but $r_{\text{ref}}=1$, with the model only outputting a small subset of possible answers. In such a case, the policy is already converged, and the KL divergence is the same. Though this example does not violate the inequality (as it is not a strict inequality), the message after Theorem 2 (L281) "Theorem 2 guarantees the closer sampling to $p_{\text{data}}$ or ideal MDM $θ^*$ than $g_{\text{ref}}$" is somewhat misleading to me.
- [W3] In Theorem 2, the assumption $\text{supp}(p_{\theta*}(\cdot|q))\subseteq \text{supp}(p_{g\text{ref}}(\cdot|q))$ looks too strong, especially when considering the deterministic nature of the "confidence" unmasking policy.
- [W4] While $g_{\text{conf}}$ is deterministic, since you have a sampling randomness with $\pi_\theta$ in addition to the positional choice $g$, the sampling results should still be random. I am not sure why Max-Confidence is treated as if a constant in Figure 1, which might be misleading.
- [W5] The proofs are not written in a careful way. For example:
    - [W5-1] At least in Equations 30, 31, 36, 38, 39, 71, and 73, the parentheses are not appropriately closed. Some are just trivial mistakes but some make the equations ambiguous. At this rate of obvious mistakes left, I'm not sure if the draft is appropriately checked by the authors themselves.
    - [W5-2] In Eq. 82, why is it a strict inequality rather than "$\leq$"?
    - [W5-3] The equality from Eq. 92 to Eq. 93 is valid only when $p_{g\phi}$ is abosolute continuous w.r.t. $p_{g\phi\text{old}}$. Is it an additional assumption?

While the proposed method consitently improves upon existing methods, the current manuscript has a large room for improvement as discussed above, which makes me lean to rejection.

Typos:
- L95: $\sum T_{x'\in X}$ -> $\sum_{x'\in X}T$
- L888: {1, L} -> {1, ..., L}?
- L937: Let let -> Let

**Questions:**

In addition to the points raised in the Weaknesses section, I have the following question:
- [Q1] As the authors write in +, Huang et al. (2025) applies RL jointly to the masked diffusion model and unmasking policy so the existing method is heavy. Doesn't it mean that, by fixing the diffusion model, the existing method can train only the policy model for lighter computational cost? Can the authors discuss a bit more on the difference and the preposed method's original contribution? (I assume it is basically about initialization and KL regularization though.)

---

> ### Author Response · Authors · 2025-11-22
>
> We sincerely thank the reviewer for engaging so deeply with the theoretical aspects of our work. The reviewer’s careful reading has brought up points that we had not fully considered, and it has helped improve the paper. The concerns raised are all reasonable, and we show that they can be resolved without major changes to our original arguments. In the revised version of the paper, we have added the missing experiments, clarified assumptions, and refined several proofs accordingly.
>
> >### **W1. The theory assumes a binary reward, whereas the experiments use dense rewards, making it appear that the theoretical analysis and practical implementation are misaligned.**
>
> Thank you for pointing this out. To directly evaluate the alignment between our theoretical setting and practical performance, we repeated the experiments in Section 4.3 using strict binary rewards, exactly matching the assumption in our analysis. More specifically, we set reward = 1 only when the entire grid is correct for SUDOKU, and reward = 1 only when the final answer is correct for GSM8K.
>
> | Benchmark | Random | Max-Confidence | Ours (Binary Reward) | Ours (Dense Reward in Paper) |
> |-----------|--------|-----------|-----------------------|----------------------|
> | SUDOKU    | 0.616  | 0.705          | 0.801                 | 0.817                |
> | GSM8K     | 0.612  | 0.684          | 0.701                 | 0.703                |
>
> Even under this binary setting, our learned policy still outperforms the max-confidence reference, confirming that the theoretical analysis and its most faithful instantiation are indeed aligned. We include the results and reward curves in Appendix D.1 of the revised version.
> We appreciate the reviewer’s suggestion, which helped us strengthen the connection between theory and experiment. Finally, while dense rewards yield slightly higher accuracy, we would be grateful if the reviewer could view this as an indication that our work not only provides a solid theoretical framework but also offers a practical, real-world applicable heuristic.
>
> >### **W2. Concerns about Theorem 2**
>
> Thank you for raising this insightful point. Since the question involves several concerns, we address them one by one below.
>
> **W2-1. A counterexample exists when** $r_{g_{\text{ref}}}=1​$
>
> First, there is an implicit assumption that was not explicitly stated in the original version: Theorem 2 inherits the same conditions as Theorem 1, since it is derived directly from the convergence property specified in Eq. (13) of Theorem 1 (it is noted in Theorem 2). Meanwhile, we acknowledge that this might confuse readers; therefore, we restate the assumption below:
>
> **Condition\***: Assume we select well-defined reward $r(\mathbf{q},\mathbf{a})\in\{0,1\}$ such that $p_{\text{data}}(\mathbf{q},\mathbf{a})>0\Rightarrow r(\mathbf{q},\mathbf{a})=1$ and $\underbrace{0<r_{g_{\text{ref}}}<1}_{\text{abbreviated condition}}$
>
> for $p_{\text{data}}$ where data is composed of prompt and oracle answer $\{\mathbf{q},\mathbf{a}\}​$. This condition is realistic for GRPO-style tasks, where correctness-based binary rewards naturally satisfy it.
>
> Under **Condition\***, the counterexample raised by the reviewer does not occur. We have explicitly added this condition to Theorem 2 in the revised paper.
>
> **W2-2, The theorem states a non-strict inequality but claims guaranteed “closer” sampling.**
>
> Furthermore, with this correction, we slightly refined the proof and strengthened the statement to a strict inequality, ensuring that the learned sampler is strictly closer to $p_{\text{data}}(\mathbf{x})$ than the reference policy. We updated the proof in the revised manuscript.
>
> **W2-3. The KL term may reflect only the reward (rate of correct answers), not actual closeness to $p_{\text{data}}$.**
>
> We interpret the reviewer’s question as asking whether the KL tightening in Theorem 2 truly reflects closeness to $p_{\text{data}}$, or merely an increase in reward (rate of correct answer). This is an excellent conceptual point, and our view is that both interpretations are valid. A complementary way to understand Theorem 2 is the following:
> - **Any reward function that satisfies** **Condition\***—whether it evaluates final correctness, the absence of reasoning errors, grammatical validity, or even a combination of these—naturally induces learning dynamics that move the sampling distribution closer to $p_{\text{data}}$.
>
> Consequently, one may view any reward that satisfies **Condition\*** as a “good” reward in the sense that it provably induces learning dynamics that bring the sampling distribution closer to pdata. The designer may choose which aspect of the data distribution to tighten the KL divergence toward by selecting an appropriate reward, as long as the reward satisfies **Condition\***. For GRPO tasks, answer correctness is the primary objective, so we adopt a correctness-based reward. We have added this interpretation to Section 3.3 in the revised version of the paper.

---

> ### Author Response · Authors · 2025-11-22
>
> >### **W3. The Support assumption in Theorem 2 looks too strong.**
>
> We respectfully disagree that the support assumption in Theorem 2 is too strong. For any deterministic $g_{\text{ref}}$ and any terminal sequence $\mathbf{x_0} \in \mathcal{V}^L$, the MDM denoiser $\pi_\theta(\cdot \mid \mathbf{x_n})$ assigns non-zero probability to every token in the vocabulary. Therefore, regardless of how $g_{\text{ref}}$ selects the unmasking positions $a_n$, there always exists at least one valid path that leads to $\mathbf{x_0}$, implying $p_{g_{\text{ref}}}(\mathbf{x_0} \mid \mathbf{q}) > 0$. By definition, the terminal-output probability is the marginalization over all possible denoising trajectories:
>
> $$p_{g_{\text{ref}}}(\mathbf{x_0} \mid \mathbf{q})= \sum_{\mathbf{x_1},\dots,\mathbf{x_L}}p_{g_{\text{ref}}}(\mathbf{x_{0:L}} \mid \mathbf{q})= \sum_{\mathbf{x_{1:L}},\, a_{1:L}}\prod_{n=1}^Lg_{\text{ref}}(a_n \mid \mathbf{x_n},\mathbf{q})\cdot\pi_\theta(\mathbf{x_{n-1}} \mid \mathbf{x_n}, a_n, \mathbf{q}).$$
>
> Because $\pi_\theta$ has full support, the product above is strictly positive for at least one trajectory. To illustrate, take any target sequence $\mathbf{x_0^\ast}$. Starting from the fully masked state, $g_{\mathrm{ref}}$ selects an index $a_L$; since $\pi_\theta(x^{a_L}=x_0^{\ast,a_L} \mid \mathbf{x_L})>0$, the next state $\mathbf{x_{L-1}}$ satisfying $x_{L-1}^{a_L}=x_0^{*,a_L}$ is reachable. Repeating this inductively shows that $\mathbf{x_0^\ast}$ is reachable regardless of $g_{\mathrm{ref}}$. Thus, the support assumption holds.
>
> Because this point may be subtle to readers, we have included a detailed explanation as a **Remark** in the appendix of the revised paper.
>
> >### **W4: Why is the sampling result for max-confidence shown as a constant in Figure 1, even though $g_{\text{conf}}$ is deterministic but $\pi_\theta$ is stochastic?**
>
> Thank you for pointing this out. The figure may have caused a misunderstanding. For controlled comparison to analyze the effect of generation ordering and following the default setting used in LLaDA, we conducted sampling using greedy decoding, i.e., selecting $\arg\max_{x}\pi_\theta(\cdot|\mathbf{x_n},a_n,\mathbf{q})$ at each step. Under this setting, the stochasticity of $\pi_\theta$ does not manifest in the final outputs, which is why the max-confidence appear constant. This experimental choice is intended since we want to see how Pass@N increases purely from using a better sampling order. To avoid confusion, we have added an explicit explanation of this decoding choice in the main paper.
>
> >### **W5. Proof minor concerns**
>
> **W5-1. Parentheses and notation are unclear in some parts of the proof**
>
> We have fixed the unclear or missing parentheses and notational issues in the revised version. Specifically, in Theorem 2, parentheses are resized for clearer hierarchy, and in Proposition 1, parentheses are visually grouped using color for readability.
>
> **W5-2. The statement uses “strict inequality,” but the proof uses a non-strict inequality.**
>
> Thank you for pointing this out. This was a typo in the statement, and we have corrected the statement to non-strict inequality ($\le$).
>
> **W5-3. Is an absolute continuity assumption required when going from Eq. 92 → 93?**
>
> We agree that absolute continuity is required when going from Eq. 92 → 93. However, this is not an assumption, and we can show $p_{g_{\phi}}$ is structurally absolutely continuous w.r.t. $p_{g_{\phi_{\text{old}}}}$. For short, this is because the existence of a specific path depends on MDM, which is shared by $p_{g_{\phi}}$ and $p_{g_{\phi_{\text{old}}}}$, not by $\phi$ and $\phi_{\text{old}}$ themselves. Therefore $p_{g_{\phi}}$ is absolutely continuous w.r.t. $p_{g_{\phi_{\text{old}}}}$, and we add a complete explanation in the revised paper (in the proof of Proposition 3) clarifying why it holds in our setting.

---

> ### Author Response · Authors · 2025-11-22
>
> >### **Q1. Couldn’t DCoLT simply freeze the backbone as well? Please clarify more precisely how your method differs from DCoLT.**
>
> First of all, the reviewer’s interpretation of the relationship between our method and DCoLT [1] is correct. To clarify the differences more explicitly:
> - At a high level, both methods share similar objectives, but we additionally introduce the initialization strategy, KL-based regularization, and Top-K realization.
> - DCoLT offers strong engineering insights and demonstrates the **practical potential** of unmasking policy models through large-scale training, whereas our work aims to provide the **theoretical foundations** explaining why a learned unmasking policy can outperform fixed heuristics such as max-confidence, why token-level reward optimization is valid, and how we can gain improvements by learning only the policy (e.g., with Top-K).
>
> Furthermore, we have already included the experimental analysis. While DCoLT jointly trains the LLaDA model in their main experiments, it is of course possible to freeze the backbone and train only the unmasking policy. Then, DCoLT corresponds to one of the options we analyze, training $\mathcal{L_{\text{UPO}}}(g_\phi,\varnothing,\varnothing)$ (initial submission: lines 414–416, revised paper: lines 426-427). We evaluated this setting in Figure 3 and discussed in Section 4.2 that our method achieves stronger performance than DCoLT under backbone-freezing.
>
> [1] Huang et al., “Reinforcing the Diffusion Chain of Lateral Thought with Diffusion Language Models,” NeurIPs’25
>
> We hope our answer clearly addresses the reviewer’s question, and if the reviewer has further questions, we are happy to answer them.

---

> ### Comment · Reviewer_9cnV · 2025-11-22
>
> Thank you for a timely rebuttal. Basically all my concerns got resolved. My thoughts after reading the rebuttal and the revised paper as follows:
> - It is good to confirm that the sparse reward, which is what actually the theory considers, works in practice with SUDOKU etc.
> - The added explicit assumtion on the reward and the "interpretation" after theorem 2 greatly helps understanding. Also, the author's answer to W2-3 that there is a degree of freedom in the choice of reward (that you can choose a "weaker" reward as long as it satisfies the condition) is reasonable.
> - Finally, I apologize for my initial misunderstanding on W3: I guess I had been confused around the determintic sampling question in W4 when thinking about the assumption.
>
> I believe the revised draft is substaintially better than the initial submission and worth acceptance. Let me raise my score to 8.

---

> > ### Author Response · Authors · 2025-11-22
> >
> > Thank you for the thoughtful follow-up and for updating the score.
> >
> > We truly appreciate the time you invested in carefully considering the theoretical components of our work. The detailed feedback helped us substantially improve both the clarity and rigor of the paper. We are glad to hear that the revised version addressed your concerns.

---

### Official Review · Reviewer_BVXC · 2025-10-31

**Soundness:** 4
**Presentation:** 4
**Contribution:** 4
**Rating:** 8
**Confidence:** 5

**Summary:**

This paper explores learning the unmasking order in MDMs. People have realized that a pretrained MDM has an any-order sampling property, which led them to explore heuristic unmasking policies. These use the information of per-index logits, so they don't fully leverage the information and also rely on heuristics. This work focuses on learning the unmasking policy in fine-tuning the LLaDA using RL methods, achieving better performance than baselines.

**Strengths:**

First of all, this paper is clearly well-written. The authors explain the background clearly and then the problem that they want to solve in the paper. Also, examples in Sections 2 and 3 assist the reader in grasping the context better. Moreover, the experiment design (loss design, architectural modification) is really clear to follow. I believe this paper takes a step toward finding an effective unmasking strategy beyond prior heuristic strategies.

**Weaknesses:**

Perhaps the most significant weakness of this paper is the less-than-ideal experimental results: We can see that on several tasks, the performance of the trained model's sampling strategy is comparable to the Margin sampling strategy from (Kim et al, 2025). Given that the method proposed in the paper requires additional training (while the Margin strategy is training-free), it might be questionable to some people that if we really need an approach that requires training.

**Questions:**

- In Figure 1, we can see that the Pass@N performance can actually go far beyond the Max-Confidence baseline, indicating that there's indeed a decent sampling path (that has a much higher accuracy even than Max-Confidence). The results in the paper, however, aren't that significant. I'm curious how the authors could discuss the discrepancy, i.e., what was the main factor that reduced this expected huge gap?

- I wondered what the authors meant by `Meanwhile, on GSM8K, the much larger search space makes Top-K preferable for stability. In future work, we plan to develop an unmasking policy optimization algorithm that explores the full mask tokens even when the search
space is large.'

---

> ### Author Response · Authors · 2025-11-22
>
> We sincerely appreciate the reviewer’s positive assessment of our work and the valuable feedback that strengthens our paper. The segments of the revised paper that reflect the received feedback are marked in red.
>
> >### **W1. The trained policy does not superiorly outperform rule-based heuristics, especially margin-based methods.**
>
> We first note that our work aims to prove and empirically demonstrate that our method can improve performance over the chosen reference policy. Indeed, compared to max-confidence, our method achieves improvements of +11.2% on SUDOKU, +2.5% on Zebra, +1.9% on GSM8K, and +1.2% on Math500.
>
> Nevertheless, as the reviewer noted, our unmasking policy does not achieve dramatically higher performance than rule-based heuristics such as margin-based methods on GSM8K and Math500. We would like to offer our thoughts on why this occurs. The major reason we speculate is the difference in token space and dataset size for GSM8K and Math500. Our unmasking policy model takes two types of inputs:
> 1. The Top-K probability at each masked position from the base LLaDA. This allows the model to imitate the behavior of heuristics such as max-confidence and to learn entropy-style information.
> 2. The last-layer transformer feature of LLaDA. From this, the model can learn to generate important words first based on the contextual understanding captured by the base model.
>
> For feature type (1), both SUDOKU and GSM8K involve the same dimensional structure, so there is no substantial difficulty. However, for feature type (2), SUDOKU only requires modeling digits, whereas language generation tasks like GSM8K require learning over LLaDA’s full vocabulary (124K tokens). Meanwhile, GSM8K has only 7.4K training examples. Because the dataset is small relative to the size of the token space, it is difficult for the unmasking policy to reach its optimal performance under this setting. We have added this discussion to Section 5 in the revised paper.
>
> On the other hand, we provide a future direction that may address this issue through W1 comments of the reviewer `cCbk`. We kindly ask the reviewer to refer to our discussion of this direction.
>
> >### **Q1: In Figure 1, Pass@N shows a much higher accuracy than Max-Confidence. Why doesn’t the proposed method achieve a similarly large improvement?**
>
> First, the reasons stated in our response to W1 (e.g., dataset limitations) as well as the discussion in Q2 (training instability) partially address this question. Here, we provide additional, more fundamental explanations.
>
> There are two key points:
> 1. The fact that Pass@N surpasses max-confidence indicates that there indeed exist single-shot unmasking paths that outperform the max-confidence trajectory. However, this does not imply that such an optimal path is necessarily *learnable* by a parametric unmasking policy.
> 2. More precisely, Pass@N represents an upper bound on what any single-shot unmasking policy can achieve. In particular, $\max_{\phi} r_{g_{\phi}}\le \lim_{N\rightarrow\infty}\text{Pass@N}(g_{\text{rand}})$ holds.
>
> Whether this optimal path is learnable depends on the amount of statistical regularity shared across the high-reward trajectories. For instance, in tasks such as SUDOKU or the Zebra puzzle, the optimal paths might share strong structural patterns, e.g., certain blanks must be filled early, and these patterns seem learnable. In contrast, for GSM8K, it is an empirical question whether such regularity exists to a degree that allows a learnable policy to match the upper envelope suggested by Pass@N.
>
> Consequently, the preliminary experiment and the main experiment are not in conflict; they play complementary roles. The preliminary experiment shows that high-yield trajectories exist and that their upper bound can be far above max-confidence, thus motivating the search for a learnable policy. The main experiment then reveals that the unmasking policy is indeed learnable.

---

> ### Author Response · Authors · 2025-11-22
>
> >### **Q2. Please clarify what the authors meant in the Conclusion when stating:** *“Meanwhile, on GSM8K, the much larger search space…”*
>
> In GSM8K, we observed that the Top-K realization ($g_{\phi_{\text{Top-K}}}$), which restricts actions to the Top-K mask positions suggested by the MDM confidence, outperformed both the max-confidence realization and the full softmax realization ($g_{\phi}$). Nevertheless, from a modeling perspective, we believe that a more expressive parameterization such as $g_{\phi}$ **should eventually outperform** $g_{\phi_{\text{Top-K}}}$ because its action space strictly contains that of Top-K. In general MDPs, a larger action space can achieve a higher optimal value, since a policy with restricted actions may never reach certain optimal trajectories.
>
> However, increasing the action space also increases policy gradient variance, making training significantly less stable. The fact that $g_{\phi_{\text{Top-K}}}$ performs better on GSM8K suggests that there likely exists a more effective practical optimization method for large action spaces. In contrast, SUDOKU has a much smaller action space (shorter generation length), and in this setting $g_{\phi}$ indeed outperforms $g_{\phi_{\text{Top-K}}}$.
>
> Therefore, our statement in the Conclusion refers to the following idea: for tasks with very large action spaces such as GSM8K, we expect that future work can develop a more stable optimization algorithm that enables the full-action policy $g_{\phi}$ to realize its theoretically superior performance. This is the research direction we intended to highlight. We have consolidated this point and incorporated the refined version into the conclusion.
>
> We hope our answer clearly addresses the reviewer’s question, and if the reviewer has further questions, we are happy to answer them.

---

### Official Review · Reviewer_cCbK · 2025-11-01

**Soundness:** 4
**Presentation:** 3
**Contribution:** 1
**Rating:** 4
**Confidence:** 3

**Summary:**

This paper proposes a method to improve the token generation order of pretrained Masked Diffusion Models (MDMs) through RL. Specifically, the authors train a learned unmasking scheduler using GRPO. Compared to heuristic orderings such as confidence-based or margin-based selection, the proposed approach achieves better performance across several benchmarks.

**Strengths:**

**Novelty:**
The idea of learning the generation order of a non-autoregressive model through reinforcement learning has been previously explored in classical non-autoregressive language models (see [1]). However, applying this idea to **modern diffusion-based generation models** represents an original and meaningful contribution.

**[1]** Insertion-based Decoding with Automatically Inferred Generation Order, [arXiv:1902.01370](https://arxiv.org/abs/1902.01370)

**Weaknesses:**

**Generalizability of the Method:**
Since the optimization is based on RLVR (GRPO), the method can only be applied when a **verifiable reward signal** is available. This limits its general applicability to domains where rewards can be explicitly computed.

**Suggestion:**
Could the authors explore the use of a **preference reward model** to enable training in more general settings? Alternatively, using the **generative perplexity** of a strong language model such as GPT-2 Large as a reward could be an interesting direction for learning generation order without explicit ground-truth supervision.

**Incremental Performance Gain:**
Despite requiring additional RL-based training, the performance improvements over strong baselines appear modest—except for the SUDOKU task, where the generation order plays a particularly critical role. The paper would benefit from discussing why the proposed method does not yield stronger improvements on other benchmarks.

**Questions:**

* **Pass@K:**
  Could the authors report *pass@K* results? It would be informative to know whether the proposed method sacrifices diversity (e.g., generating overly similar samples) in pursuit of higher accuracy.

* **NFE (Number of Function Evaluations):**
  How many diffusion reverse steps are used during inference? Please specify the number of denoising iterations to facilitate fair comparison with other methods.

* **Qualitative Analysis:**
  It would be helpful to visualize the learned unmasking order on SUDOKU and compare it to existing baselines such as max-margin or max-confidence. A qualitative comparison could clarify what the learned policy actually discovers beyond heuristics.

---

> ### Author Response · Authors · 2025-11-22
>
> We thank the reviewer for the interest in our work and the valuable feedback that strengthens our paper. The segments of the revised paper that reflect the received feedback are marked in red.
>
> >### **W1. Please check whether the proposed method can generalize to preference reward models (e.g., GPT-2 Large perplexity).**
>
> Thank you for the thoughtful suggestion. We first note that using GPT-2 Large perplexity as a reward model for LLaDA is not an ideal experimental direction. GPT-2 Large is a 0.8B model released in 2019, and even GPT-2 XL has only 1.5B parameters. In contrast, LLaDA is an 8B model released in 2025, with performance comparable to 8B-scale autoregressive models such as LLaMA-3 (2024). Using a much weaker model to guide an 8B MDM is conceptually inappropriate.
>
> Instead, we followed the reviewer’s intention: testing compatibility with a preference-style reward from a stronger ARM, thus conducted experiments using Qwen-3 32B (2025). Specifically, we used its average log-likelihood (conceptually equivalent to negative log-perplexity) as the reward signal. The corresponding reward curves over training are included in Appendix D.2 of the revised paper. We observe that the log-likelihood consistently increases as training progresses, meaning that the perplexity decreases, indicating that our unmasking policy indeed adapts to this preference-based signal. Furthermore, **even trained with log-likelihood of Qwen-3 32B, the accuracy on GSM8K has increased to 70.5% where max-confidence gives 68.2%.**
>
> Although our main results are built around GRPO-style correctness rewards, we agree that exploring more general preference reward models, especially those derived from large ARMs, could be an exciting extension of our framework. We appreciate the reviewer’s suggestion and view this as a promising direction for future work. However, we kindly ask the reviewer to view this direction as a potential avenue for future work rather than a shortcoming of our current experimental design. Qwen-3 32B itself requires around 80GB of VRAM, making training substantially slower, and such a large oracle model might be unavailable in realistic settings. We therefore hope the reviewer to consider that our method demonstrates performance improvements using only verifiable (correctness-based) rewards, even without relying on a large external model.
>
> >### **W2. The performance improvements on math benchmarks appear relatively modest. Providing more insight into why this happens would make the paper more concrete.**
>
> Thank you for pointing out this important aspect. Basically, it stems from the fact that generation order plays a much more important role in SUDOKU, as the reviewer pointed out, but another major reason is the difference in token space and dataset size for GSM8K. Our unmasking policy model takes two types of features as input:
>
> 1. The Top-K probability at each masked position from the base LLaDA. This allows the model to imitate the behavior of heuristics such as max-confidence and to learn entropy-style information.
> 2. The last-layer transformer feature of LLaDA. From this, the model can learn to generate important words first based on the contextual understanding captured by the base model.
>
> For feature type (1), both SUDOKU and GSM8K involve the same dimensional structure, so there is no substantial difficulty. However, for feature type (2), SUDOKU only requires modeling digits, whereas language generation tasks like GSM8K require learning over LLaDA’s full vocabulary (124K tokens). Meanwhile, GSM8K has only 7.4K training examples. Because the dataset is small relative to the size of the token space, it is difficult for the unmasking policy to reach its optimal performance under this setting. We have added this discussion to Section 5 in the revised paper.
>
> One may ask why GRPO-style methods typically achieve large gains on mathematical benchmarks such as GSM8K. There are two reasons. First, standard GRPO methods fine-tune the entire LLaDA model, whereas we keep the backbone fixed and train far fewer parameters. Second, improvements from backbone-finetuning GRPO approaches on GSM8K often benefit from formatting. Accuracy in mathematical reasoning benchmarks is usually computed by parsing the number inside a “\boxed{…}” or inside “<answer>…</answer>” and checking whether it matches the ground truth. GRPO methods such as d1 [1] or wd1 [2] add rewards for producing tokens like “\boxed” or “<answer>”, so as to increase the probability of such tokens being generated. Our method cannot increase accuracy through such formatting cues, as our method does not explicitly change the prediction of the base model. Even though we acknowledge that formatting is important, we do not consider this a critical weakness of our approach since it slightly departs from the genuine ability of language models.
>
> (continued in next comment)

---

> ### Author Response · Authors · 2025-11-22
>
> For these reasons, the direction suggested in W1 appears to be an important future research direction. For example, rather than relying on GSM8K alone, one could gather a wide variety of QA benchmarks and train the unmasking policy using preference rewards from stronger models. This would remove limitations due to dataset size and eliminate the need to construct explicit verifiable rewards by hand. While our experiments with respect to W1 already hint at this possibility, we intend to pursue it as a standalone research project. For the present paper, we hope the reviewer sees that even training only the unmasking policy with GRPO for a specific task can already improve accuracy, and that our theory supports the potential of this approach.
>
> [1] Zhao et al., “d1: Scaling Reasoning in Diffusion Large Language Models via Reinforcement Learning,” NeurIPS’25
>
> [2] Tang et al., “wd1: Weighted Policy Optimization for Reasoning in Diffusion Language Models”, Arxiv’25
>
> >### **Q1. Please report the Pass@N performance of the proposed method.**
>
> In response to the reviewer’s request, we measured Pass@N on GSM8K. An additional strength of our method is that we can control diversity during sampling by applying Gumbel-Softmax to the unmasking policy. The noise scale $\sigma$ interpolates between deterministic, accuracy-driven sampling ($\sigma=0$) and diverse multi-path sampling ($\sigma>0$). With sufficiently large noise, the behavior becomes effectively identical to the Top-K heuristic reported in our preliminary experiments. Below, we report the Pass@N evaluation of our unmasking policy on GSM8K. Full Pass@N curves and additional explanations are provided in Appendix D.3.
>
> | | Acc. | Pass@2 | Pass@3 | Pass@4 | Pass@5 |
> | --- | --- | --- | --- | --- | --- |
> | Random | 0.612 | 0.718 | 0.786 | 0.826 | 0.848 |
> | Top-K | 0.679 | 0.799 | 0.834 | 0.853 | 0.865 |
> | Ours (T=0) | 0.702 | - | - | - | - |
> | Ours (T=5) | 0.691 | 0.795 | 0.834 | 0.855 | 0.870 |
>
> Our method with $\sigma=0$ achieves the highest single-trajectory accuracy. When diversity is enabled through $\sigma>0$, our method’s Pass@N becomes nearly identical to that of the Top-K baseline. In practice, this noise parameter provides the user a simple way to adjust the diversity–accuracy balance.
>
> >### **Q2. Please clarify how many reverse diffusion steps (NFE) your method requires for fair comparison with other baselines.**
>
> NFE is equal across our method and all baselines. As described in Algorithm 1 (Generalized Sampling of MDMs) and Appendix B.2, all sampling methods—whether our learned policy $g_\phi$ or rule-based heuristics ($g_{\text{conf}}$, $g_{\text{rand}}$, … ) shares same algorithm, therefore require exactly one model inference per unmasking step, and have the same NFE.
>
>
> >### **Q3. Please provide qualitative examples showing how the learned unmasking order on SUDOKU differs from max-confidence.**
>
>
> Thank you for the helpful suggestion. We have added qualitative visualizations comparing the learned unmasking order with max-confidence on SUDOKU. These examples are now included in the revised paper.
>
>
>
> We sincerely appreciate the reviewer’s suggestion, which helped us identify a meaningful future direction and further strengthen the paper. We would also be grateful if the reviewer could take into consideration our **theoretical contributions**: as acknowledged by reviewers (`DEPM`, `9cnV`, `BVXC`), our work formalizes the fundamental ordering challenge in MDMs, establishes an RL-based mathematical framework to address it, proves performance improvement guarantees over a reference policy, and provides a practical surrogate objective that enables implementation in real systems. We hope our answer clearly addresses the reviewer’s question, and if the reviewer has further questions, we are happy to answer them.

---

### Official Review · Reviewer_DEPM · 2025-11-01

**Soundness:** 3
**Presentation:** 3
**Contribution:** 2
**Rating:** 4
**Confidence:** 4

**Summary:**

The paper proposes learning an unmasking position-selection policy for masked diffusion language models (MDMs) by casting denoising as a KL-regularized Markov Decision Process and optimizing a GRPO-style objective against a strong reference scheduler (e.g., max-confidence or Top-K). The authors prove (i) fixed-point convergence that improves expected reward over the reference and (ii) a “reference-KL tightening” result showing the terminal output distribution becomes closer to the data than that induced by the reference policy [Thm. 1–2]. To make training tractable, they derive a surrogate token-level objective (LUPO) whose gradient aligns with the intractable output-level objective and instantiate realizations for different references [Eq. (19), Table 1]. Empirically, a lightweight policy model (≈134M) plugged into a frozen 8B MDM improves accuracy across SUDOKU, ZEBRA, GSM8K, and MATH500 over random, margin, entropy, and max-confidence schedules, and adds gains on top of diffu-GRPO post-training.

**Strengths:**

- Clear problem focus: unmasking order materially affects discrete diffusion inference; paper formalizes learning this policy rather than relying on heuristics.

- Theoretical contributions: output-level GRPO formulation with stated convergence and a KL tightening guarantee relative to a reference policy; proofs sketched in main text and detailed in Appendix C.

- Practical surrogate: token-level objective with clipping and an empirical divergence estimator, plus realizations for max-confidence/softmax/Top-K with explicit training recipe.

- Solid empirical signal: consistent wins vs. strong baselines on four benchmarks (e.g., SUDOKU +11.2% over max-confidence; GSM8K +1.9%) and compatibility with diffu-GRPO (GSM8K +1.3% over max-confidence).

- Efficient design: trains a small policy while keeping the 8B MDM frozen; memory-efficient algorithm described (Algorithm 2).

**Weaknesses:**

- Sec. 2 reverse-process mismatch is under-explained. The section claims that there exist states where the true reverse process and the model’s reverse process diverge, but the accompanying example is hard to follow and it’s unclear what concrete training pathology this implies. Specifically, how this mismatch arises under the stated pretraining objective, how it manifests during inference-time unmasking (e.g., biased token selection vs. compounding calibration error), and whether it is an artifact of modeling choices (masking schedule, likelihood factorization) are not clearly connected.

- Evaluation breadth. Benchmarks are primarily small-format puzzles and math QA; no natural-language generation or long-form tasks to test the generality of unmasking policy beyond structured reasoning.

**Questions:**

1. In Sec. 2, please (i) formalize the exact training objective under which the “true vs. model reverse-process” divergence occurs, (ii) explain the mechanism by which this leads to inference-time errors in the unmasking policy (with a minimal worked example), and (iii) discuss whether the issue can be mitigated during pretraining (e.g., via alternative noise schedules, calibration terms, or reverse-process consistency regularization) rather than deferred to policy learning at inference.

2. Does the learned policy transfer across MDM backbones (e.g., Dream-7B), or between tasks without re-training? Any negative transfer?

---

> ### Author Response · Authors · 2025-11-22
>
> We thank the reviewer for acknowledging various strengths of our work and the valuable feedback that strengthens our paper. The segments of the revised paper that reflect the received feedback are marked in red.
>
> >### **W/Q1. Provide a detailed explanation of the reverse-process mismatch in Section 2.**
>
> We thank the reviewer for the opportunity to clarify this point. Since Section 2 is intended as a concise preliminary overview, providing full technical details there would make the exposition heavy. To assist readers, we have added a clarifying sentence in Section 2 that directs them to Appendix A, where a more complete explanation addressing the reviewer’s questions is now included in the revised version.
>
> **W/Q1.1. Why do there exist states the MDM reverse process diverges from the true reverse process? (Formalize the exact training objective that occurs)**
>
> The MDM training objective (Eq. (1) in the main paper) is the KL divergence between the model reverse process and the true reverse process [2, 3], which is well known as diffusion loss and NELBO of true data distribution. Formally, Eq. (1) is equal to minimizing $D_{\text{KL}}(p(x_s|x_t,x_0),\pi_\theta(x_s|x_t))$. Kim et al. [1] prove that for a non-zero measure of masking patterns, the true conditional corresponds to subproblems that fall into the planted CSP hard phase, where no polynomial-time algorithm can approximate these conditionals. Thus, we cannot find an optimal model that makes MDM pretraining loss (reverse KL divergence) to be 0, and Kim et al. also show empirical evidence that the MDM reverse loss remains high on text and L&O-NAE-SAT datasets, consistent with this theoretical limitation.
>
> [1] Kim et al. “Train for the Worst, Plan for the Best: Understanding Token Ordering in Masked Diffusions,” ICML’25 (Outstanding paper award)
>
> [2] Sahoo et al. “Simple and Effective Masked Diffusion Language Models,” NeurIPS’24
>
> [3] Zheng et al. “Masked diffusion models are secretly time-agnostic masked models and exploit inaccurate categorical sampling,” ICLR’25
>
> **W/Q1.2. What causes this divergence? (pretraining objective vs. inference vs. modeling choices?)**
>
> The divergence is **not** a consequence of the particular ELBO form, nor of architectural or modeling choices, nor of inference-time ordering. Rather, it is structural to the Any-order generation problem. Random masking induces conditional prediction tasks. Following Kim et al. [1], a fraction of these induced subproblems are computationally equivalent to planted constraint satisfaction problems in the hard phase [4]. In this phase, the true conditional reverse distribution is computationally unreachable by any polynomial-time learner, independent of architecture, schedule, or likelihood factorization. Thus, the divergence originates from the problem class itself, not from the specifics of MDM pretraining, inference, or modeling.
>
> [4] Krzakala & Zdeborova, “Hiding Quiet Solutions in Random Constraint Satisfaction Problems,” Physical Review Letters ’09.
>
> **W/Q1.3. How does this divergence manifest at inference time? (minimal worked example)**
>
> We kindly direct the reviewer to the Sudoku example provided in Figure 10.(b) of the revised paper, which illustrates how reverse-process divergence manifests as inference errors in practice. In the example, the max-confidence heuristic selects an unmasking position whose correct value is not yet identifiable. The base model (LLaDA) then predicts a token that does not violate any explicit Sudoku rule at that moment, yet places the puzzle into a state from which no valid solution is reachable. Subsequent predictions, therefore, fail, resulting in a final output that cannot occur under the true data distribution. This demonstrates that when the learned reverse process deviates from the true reverse process, the model may generate states outside the support of $p_{\text{data}}$, ultimately failing to reconstruct the target distribution.
>
> Furthermore, prior theoretical work [5] also has shown that full sequence (sentence) error is lower bounded by a high value when a small training error (reverse KL divergence) exists (Assumption 4.1 & Theorem 4.4 in the referred paper).
>
> [5] Feng et al., “Theoretical Benefit and Limitation of Diffusion Language Model,” Arxiv’25

---

> ### Author Response · Authors · 2025-11-22
>
> **W/Q1.4. Can this mismatch be solved during pretraining? (e.g., noise scheduling)**
>
> Because the divergence arises from the structural hardness of Any-order generation, no choice of noise schedule, calibration, or consistency regularization can eliminate it in principle.
>
> Meanwhile, we agree that such techniques can mitigate the problems. For example, if the forward and reverse noise scheduler forces the sampled problem that the base model can solve easily, the model will tend to avoid such hard problems. However, we want to clarify that this direction differs from the primary motivation of our work. Rather than modifying the pretraining regime or forward process, our goal is to improve inference-time behavior of already pretrained large MDMs by learning a better unmasking policy.
>
> **W/Q1.Additional comment**
>
> Our reliance on Kim et al. [1] is intentional: their result identifies that Any-order MDM intrinsically contains computationally hard conditional prediction tasks. Importantly, despite this impossibility, empirical evidence (e.g., LLaDA) shows that simple rule-based samplers such as max-confidence and max-margin outperform ARM-style generation, even though MDM pretraining is harder. Our contribution is to go beyond these heuristics and provide **theoretically grounded unmasking policies** (Theorem 1, Theorem 2) that improve the match to $p_{\text{data}}$ relative to any fixed rule-based sampler.
>
>
> >### **W2. Lack of natural language generation task**
>
> Thank you for the helpful suggestion. GRPO-style methods are fundamentally designed for tasks where a verifiable reward (i.e., correctness) is available, and our paper also focuses on this setting, with the theoretical results aligned accordingly. We understand, however, that the reviewer’s intention is to check whether an unmasking policy can be applied more generally to MDMs, including natural language generation. For this reason, we conducted an additional experiment for natural language generation.
>
> We used the Natural Questions benchmark [6], which contains real user queries issued to Google Search and covers a wide range of question types. In this experiment, we replaced the verifiable reward $r(\cdot)$ in our $\mathcal{L}_{\text{UPO}}$ objective with the average log-likelihood of the generated text under Qwen-3 32B. Since Qwen-3 32B is significantly stronger than LLaDA 8B, the log-likelihood assigned by this model can serve as a meaningful indicator of natural language generation quality, consistent with the common practice of evaluating smaller models using the perplexity of a larger ARM. In this setting, we observed that the log-likelihood assigned to the sequences generated by the unmasking policy steadily increases during training, **indicating improved natural language generation performance**. Following a similar request from reviewer `cCbk`, we also performed the same experiment on GSM8K and again observed an increase in log-likelihood. The training dynamics for both datasets and a detailed explanation can be found in Appendix D.2 of the revised paper.
>
> These results suggest that our method can be used for more general tasks as well. As future work, it would be exciting to extend this direction to general reward within the general dataset to obtain even stronger and more general unmasking policies.
>
> [6] Kwiatkowski et al., “Natural Questions: A Benchmark for Question Answering Research,” ACL’19.
>
> >### **Q2: Could the learned unmasking policy transfer across different MDM backbones or across different tasks? Any negative effects?**
>
> As shown in the model architecture of Figure 2, the unmasking policy model is explicitly conditioned on the feature representations produced by the MDM backbone. Because the policy depends on these backbone-specific features, a policy trained on one MDM cannot be directly transferred to another backbone.
>
> Similarly, transferring the policy across different tasks is also difficult. This limitation is shared by most RL–based post-training methods, since the learned policy tends to specialize in the reward and data distribution of the target task.
>
> Nevertheless, our model retains the advantages mentioned by the reviewer: it is lightweight, memory-efficient to train, and additionally, can be detached. These benefits are not available to standard GRPO-style methods, which must fine-tune the entire model for each task. In particular, the unmasking policy model can simply be disabled when it is not suitable for a given task. We would be glad if the reviewer considers this a practical strength of our approach.
>
> We hope our answer clearly addresses the reviewer’s question, and if the reviewer has further questions, we are happy to answer them.

---

> > ### Comment · Reviewer_DEPM · 2025-11-28
> >
> > Thank you for the detailed response. I have carefully read your discussion, which has resolved most of my concerns, and I will take it into account in my final evaluation.

---

> > > ### Author Response · Authors · 2025-11-28
> > >
> > > Thank you for your time and constructive feedback. We are glad to hear that most of your concerns have been resolved, and we appreciate you considering our responses in your final evaluation.

---

### Author Response · Authors · 2025-11-22
**Response to all reviewers**

We are grateful for the constructive feedback provided by the reviewers. We propose a KL-regularized framework for learning unmasking policies in masked diffusion models (MDMs), and provide theoretical guarantees showing that the learned policy can outperform the reference policy.

Reviewers highlighted several strengths of the work, including the clear problem formulation (`DEPM`, `BVSC`), the theoretical improvement guarantee over the reference policy (`DEPM`, `9cnV`), the practical surrogate objective enabling tractable optimization (`DEPM`, `BVSC`), and the efficient lightweight model design (`DEPM`, `BVSC`). Reviewers also noted that the approach performs consistently well across multiple benchmarks (`DEPM`) and that learning an unmasking order represents a meaningful contribution to modern diffusion-based generation (`cCbK`).

We appreciate that the suggestions provided by reviewers have meaningfully strengthened the paper, and we summarize the key improvements below:
- **Scalability of unmasking policy**: Reviewer `DEPM` raised the question of whether the proposed unmasking policy could extend to natural language generation, and reviewer `cCbK` asked whether the framework could operate under preference-style rewards rather than only verifiable correctness signals. Motivated by these suggestions, we conducted additional experiments where the policy is trained using the log-likelihood of Qwen-3 32B as the reward. The results (Appendix D.2) show that perplexity improves steadily on both Natural Questions and GSM8K, and on GSM8K the accuracy also surpasses the max-confidence baseline. While these experiments go beyond the exact theoretical setting of our GRPO formulation, they provide promising evidence that the method can scale toward more general reward signals and broader language settings, and establish a clear direction for future work.
- **Strengthening theoretical clarity**: Reviewer `9cnV` pointed out areas where theoretical assumptions and proof structure could be ambiguous to readers. Accordingly, we improved the readability of the proof, added additional interpretation of the theorem, and provided justification for why the assumptions in Theorem 2 and Proposition 3 hold.

Additionally, we included a new Sudoku visualization (Appendix F) that highlights why unmasking policy matters and how learned ordering differs from heuristic schedules.

All reviewer comments were addressed, and the requested experiments have been incorporated. **Modified content is marked in red text** in the revised manuscript. We thank the reviewers again; feedback provided helped refine the paper and expand its scope in meaningful ways.

---

### Meta-Review · Area_Chair_frh6 · 2026-01-06

**Summary:**

The key decision factors were (i) whether the proposed KL-regularized framework (GRPO) for learning unmasking policies in masked diffusion models is theoretically sound and practically useful, (ii) whether it yields meaningful and robust improvements over strong rule-based schedulers (e.g., max-confidence, margin) beyond narrow settings, and (iii) whether the framework can plausibly extend to broader reward signals beyond strictly verifiable correctness. Reviewers generally viewed the formulation as well-motivated and the theoretical guarantees as valuable, but scrutinized the alignment between the theoretical setting and the practical reward design, the magnitude and generality of the empirical gains, and the extent to which the learned policy can operate under more realistic preference- or model-based rewards. After rebuttal, the authors clarified key assumptions, strengthened proofs, and added experiments that directly targeted these points, so the remaining debate mainly concerns the modest gains on language tasks and limited transferability rather than core correctness.

**Reviewer Concerns:**

Addressed concerns: The rebuttal materially improved theory–practice alignment by adding binary-reward experiments that match the assumptions used in the analysis, and by clarifying proof structure, assumptions, and the interpretation of the KL-related guarantee. The authors also broadened the empirical support for compatibility with non-verifiable rewards by adding Qwen-based log-likelihood rewards on Natural Questions and GSM8K, and provided additional diagnostics including Pass@K behavior, NFE clarification, and qualitative visualizations of learned unmasking policies (e.g., on Sudoku) to make the policy behavior more interpretable.

Remaining concerns: While most technical concerns were resolved, performance gains on language benchmarks remain modest and appear less consistently separated from strong training-free heuristics, especially outside structured puzzles.

**Reviewer Scores:**

DEPM: likely 6
BVXC: likely 8
cCbK: likely 4 or 6
9cnV: likely 8

---

### Decision · Program_Chairs · 2026-01-26

Accept (Poster)